**Understanding the Role of Contrails and Contrail Cirrus in Climate Change: A Global Perspective**

Dharmendra Kumar Singh, Swarnali Sanyal, Donald J. Wuebbles

Department of Climate, Meteorology & Atmospheric Sciences (CliMAS), University of Illinois Urbana-Champaign, Urbana, IL 61802, Illinois, USA

*Correspondence to*: Dharmendra Kumar Singh (dksingh8@illinois.edu)

**Abstract**

Globally, emissions from aviation affect Earth's climate via complex processes. Contrail cirrus and carbon dioxide emissions are the largest factors contributing to aviation's radiative forcing on climate. Contrail cirrus, like natural cirrus clouds, impacts Earth's climate. Even with the extensive ongoing research, the relative importance of the climate effects of contrails compared to other aviation effects on climate still has major uncertainties requiring further research. Contrail cirrus encompasses linear contrails and the associated cirrus clouds; these are characterized by ice particle properties, e.g., size, concentration, mixing, extinction, ice water content, optical depth, geometrical depth, and cloud coverage. The climate impact of contrails may intensify due to projected increases in air traffic. The radiative forcing from global contrail cirrus has the potential to triple and could reach as much as 160 mW m$^{-2}$ by 2050. This projection is based on anticipated growth in air traffic and a potential shift to higher altitudes. The future climate impact of contrail cirrus is influenced by factors like the magnitude and geographical spread in air traffic, advancements in fuel efficiency, the effects of the use of alternative fuels, and the effects of the changing climate on the background atmosphere. This study reviews the microphysical processes affecting contrail formation and the aging of contrails and contrail-cirrus. Furthermore, the study explores global observational datasets for contrails, current analyses, and future projections and will aid in evaluating the effectiveness and trade-offs associated with various mitigation strategies. The research highlights gaps in knowledge and uncertainties while outlining research priorities for the future.

Keywords: Aviation, contrails, contrail cirrus, radiative forcing, climate model, climate change

## 1 Introduction

The increase in the atmospheric carbon dioxide ($CO_2$) concentration has had the most substantial impact on human-induced climate change over recent decades (IPCC, 2021). Emissions from aviation are a minor but significant contributor to these changes in climate, especially because of emissions of $CO_2$ and the effects of contrail formation from emitted water vapor ($H_2O$). Aviation expansion is outpacing economic growth, with projections indicating that over the next two decades, the demand for aviation could grow to about 3 times its present level (Wuebbles et al., 2007), with the International Civil Aviation Organization having previously projected a 4.3% annual growth to 2050 (ICAO 2012). Recent analyses by Boeing and Airbus suggest slightly lower rates of increase of 3.6-3.8% (Boeing, 2022; Airbus, 2023). These analyses suggest that aviation could become an even more important contributor to climate change in the future. Taking both $CO_2$ and non-$CO_2$ effects into account, The global aviation industry comprises about 3.5-4% of the current human-made (anthropogenic) activities forcing on climate (Lee et al., 2020; Klöwer et al., 2021; Grewe et al., 2021).

Aviation-induced cirrus is one of the most significant radiative forcing contributors from the aviation sector. (IPCC, 1999; Lee et al., 2009, 2021; Brasseur et al., 2016). However, considerable ambiguities persist, stemming from various origins, such as our incomplete understanding of cirrus cloud characteristics, their geographical distribution, and their life span. (Kärcher 2018; Burkhardt et al., 2018; Schumann and Heymsfield 2017). With current aircraft largely burning fossil fuel, $CO_2$ emissions are the other major forcing. Additionally, $NO_x$ (oxides of nitrogen) emission also has a notable impact through the production of tropospheric ozone, but this is partially counteracted by chemical feedback effects on concentrations of atmospheric methane ($CH_4$). Fig. 1 (Lee et al., 2021) provides the most current assessment of the climate-forcing effects resulting from different aviation emissions on a global scale, spanning from 1940 to 2018; these effects are shown in terms of radiative forcing (RF) and effective radiative forcing (ERF). Radiative forcing in the context of contrails is due to the net radiative energy flux at the top of the atmosphere (TOA) caused by the presence of contrails in the atmosphere (Fuglestvedt et al., 2010). A positive radiative forcing (RF) signifies an increase in atmospheric warming, as depicted by the red bars in Figure 1. ERF, on the other hand, adjusts the RF by accounting for rapid responses occurring within the Earth's climate system. While RF and ERF provide valuable insights, it's important to acknowledge that these are global metrics. Contrails primarily affect the upper troposphere and are short-lived, leading to potentially larger regional effects, particularly in densely populated areas with high air traffic.

In the scientific literature, contrail cirrus is defined as encompassing both linear contrails and the resulting formation of cirrus clouds. Figure 2 provides an overview of aviation exhaust plume emissions and factors influencing contrail formation. The characteristics of contrail cirrus include average ice particle sizes and concentrations, extinction, ice water content, optical depth, geometrical depth, and contrail coverage. Integral contrail properties involve parameters such as contrail cirrus volume, total number of ice particles, overall ice water content, and total extinction (area integral of extinction) per contrail length.

While significant progress has been made in recent years to incorporate contrail effects into global climate models, as evidenced by the pioneering work of Burkhardt and Kärcher (2011) and the continued advancements by others, like Bock and Burkhardt (2016), Bier and Burkhardt (2022), Chen and Gettelman (2013), and Schumann et al. (2015), uncertainties remain in how well these models capture the long-term influence of contrails on cirrus clouds. These uncertainties arise from

limitations in our understanding of how contrails age and spread (Bickel et al., 2020), the properties
of the tiny ice crystals that make them up, and the behavior of contrails in regions with high ice
content. Further research is crucial to address these uncertainties and improve the accuracy and
reliability of contrail modeling for future climate projections.
The generation of persistent contrails depends on coming across ice-supersaturated conditions
along a flight path, and these conditions show variability in both space and time within the
troposphere and tropopause region (Irvine et al., 2013; Lamquin et al., 2012; Bier et al., 2017).
Estimating the RF from contrail cirrus requires knowledge of complex microphysical processes,
radiative transfer, and the interaction of contrails with background cloudiness (Burkhardt et al.,
96  2010).

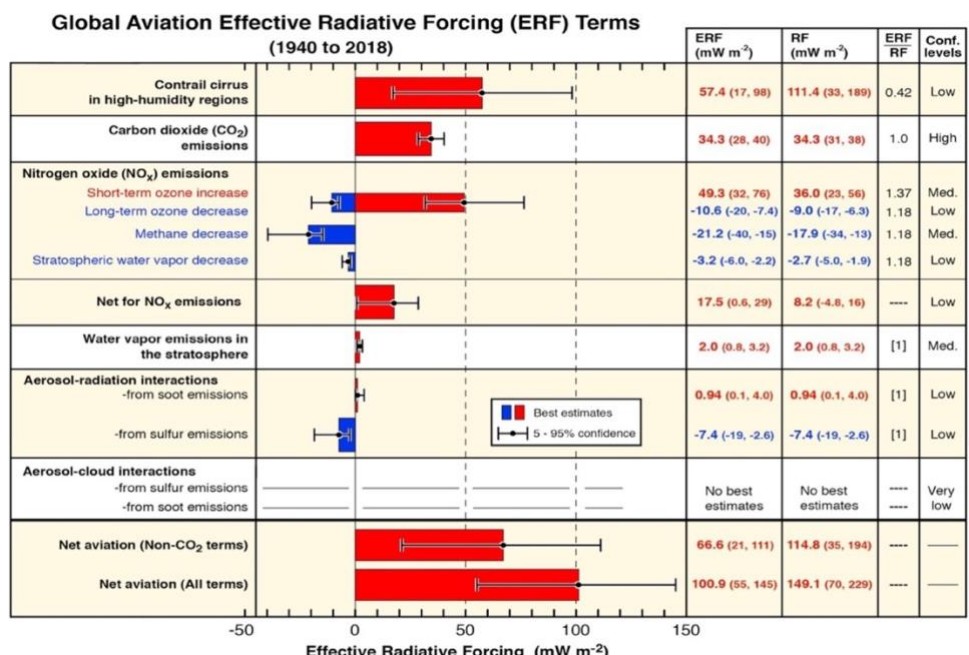

**Figure 1.** Current best evaluation of the climate forcing between 1940 and 2018 from commercial
aviation for different types of emissions. ERF accounts for short-term feedback in the climate
system not accounted for in the traditional RF evaluation.  Adapted with permission from Lee et
al. (2021).

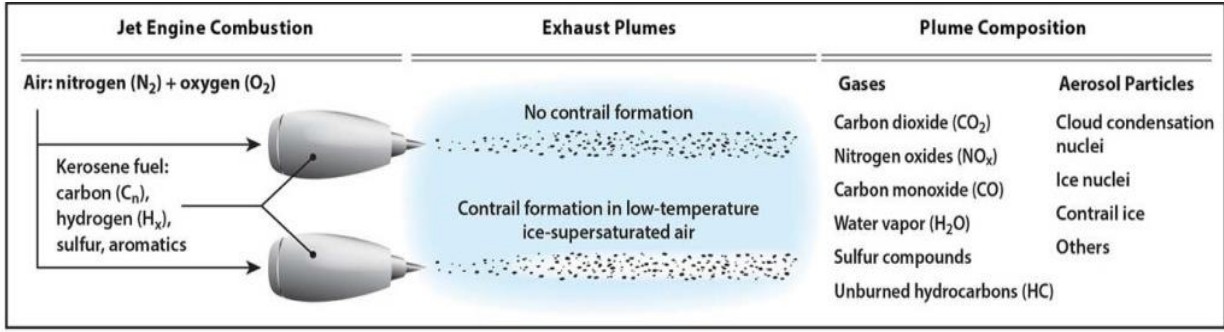

**Figure 2.** Schematic overview of exhaust plumes (contrail formation in low temperature) and their
composition.  Adapted with permission from Lee et al. (2021).

This review article aims to provide a comprehensive overview of our understanding of contrails,
encompassing their formation, progression, and the consequent repercussions on Earth's climate
system. In the process, the microphysical processes underlying contrail formation are explored
along with their representation in the simulation of contrails in current climate-chemistry models
of the global atmosphere. Furthermore, we address the uncertainties surrounding this topic and
identify areas requiring further research in the future.

## 124 2 Microphysics of Contrail Formation, Aging, And Transition to Contrail Cirrus

Microphysical processes play a pivotal role in the formation of contrails and their evolution in the
atmosphere. This section is aimed at discussing these processes, their effects on the lifetime of
contrails, and the possible transition of contrails to contrail cirrus in the upper troposphere.

### 128 2.1 The Physics of Contrail Formation and Aging

The fundamental principles for determining the formation of contrails were independently
developed by E. Schmidt in 1941 and H. Appleman in 1953. Contrails appear when the hot and
moist exhaust from aircraft quickly mixes with the cold and humid surrounding air, causing the
humidity in the exhaust gases to exceed the saturation point for liquid water (Appleman, 1953;
Schmidt, 1941; Schumann, 1996). A schematic representation of the Schmidt-Appleman criterion
for contrail formation is shown in Figure 3. In this diagram, the two solid curves depict saturation
levels concerning liquid water (upper curve) and ice (lower curve). The phase trajectory of the
mixture, consisting of exhaust gases and ambient air, follows a straight line from the upper right
to the lower left in the e-T diagram, where 'e' represents the partial pressure of water vapor in the
mixture, and 'T' represents its absolute temperature (as denoted by the dashed lines). The path that
lines up with the water saturation curve (dotted) specifies the highest temperatures at which
contrail formation can take place. When the trajectory terminates in an ice-supersaturated state,
the persisting contrails have the potential to disperse and transform into contrail cirrus.
In cases where the conditions do not meet the criteria, the contrail formed will have a brief lifespan,
lasting only a few minutes. The mixing process is presumed to occur isobarically, resulting in a
straight-line mixing (phase) trajectory on an e-T diagram. The Schmidt–Appleman criterion
(SAC), which is primarily based on thermodynamics, establishes the threshold temperature for
contrail formation. This threshold is determined by several factors, such as ambient air pressure,
humidity, the amount of water and heat released by the aircraft per unit of fuel, and the overall
efficiency of the aircraft engine's propulsion system. (Jensen et al., 1998; Schumann, 1996).
Accurate assessments of contrails necessitate precise data on ambient meteorological conditions.
The persistence of contrails is contingent on meeting a specific humidity threshold, as defined
below.
The persistence of contrails hinges on the atmospheric-specific humidity. Temperature and vertical
motions within the atmosphere play key roles in determining this humidity. Additionally, the
optical characteristics and lifespan of contrails are key aspects of contrail formation and are
influenced by various meteorological factors. These factors include turbulence, vertical motions,
wind shear, and the presence of ambient cirrus clouds. The size and duration of the ice-
supersaturated region, the availability of water vapor for ice crystal growth, and consequently, the
sedimentation of these ice crystals within the contrail are also essential considerations (Bock and
Burkhardt, 2016). Crucially, the radiative forcing (RF) and the ensuing climate impacts depend on
a multitude of meteorological parameters. Additionally, factors such as the albedo and the
brightness temperature of the atmosphere in the absence of contrails play a significant role, as
outlined by Schumann et al. (2012).
Upon reaching the ambient temperature through the process of mixing with the surrounding air,
contrails undergo changes in size depending on the ambient humidity. When the ambient relative
humidity remains supersaturated concerning ice (RHice >100%), contrails experience growth in
ice water content and can remain detectable for up to 5 hours or even longer, as highlighted in
various studies (Gierens and Vázquez-Navarro, 2018; Schumann and Heymsfield, 2017; Bier et
al., 2017). This extended persistence contrasts with the mean lifetime of contrails, typically noted
to be around 2-3 hours (Schumann et al., 2015; Vázquez-Navarro et al., 2015; Tesche et al.,
2016; Sausen et al., 1998). Contrails, under such conditions, have the potential to disperse and
transform into thin cirrus layers. On the other hand, when the ambient relative humidity is not
supersaturated concerning ice, contrail ice particles sublimate and dissipate. The timescale of this
sublimation and dissipation process is contingent on the sizes of the contrail ice particles and the
ambient air RHice (Schumann, 2012).
Combinations of satellite information and ground-based lidar measurements, such as in the Atlas
et al. (2006) case study, showed a lifetime of more than 2 hours and a mean optical thickness
(integrated extinction coefficient over a vertical column of unit cross section) of 0.35. Duda et al.
(2004) studied the development of contrail clusters over the Great Lakes and derived optical
thickness (optical depth, τ) from 0.1 to 0.6 for contrails that lasted several hours (Kärcher et al.,
2009). Graf et al. (2012) studied the cirrus cover cycle and observed timescales between 2.3 to
4.1 hours for contrail cirrus and 1.4 to 2.4 hours for linear contrails.

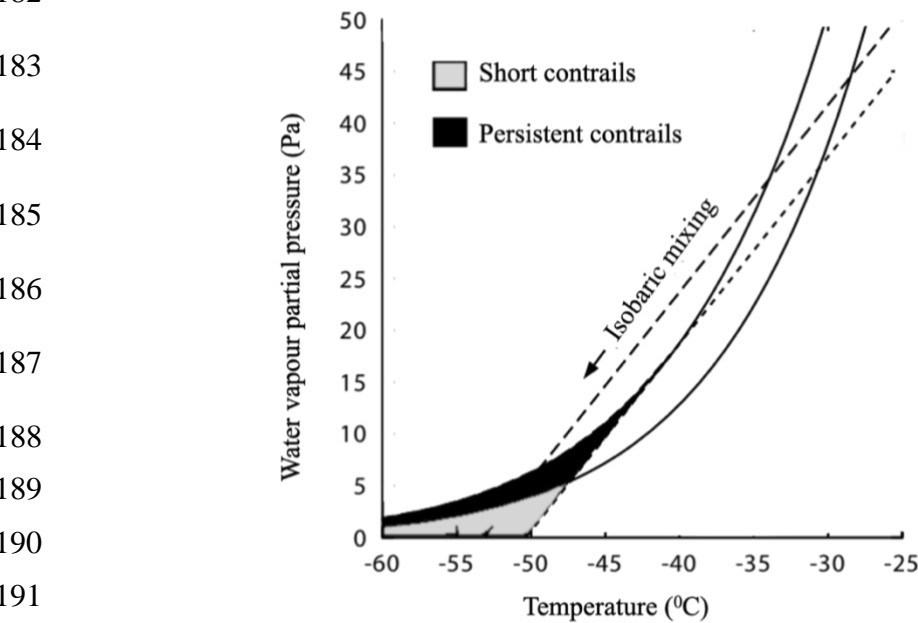

**Figure 3.** Phase diagram with mixing lines for aircraft exhaust Adapted from Gierens et al. (2008)
(Licensed under CC BY 4.0).
The critical factor influencing this contrasting behavior lies in the concept of ice supersaturation.
Ice supersaturation refers to conditions where the ambient humidity exceeds the saturation point

for ice, even at temperatures below freezing. These conditions are essential for the persistence and growth of contrails. While the initial formation process might occur at varying humidity levels, studies like Ovarlez et al. (2002) highlight the presence of ice supersaturation within cirrus clouds, which can significantly impact contrail persistence. Jensen et al. (2001) revealed a surprisingly high frequency of ice supersaturation in the upper troposphere, even in the absence of visible ice clouds. This finding highlights the importance of considering ice supersaturation for accurate assessments of contrail persistence and its potential transformation into cirrus clouds.

Upon reaching the ambient temperature through the process of mixing with the surrounding air, contrails undergo changes in size depending on the ambient humidity. When the ambient relative humidity remains supersaturated concerning ice (RHice >100%), contrails experience growth in ice water content and can remain detectable for up to 5 hours or even longer. The microphysical properties of contrails, including ice crystal size distribution and habit, significantly influence their radiative effects and persistence. These properties are determined by the initial conditions during contrail formation and subsequent processes. Ice crystal shapes in contrails are primarily randomly oriented, small ice crystals, with a possibility of some horizontally oriented plates existing as well. This finding is supported by Iwabuchi et al. (2012) who analyzed contrails using a combination of MODIS and CALIPSO satellite data. Their study showed that contrails have larger backscattering coefficients than neighboring cirrus clouds, indicative of smaller ice crystals. The analysis also suggested a tendency for stronger depolarization when ice crystals are small, with a mean linear depolarization ratio (LDR) of approximately 0.4-0.45 for young contrails and contrail cores (Iwabuchi et al., 2012).

There has been ongoing uncertainty regarding whether persistent contrails exclusively form in cloud-free supersaturated areas or if they can also develop within existing cirrus clouds (Burkhardt et al. 2008). Subsequent modeling work (Tesche et al. 2016) suggested that persistent contrails indeed have the potential to form within cirrus clouds. While the formation of contrails is commonly observed in clear skies, they also emerge in conditions where the sky is covered with thin or even subvisible cirrus clouds (Immler et al. 2008).

The temperature threshold for contrail formation, known as the SAC threshold, is slightly higher in cirrus clouds compared to clear air (Gierens, 2012). This difference is attributed to the additional humidity introduced by the ice water content (IWC) from cirrus clouds (Verma and Burkhardt 2022). In situ measurements have indicated high ice supersaturation both inside and outside cirrus clouds (Comstock et al. 2004). Moreover, observational evidence suggests that contrails embedded within cirrus clouds are not significantly thinner than contrails forming in clear air (Poellot et al. 1999).

A reliable estimation of the radiative effects stemming from contrail cirrus relies heavily on their optical properties, which are intricately linked to their microphysical properties and age, as well as their geographical distribution. The microphysical characteristics of contrail cirrus at various plume ages, as observed in diverse airborne campaigns, have been systematically compiled and detailed (Schröder et al. 2000; Schumann et al. 2017; Chauvigné et al. 2018). Fresh contrails, typically observed in plumes around 2 minutes old, exhibit distinct features, including an ice crystal number concentration reaching thousands per cubic centimeter and ice crystals measuring up to a few micrometers in diameter. This characterization is consistent with observations (Märkl et al., 2024, Bräuer et al., 2021a; Petzold et al., 1997). Contrails in their initial 2-5 minutes of existence have been frequently measured (Voigt et al., 2011; Gayet et al., 2012). The distinctive feature of these young contrails lies in their notably high ice crystal number concentration of small

ice particles, making them easily discernible from natural cirrus formations. However, contrails often coexist with natural cirrus and may be incorporated into thin or subvisible cirrus (Kübbeler et al., 2011; Gierens, 2012; Unterstrasser et al., 2017). This coexistence poses a challenge in distinguishing between aged contrails and natural cirrus, complicating efforts to clarify the contribution of contrail cirrus to the radiative balance. Contrail cirrus is characterized by a low ice water content (IWC) ranging from 0.1 to about 10 mg m$^{-3}$, a feature it shares with natural cirrus of in-situ origin, as observed by Schumann et al. (2017). Ice crystals within these clouds have formed and enlarged in an ice-cloud environment (Luebke et al., 2016; Krämer et al., 2020). In contrast to contrail cirrus and in situ-origin cirrus, cirrus clouds initiating from liquid often yield a higher IWC since their ice crystals originally form as liquid drops (Krämer et al., 2016, 2020) in a warmer atmosphere with an ambient temperature exceeding 235 K (-38°C), particles undergo freezing as they are lifted into the cirrus temperature region of the atmosphere.

Chauvigné et al. (2018) applied a method based on principal component analysis to differentiate between particles in contrail cirrus at various stages and those observed in natural cirrus during the CONCERT 2008 campaign (Voigt et al., 2010). The success of the campaign stemmed from the fact that contrails sampled during the CONCERT initiative were relatively young and exhibited greater distinctiveness compared to natural cirrus formations. Despite this success, the comprehensive acquisition of all necessary optical and microphysical parameters proved challenging during single aircraft campaigns, limiting the widespread application of this method. The CONCERT dataset is relatively small, encompassing approximately 4.0 hours of sampling time in total (Kübbeler et al., 2011). A commonly held assumption regarding the formation and evolution conditions of contrail cirrus is that it tends to persist particularly in ice-supersaturated regions (ISSRs) (Kärcher, 2018).

The collective presence of commercial, military, and other aircraft contributes to a global increase in cloudiness, primarily facilitated by the formation of persistent contrails when the surrounding atmosphere reaches supersaturation. Contrail cirrus exhibits both cooling and warming effects, with the nighttime impact being predominantly warming. Previous assessments of aviation's influence on climate (IPCC, 1999; Lee et al., 2009; Brasseur et al., 2016) were limited by the challenge of accurately quantifying the role of cloudiness arising from aging and spreading contrails (Minnis et al., 2013). The formation of a persistent contrail necessitates ice-supersaturated conditions along the aircraft's flight path. The life cycles of contrail cirrus are contingent upon the temporal and spatial scales of ice-supersaturated regions, which exhibit high variability in the troposphere and tropopause region (e.g., Lamquin et al., 2012; Irvine et al., 2013; Bier et al., 2017). Estimating the impact of contrail cirrus on upper tropospheric cloudiness requires the simulation of complex microphysical processes, contrail spreading, overlap with natural clouds, radiative transfer, and interaction with background cloudiness (Burkhardt et al., 2010).

Petzold et al. (2020) conducted a study investigating the frequency distribution of ice-supersaturated regions (ISSRs) through regular in-situ observations made by passenger aircraft across northern mid-latitude areas. Their research underscores the seasonal and geographical variability of ISSRs, indicating a higher likelihood of occurrence during winter and in specific geographic regions. This variability underscores the significance of considering the spatial and temporal distributions of ISSRs when assessing the potential for contrail formation.

Reutter et al. (2020) delved into the characteristics of ice-supersaturated regions using a combination of in-situ water vapor measurements obtained from the IAGOS research program and

data from ERA-Interim reanalysis. Their findings showcase the potential for validating reanalysis
data with high-resolution aircraft observations. This validation process is critical for enhancing the
accuracy of global models utilized in evaluating contrail formation and their subsequent climatic
impacts.
The ERF of contrail cirrus was estimated for 2011 (relative to an atmosphere without contrails) as
mWm$^{-2}$ by Boucher et al. (2013), with uncertainty ranging from 20-150 mWm-2. Lee et al.
(2021) presented a new estimate derived from the outcomes of global climate models
implementing process-based contrail cirrus parameterizations. Recent analyses by Lee et al. (2021)
of the current aviation fleet emissions evaluate the ERF through 2018 as 57 mWm$^{-2}$, with an
uncertainty range of 17-98 mWm$^{-2}$. Given the limited availability of independent estimates,
assessing uncertainty becomes crucial. This necessitates analyzing the sensitivities of relevant
processes and incorporating the uncertainties associated with the underlying parameters and fields.
Contrails form in the early exhaust plumes of airplanes during flight (Kärcher et al., 2015; aufm
Kampe, 1943; Weickmann, 1945; Schumann and Wendling, 1990; Schumann, 1996). This
happens when supercooled water droplets freeze into ice particles. At cruising altitudes, the
atmospheric relative humidity (RH) is usually too low to sustain liquid water droplets but can
support ice-phase particles (Gettelman et al., 2006; Lamquin et al., 2012). Consequently, freezing
must occur soon after droplet formation within the moist exhaust plume to create persistent
contrails. Appleman (1953) identified a key requirement for contrail formation, suggesting that
many aerosol particles in the plume serve as centers for water condensation, known as the "water-
saturation constraint." Another essential factor is the visibility constraint, which requires at least
$10^4$ cm$^{-3}$ within a plume age of 0.3 seconds for small-scale research aircraft (Appleman, 1953;
Kärcher et al., 1996). These are the main constraints for contrail formation.

## 2.2 Transition of contrails into contrail cirrus and contrail-cirrus-soot interaction

Contrail cirrus consists of elongated contrails trailing high-altitude aircraft and thin cirrus patches
formed by the dispersion of persistent contrails. The morphology of contrails evolves based on
factors such as humidity, shear, stratification, waves, turbulence, and radiative heating. Contrails,
individually and collectively, interact with other cirrus formations, giving rise to what is termed
"contrail cirrus" (Schumann and Wendling, 1990). The total extinction is less pronounced when
contrails overlap due to humidity competition. Consequently, the climate impact does not always
exhibit a linear correlation with air traffic density (Unterstrasser and Sölch, 2012; Bickel et al.,
2020). The transformation of a single contrail into a contrail cirrus is observable during aircraft
spiral flights, as depicted in Figure 4 (Haywood et al., 2009). "Contrail outbreaks" describe
scenarios where numerous aged and young contrails coexist, often spanning expansive areas within
the same airspace (Duda et al., 2001).
Soot emissions, composed of black carbon particles from aviation, have the potential to alter cirrus
properties independently of contrail processing, leading to the formation of "soot cirrus" (Lee et
al., 2010). Climate impact assessments of aviation-induced soot cirrus remain inconclusive due to
uncertainties in soot abundance and their ice-nucleating properties (Gettelman and Chen, 2013;
Zhou and Penner, 2014; Righi et al., 2021). Increased concentrations of small-sized cirrus particles
resembling aviation soot emissions have been detected in cirrus regions with dense air traffic
(Kristensson et al. 2000). It remains uncertain whether the soot enters the ice during the initial
nucleation process or at a later stage through scavenging. The exact mechanism of soot
incorporation into cirrus ice remains unclear. Uncertainties persist regarding whether the soot
enters the ice during the initial nucleation process or at a later stage through scavenging.
A recent study by Testa et al. (2024) examines the processing of aviation soot within contrails and
offered further insights into the potential influence of soot on ice nucleation. They specifically
investigate the ice nucleation ability of contrail-processed soot particles at cirrus temperatures and
found that these particles were generally poor ice nucleating agents. This suggests that soot cirrus
formed from processed aviation soot may have a weaker influence on cirrus cloud formation than
previously thought.
Dischl et al. (2024) demonstrate a significant reduction in non-volatile particle emissions from
aircraft engines using sustainable aviation fuel (SAF) compared to traditional jet fuel. This finding
indirectly supports the notion that some soot particles might remain dry throughout contrail
formation or that some ice particles might sublimate, as these processes could be influenced by
particle properties.
The sublimation of cirrus particles containing soot and sulfate may lead to the formation of soot
aggregates that can potentially act as efficient ice nuclei. A cirrus pattern observed near Munich,
Germany, on November 3, 2012, displayed characteristics suggesting an association with aged
soot plumes. The cirrus observed between 9.1 and 9.5 km in height, 8-10 km wide, and 35-50 km
long, exhibited a distinctive pattern resembling parallel line clouds. Back trajectories indicated that
soot was emitted upstream approximately 12 hours before the event by aircraft. The air ascended,
forming a cirrus about 4 hours before the event, lasting for about 1-2 hours before subsiding or
sublimating. Analyses propose that this could be cirrus formed on preactivated aircraft soot, but
certainty is limited, and it cannot be ruled out that the same pattern might have formed without air
traffic. (Schumann and Heymsfield, 2017).

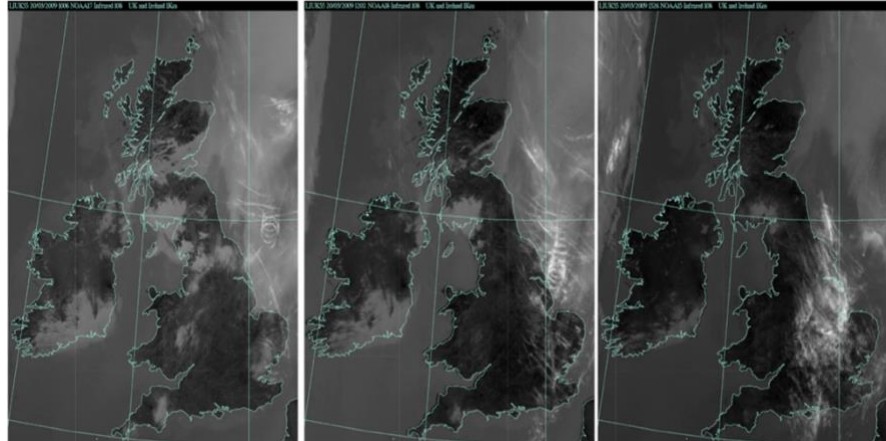

**Figure 4**. Evolution of contrail cirrus over 1, 3, and 6.5 hours. Contrail cirrus is identified
by bright white areas with low infrared (10.8 mm) brightness temperature. The satellite
scenes are from NOAA AVHRR for 3 UTC times: (left) 1006 (age after emission = 1 hour),
(middle) at 1202 (elapsed time: 3 hours), and (on the right) 1526 (elapsed time: 6.5 hours).
Adapted with permission from Haywood et al. (2009).


A comprehensive understanding of the interplay between contrails, cirrus clouds, and aircraft emissions is essential for accurate assessments of their combined effect on Earth's climate. Recent investigations have illuminated this complex relationship, providing valuable scientific insights. Notably, Urbanek et al. (2018) documented ice clouds exhibiting atypical characteristics over Europe, suggesting potential deviations from standard ice crystal behavior. Additionally, these clouds displayed lower ice supersaturation, hinting at a possible modification in the usual ice formation processes (Urbanek et al., 2018). Intriguingly, the spatial distribution of these cloud formations appeared to correlate with regions experiencing high air traffic. While a definitive causal link between these observations and contrail cirrus formation remains elusive, they do suggest a potential indirect influence stemming from aircraft emissions (Urbanek et al., 2018).

In contrast, Kärcher and colleagues (2021) examine the impact of soot particles emitted by aircraft. Kärcher et al. (2021) use modeling to study how these soot particles influence the birth of new cirrus clouds. they found that only a tiny fraction of the soot particles played a role in cloud ice formation via the freezing of liquid aerosol droplets. Consequently, cirrus clouds affected by soot displayed negligible deviations in overall cloud ice content and optical depth compared to their naturally formed counterparts (Kärcher et al., 2021). These findings question the accuracy of current climate models, suggesting they might be overestimating the global radiative impact of interactions between aircraft soot and expansive cirrus clouds.

In a related study, Wolf et al. (2023) delve into the radiative effect (RE) of cirrus clouds, shedding light on the pivotal role played by ice crystal properties. Their investigations unveiled that ice crystal size and concentration are the linchpins dictating the overall RE of cirrus clouds. Intriguingly, smaller ice crystals, often associated with contrails, wield a dual-edged radiative effect, inducing both cooling and warming influences. This intricate interplay generally tips the scale towards an overall warming effect exerted by cirrus clouds (Wolf et al., 2023).

Beyond the realm of ice crystal properties, Wolf et al. (2023) underscores the importance of other variables. Surface albedo emerges as a contributing factor, with cirrus clouds potentially inducing cooling effects over highly reflective surfaces, like ice, while warming effects over less reflective ones. Additionally, the presence of underlying liquid water clouds and the solar zenith angle also influence the radiative effect. Though their focus primarily centered on plane-parallel clouds, Wolf et al. (2023) acknowledge the potential significance of three-dimensional scattering effects, particularly under high sun angles.

### 2.3 Evolution of contrails and stages of aviation-induced cloud evolution

The review by Paoli and Shariff (2016) examined the primary physical processes and simulation efforts involved in four distinct phases of contrail evolution. These phases are commonly categorized as the jet, vortex, vortex dissipation, and diffusion phases. Contrail evolution is conveniently divided into these four regimes for clarity (Gerz et al., 1998) (see Fig.5).

In the initial seconds after emission (the jet regime), the vortex sheet shed by the wings rolls up into a pair of counter-rotating vortices known as the primary wake. Simultaneously, newly formed ice crystals in the engine exhaust become entrapped around the cores of these vortices. Following this, in the minutes that follow (the vortex regime), the vortices descend into the atmosphere due to their mutually induced downward velocity.

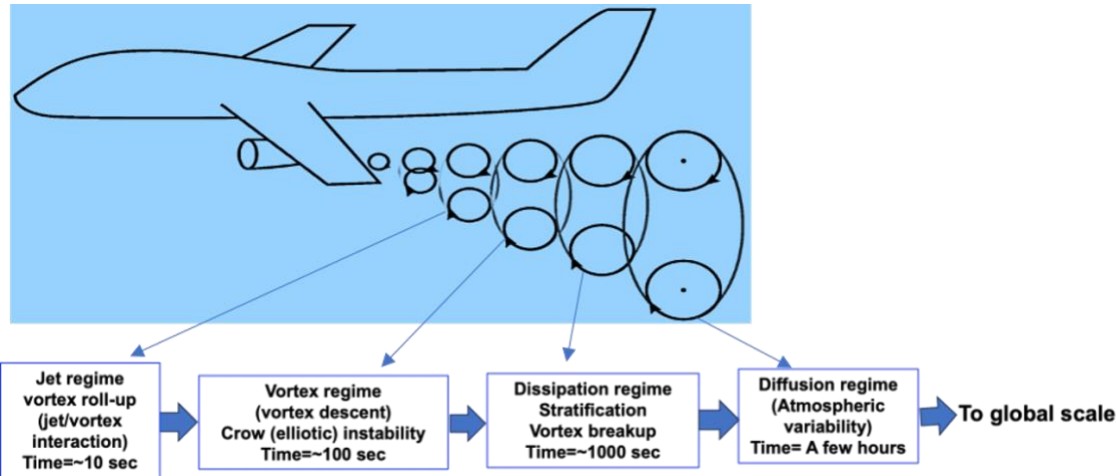

**Figure 5.** Schematic classification of aircraft wakes evolution into four regimes. Adapted from Paoli and Shariff (2016).

The descent of vortices effects in a contrast in density between the air contained within the vortices (within an oval-shaped region) and the ambient air. This process leads to the generation of vortices with opposite signs (in a stable stratified atmosphere) along the oval boundary. The vorticity shed in the upward direction gives birth to a "secondary wake." Within this secondary wake, a portion of the exhaust gases and ice particles becomes incorporated. Within the regime of vortex dissipation, the primary vortex pair and secondary vorticity disintegrate and disperse, releasing both exhaust and ice crystals that eventually endure sublimation. Ice crystals released in the secondary wake can endure for a longer duration due to the lower temperature. In the fourth regime, known as the diffusion regime, the horizontal and vertical spreading of the contrail is influenced by atmospheric turbulence, particle sedimentation, radiative processes, and wind shear until complete mixing takes place, typically within a few hours.

Figure 6 illustrates the processes that influence the various stages of contrail formation. Exhaust plumes, generated by burning fuel-air mixtures at high temperatures and pressure within turbofan jet engines, contain both gaseous and particulate matter. In the freely expanding and cooling plumes (jet regime), particle types include emitted soot particles and aqueous aerosol particles formed within the exhaust, along with entrained ambient aerosol particles. In scenarios where turbulent mixing occurs, leading to cooling and generating plume supersaturation over liquid water, a significant number of plume particles transform water droplets (depicted as gray circles in Fig. 6).

Water droplets rapidly freeze and increase in size by absorbing water vapor from their surroundings, eventually leading to the formation of a visible contrail. This occurs when the ambient temperature drops below the formation threshold. In exhaust rich in soot, most droplets contain soot inclusions. While droxtals, frozen water droplets with faceted surfaces, and hexagonal prisms and columns likely emerging from them may offer a realistic depiction of small ice crystals in fresh contrails, there is currently no direct observational evidence for their specific shapes.

Plumes from multiple aircraft engines combine with the two wingtip vortices, creating an inhomogeneous wake. The subsequent evolution of ice crystals is contingent upon fluid-dynamical processes, particularly in the vortex regime. The downward movement of the vortex pair warms

the surrounding air, resulting in the sublimation of ice crystals in the lower portion of the wake. A
small fraction of contrail is amplified by the detrainment of air from the lower wake. Ice crystals
present in the upper wake continue to grow by the uptake of entrained ice-supersaturated ambient
water vapor. A few minutes past emission, the organized flow pattern collapses and mixes with
ambient air (dissipation regime). Table 1 summarizes quantitative comparisons of contrail
characteristics across different stages of evolution, including measurement techniques and
modeling considerations.

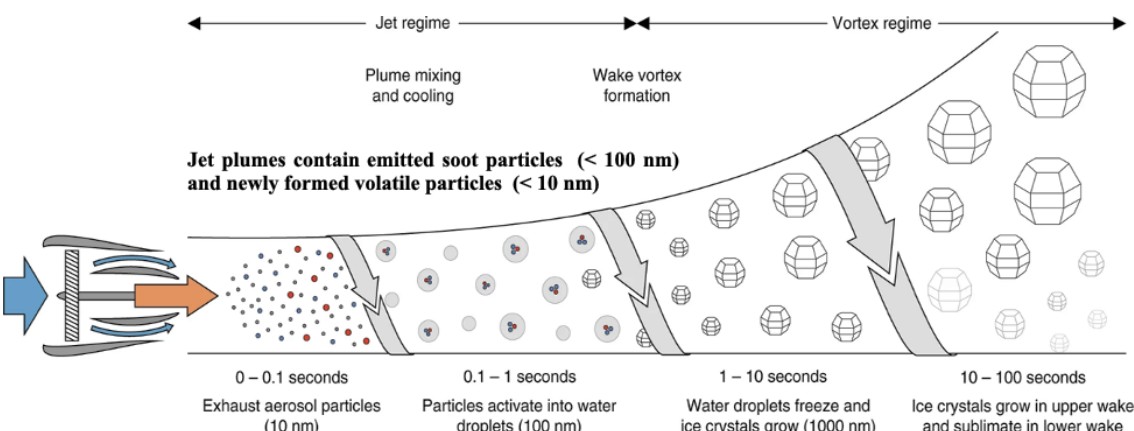

**Figure 6.** Processes influencing the contrail formation stage. Figure adopted with permission from Kärcher (2018). (Licensed under CC BY 4.0)

**2.4 Contrail coverage**
The radiative forcing (RF) for contrails, as determined by Lee et al. (2010), is defined by the
product of contrail coverage and optical depth. For a single contrail segment, this product equates
to total extinction. However, defining contrail coverage presents challenges for several reasons.
Assessing contrail coverage separately from optical depth introduces difficulties. The
identification of young contrails versus other cirrus clouds often relies on their distinctive line
shape. Yet, using geometric characteristics for contrail coverage classification introduces
uncertainty, particularly when contrails undergo deformation or shape changes (Mannstein et al.,
1999). Only a small fraction of all linear contrails is detectable from satellites (Kärcher et al., 2009;
Mannstein et al., 2010; Minnis et al., 2013).
Earlier studies examined global contrail coverage using regional satellite observations, estimating
potential contrail coverage based on specific temperature and humidity data and traffic density
(Sausen et al., 1998). The integration of observations to achieve hemispheric coverage has only
recently become available (Duda et al., 2013). The coverage by contrail cirrus may be significantly
higher than determined by line-shaped contrails, and uncertainty factors around an order of 10
have been reported (Burkhardt and Kärcher, 2011). However, this ratio is highly uncertain due to
detection issues. Burkhardt and Kärcher (2011) conducted a quantification of contrail cirrus using
the ECHAM contrail cirrus global climate model (CCmod), which incorporates a subgrid-scale
cloud class representing young contrails. This model captures the life cycle of artificial clouds and
simulates their global coverage, along with the changes in natural cloudiness they induce. Their

calculations revealed contrail cirrus coverage to be approximately 0.23%. Similar results were obtained from recent attempts to quantify contrail cirrus using MODIS data, accounting for more diffuse contrail contributions (Minnis et al., 2013).

**Table 1.** Estimated changes in contrail properties during different stages with measurement techniques and modeling consideration.
(The presented values for different stages of contrail properties and measurement techniques/modeling considerations are not based on specific references but represent a general understanding from (Schumann et al.,1998; Schumann, 2005; Schumann et al. 2013; Paoli and Shariff, 2016; Schumann et al. 2017; Heymsfield et al., 2017; Bräuer et al., 2021a, Bräuer et al., 2021b; Schumann et al., 2021; Li et al., 2023; Wang et al., 2023; Geraedts et al., 2024 )

| Stage | Ice Crystal Size ($\mu$m) | Number Density ($cm^{-3}$) | Extinction Coefficient ($km^{-1}$) | Width (m) | Length (km) | Optical Depth |
|---|---|---|---|---|---|---|
| Jet | 1-10 | $10^2 \cdot 10^4$ | $10^{-2} - 10^{-1}$ | 10 - 20 | 10 - 100 | 0.01 - 0.1 |
| Vortex | 10 - 30 | $10 - 10^2$ | $10^{-3} - 10^{-2}$ | $10^2 - 10^3$ | 1 - 10 | 0.001 - 0.01 |
| Dissipation | 30-100 | 1 - 10 | $10^{-4} - 10^{-3}$ | $10^3 - 10^4$ | 0.1 - 1 | 0.0001 - 0.001 |
| Diffusion | 1000-3000 | 0.1 - 1 | $10^{-5} - 10^{-4}$ | $10^4 - 10^5$ | 1 - 10 | 0.00001 - 0.0001 |

| Stage | Measurement Techniques | Modeling Considerations |
|---|---|---|
| Jet | Lidar, satellite imagery | Microphysics, Ice Nucleation rates |
| Vortex | Lidar, In-situ probes | Turbulence, Wind shear effects |
| Dissipation | Lidar, Ground-based observations | Sedimentation, evaporation rates |
| Diffusion | Lidar, satellite imagery | Radiative transfer, ice crystal habit |

Observations during the COVID-19 pandemic lockdowns provide valuable insights into the relationship between air traffic and contrail coverage. Dauda et al. (2023) documented a significant decrease in contrail coverage (up to 41%) over the conterminous United States during the lockdown period compared to pre-pandemic levels. This decrease highlights the direct impact of air traffic on contrail formation and underscores the potential for mitigating climate impact through air traffic management strategies. However, the study also emphasizes the importance of considering environmental factors alongside air traffic when assessing contrail coverage. While the reduction in air traffic was the primary driver of the observed decrease in contrail formation, changes in atmospheric conditions at cruise altitudes likely played a secondary, but significant, role.

In a modeling study, Huszar et al. (2013) adjusted the concentration of a tracer for contrails by multiplying it with a specific factor and then adding it to the large-scale cloud ice mixing ratio. They fine-tuned this factor through annual simulations spanning 2005 to achieve an appropriate global value of the top-of-the-atmosphere radiative forcing attributable to contrails and contrail-induced cirrus for that year, aiming for a target value of 31 mWm$^{-2}$ as reported by Burkhardt and Kärcher (2011). The distribution of this forcing aligns with Burkhardt and Kärcher (2011) findings, showing peak values exceeding 300 mWm$^{-2}$ over central Europe and the eastern U.S., with significant parts of Europe and the U.S. experiencing a contrail-induced cirrus radiative forcing above 100 mWm$^{-2}$. However, their estimates for eastern Asia surpass those of Burkhardt and Kärcher (2011), with maximum values exceeding 300 mWm$^{-2}$, while ranging between 100 and 300 mWm$^{-2}$ elsewhere. They suggested these differences could be attributed to how contrail cirrus is handled (a simpler parameterization in this study versus the more complex contrail-cirrus

module used by Burkhardt and Kärcher), variations in simulated thermodynamic conditions over
East Asia, and differences in emission inventories.
Schumann (2012) developed the Contrail Cirrus Prediction (CoCiP) tool based on a Lagrangian
approach. In this model, global contrail coverage is computed by combining the optical depth ($\tau$)
contributions from individual contrails and ambient cirrus, counting fractions of areas where
contrails cause the optical depth of the total cirrus to surpass a specific threshold (Schumann 2012).
Contrails not only augment cloud coverage but also thicken existing cirrus, consequently
influencing coverage (Minnis et al., 2013; Schumann and Graf, 2013). This thickening occurs by
generating more ice particles with smaller effective radii at constant ice water content (Kristensson
et al., 2000). While it is acknowledged that contrails consume humidity and may reduce natural
cirrus (Burkhardt and Kärcher, 2011; Unterstrasser and Görsch, 2014; Schumann et al., 2015;
Bickel et al., 2020), the prevailing evidence suggests that the thickening effect caused by numerous
additional small ice crystals tends to dominate.
**2.5 Contrail ice crystal nucleation**
Kärcher and Yu (2009) systematically investigated the relationship between nucleated contrail ice
crystal numbers and soot particle emissions. In exhaust rich in soot, there is a nearly proportional
increase in both ice crystal and soot particle numbers. Close to the contrail threshold, the formation
of ice crystals is limited as the low plume supersaturation hinders the activation of water on soot
particles, resulting in only a few ice crystals forming. Under conditions rich in soot, the quantity
of ice crystals reduces by about one hundred times compared to soot-poor conditions, reaching
constant low levels regulated by the number of particles present in the contrail environment. In
soot-poor exhaust at low ambient temperatures, there is an increase in ice crystal numbers due to
the activation of water and subsequent freezing of abundant aqueous plume particles. Figure 7
illustrates the ice crystal number emission index (per kilogram of fuel burnt) in the jet regime,
correlating with the number emission index of emitted soot particles as simulated by a parcel model
(Kärcher and Yu, 2009). Two sets of results are presented, one for an ambient temperature (T)
close to a contrail formation threshold temperature, $\Theta \approx 225$ K (-48°C), commonly encountered in
extratropical cruise conditions, and another for a lower temperature of $\Theta$ -12 K $\approx 213$ K (-60°C).
At intermediate ambient temperatures, nucleated ice crystal numbers increase due to enhanced
water activation of either soot or ultrafine aqueous plume particles.
The hatched area in Figure 7 represents the approximate range of current in-flight soot emission
indices. In exhaust rich in soot, the number of soot particles capable of water activation and
freezing increases with decreasing ambient temperature. This, in turn, raises plume cooling rates
and levels of plume supersaturation over liquid water. It's important to note that the primary driver
of the increased cooling and supersaturation is the decreasing ambient temperature, with soot
particles acting as additional ice nuclei under these cooler conditions. As the ambient temperature
(T) approaches the contrail formation threshold ($\Theta$) in soot-poor exhaust, ambient aerosol particles
mixed into exhaust plumes become the sole source of contrail ice crystals. This is because fewer
plume particles can be activated due to declining plume supersaturation. The number
concentrations of ambient particles in contrails are significantly lower than current levels of soot
emissions.
The plume cooling rate plays a crucial role in determining the timing of water activation of
entrained ambient particles and, consequently, the number of ice crystals derived from them
(Kärcher 2015). In soot-poor exhaust, a substantial number of ultrafine aqueous plume particles
are formed in the fresh exhaust, especially if the fuel contains condensable vapors besides water
vapor. These small particles are anticipated to contribute significantly to ice crystal formation at
low ambient temperatures well below average values at cruise levels ($T \approx 218$ K in the extratropic
and $T \approx 228$ K in the tropics). If the formation of ultrafine particles cannot be mitigated, ice crystal
numbers are expected to be lowest in an intermediate range of soot emissions, specifically between
$10^{13}$ and $10^{14}$ (kg-fuel)$^{-1}$.
Ice crystal number (Ni), which quantifies the concentration of ice crystals in the surrounding
atmosphere, plays a significant role in contrail properties (Kärcher, 2015). Studies have shown that
Ni can influence factors such as contrail persistence, spreading, and optical characteristics (e.g.,
Lewellen, 2014). A higher Ni value typically indicates a greater concentration of ice particles
within the contrail, which can affect its behavior in several ways.
The signs of variation in the number of ice crystals refer to the alteration in the ice particle
concentration within a contrail, such as increased ice particle concentration in the contrail, greater
persistence, and spreading of the contrail. Increased optical effects, such as increased brightness
and reflectivity. The decreased ice crystal particle concentration in the contrail leads to reduced
persistence and spreading of the contrail, and weaker optical effects. These variations in Ni may
occur due to several factors, comprising changes in atmospheric conditions, aircraft engine
emissions, and the presence of ice nuclei or other particles that impact ice particle formation.
Lewellen (2014) elucidates the pivotal role of initial ice crystal concentration (Ni) in contrail
dynamics through extensive simulations and the proposition of a simplified model. Their
simulations elucidate a discernible correlation between elevated Ni values and several significant
effects on contrails. First, contrails characterized by higher Ni values demonstrate prolonged
lifetimes, persisting for considerably extended durations, with simulations even indicating
lifetimes surpassing 40 hours (Lewellen, 2014). This protracted residence within the atmosphere
translates to amplified influence on radiative forcing, as contrails with higher Ni values possess
increased temporal opportunities for interaction with incident solar radiation. Second, heightened
ice particle concentration within the contrail leads to augmented contrail spreading. As Ni
escalates, interactions with atmospheric conditions become more conspicuous, potentially
precipitating the formation of contrail cirrus, a prevalent cirrus cloud type intimately associated
with contrail genesis (Lewellen, 2014). Finally, higher Ni values correlate with intensified optical
effects. A greater abundance of ice crystals within the contrail facilitates more efficient scattering
of incident sunlight, engendering brighter and more reflective contrails that perturb the radiative
equilibrium in the atmosphere (Lewellen, 2014).
Lewellen (2014) delineates a valuable framework for understanding the influence of Ni on contrail
significance through their simplified model: $S_\Sigma = \alpha N \times D$, where $S_\Sigma$ represents the lifetime-
integrated ice crystal surface area, a metric for contrail impact, N is the total crystal number per
length of flight path (i.e., ice crystal concentration), and D represents the number-averaged fall
distance of ice crystals before sublimation. The coefficient 'α' is approximately constant in the
regime of interest. This equation underscores the critical role of Ni (N) in determining the overall
radiative forcing exerted by contrails. A higher Ni value translates to a larger $S_\Sigma$, indicating a more
significant impact on the atmosphere due to increased light scattering and longer contrail
persistence.
**2.6 Stages of aviation-induced cloud evolution and spreading stage**
The microphysical and optical characteristics of ice crystals within contrails undergo alterations
as they disperse or transform into contrail cirrus, contingent upon the prevailing meteorological
conditions and microphysical processes (Fig. 8). Over time, persistent contrails undertake a
transformation from their initial linear form to become contrail cirrus. In regions of heavy air
traffic, these contrail cirrus formations overlap and merge, giving rise to extended layers of ice
clouds characterized by alterations in shape, depth, and longevity. These patterns differ from
natural cirrus clouds in terms of microphysical, and optical, additionally, geometric properties also
play a role. The ice-supersaturated layers that sustain these properties exhibit variations in both
vertical structure and horizontal extent, impacting the exchange of water molecules between vapor
and ice phases within them. Collectively, these factors contribute to the radiative forcing (RF)
potential of aviation-induced cirrus (AIC). Figure 8 illustrates the augmentation in cloud coverage
area attributed to the vertical shear of horizontal wind components.
The turbulent mixing or entrainment processes result in a gradual reduction (dilution) of ice crystal
concentrations over time. The sizes of ice crystals grow through the absorption (deposition) of
water vapor from layers that are supersaturated with ice. Continuous deposition growth causes
deviations in the shapes (habits) of ice crystals from their initial isometric forms.
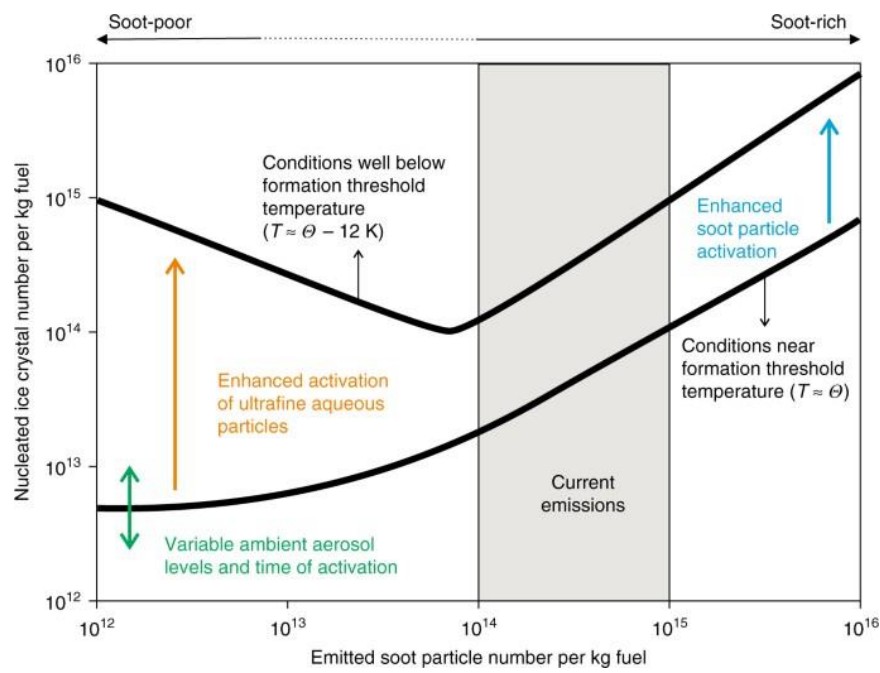
**Figure 7.** Nucleation of ice crystals in jet aircraft exhaust plumes. Figure adapted with permission
from Kärcher (2018). (Licensed under CC BY 4.0)
The shapes and sizes of ice crystals in cirrus clouds significantly impact their growth rates, fall
speeds, and optical properties. Larger ice crystals (>30 micrometers) tend to settle due to gravity
with fall speeds exceeding 100 meters per hour and are more likely to sublimate in warmer or drier
environments. Smaller ice crystals remain suspended around the flight levels due to their negligible
fall speeds, provided that some degree of supersaturation can be maintained, which depends on the
prevailing meteorological conditions.
The efficiency of sedimentation, the process by which ice crystals fall from clouds, is influenced
by the depositional growth rate, which in turn is affected by ice supersaturation, the rate of air
cooling, and the characteristics of the ice crystals themselves, including their size, shape, and
number concentration. Sedimentation increases the vertical extent of cirrus clouds, enhancing their
spreading rate and coverage under sheared flow conditions.
The settling of ice crystals, known as sedimentation, is a critical factor in determining the lifespan
of contrails. As the contrail ages, larger ice particles, due to their increased mass, fall out of the
plume more rapidly. This phenomenon can be explained by the concept of terminal velocity
(Spichtinger and Gierens, 2009). Researchers have explored methods to calculate this velocity,
considering the distribution of ice crystal sizes within the plume (Spichtinger and Gierens, 2009).
This approach provides a more comprehensive understanding of the sedimentation process.
The size of an individual ice crystal directly influences its falling speed. Larger crystals, typically
associated with persistent contrails, descend from the plume quicker than their smaller counterparts
found in ephemeral contrails. This sedimentation process significantly impacts the vertical
distribution of ice crystals within the contrail and affects its overall persistence. Ice crystals lofted
higher, potentially due to weaker updrafts, may experience slower sedimentation and persist for
longer durations, especially in colder atmospheric regions.

**2.7 Lessons learned from observations of contrails and their properties**
Contrails, as well as cirrus clouds, are frequently observed in air that lacks saturation (Kübbeler et
al., 2011; Krämer et al., 2009). The prevailing explanation suggests that these clouds consist of
large ice particles formed under supersaturated conditions, which then descend to lower and drier
levels, undergoing slow sublimation. Notably, instances of contrails and contrail cirrus have been
documented in ice-sub-saturated air across various scenarios, extending beyond dedicated contrail
research flights (Schumann et al., 2017; Voigt et al., 2011; Kübbeler et al., 2011; Gayet et al.,
2012; Chauvigné et al., 2018). Such observations have been derived not only from contrail-specific
research endeavors but also from In-service Aircraft for a Global Observing System (IAGOS)
commercial aircraft data collected in the North Atlantic region (Petzold et al., 2017).
Apart from a high number of small contrail ice particles, large particles (ice particle diameter
greater than 100 µm) were also detected but at relatively low concentrations (Voigt et al., 2010).
Such large ice crystals were also observed in contrail cirrus during the ML-CIRRUS (Midlatitude
Cirrus experiment) campaign (Voigt et al., 2017). However, attention to contrail cirrus in ice-sub-
saturated environments and the role that large ice particles play in contrail cirrus was raised by
Kübbeler et al. (2011) and Schumann (2012).

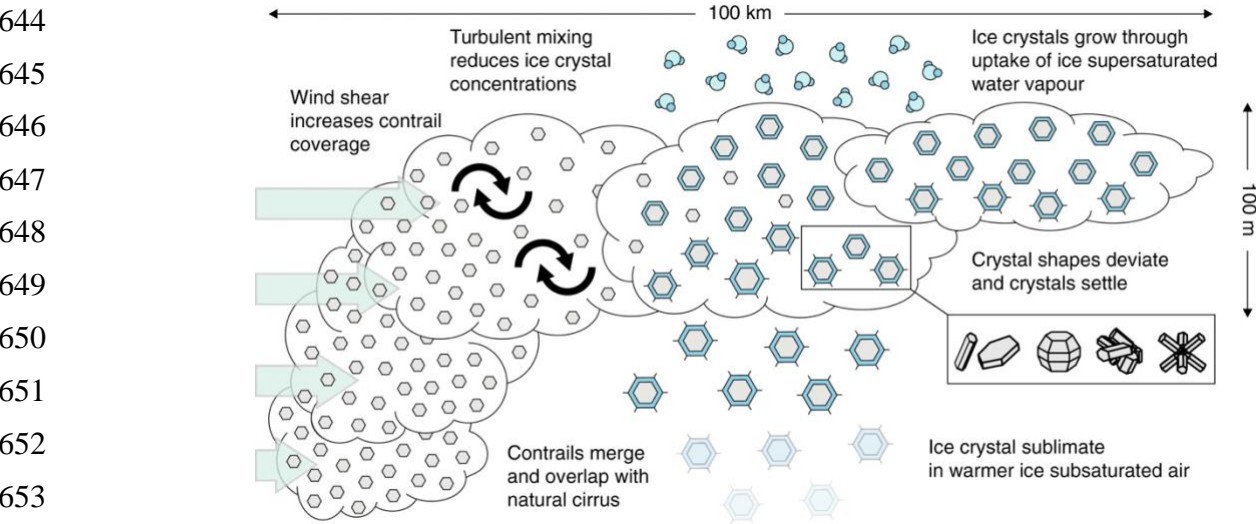

**Figure 8.** Influential factors in aircraft-induced cloud formation. Figure adopted with permission from Kärcher (2018). (Licensed under CC BY 4.0)

Kübbeler et al. (2011) suggested that the subsaturation observed in contrail cirrus during the CONCERT campaign is caused by the sublimation of large ice particles that may have fallen from higher altitudes after forming under ice supersaturated regions (ISSRs). However, due to the limited contrail cirrus data available, it is difficult to confirm whether contrail cirrus commonly occurs in ice-subsaturated environments.

Li et al. (2023) explored an extended dataset derived from 14.7 hours of cirrus cloud sampling conducted at a frequency of 1 Hz, with a maximum speed of approximately 290 m s$^{-1}$, obtained during the ML-CIRRUS 2014 campaign. Utilizing readily available parameters that characterize cirrus microphysical properties-including ice number concentration, ice crystal sizes, and ice water content (IWC)-they employed a more straightforward statistical method to distinguish aviation-induced cirrus from natural cirrus, in contrast to the approach adopted by Chauvigné et al. (2018). It consists of the SAC, covering the most common aircraft cruising altitude range, and a recently developed algorithm for detecting aircraft exhaust plumes to identify matured contrail cirrus (>0.5 h lifetime; Voigt et al., 2017; Schumann et al., 2017), and natural cirrus. Contrail cirrus showed sharp differences from natural cirrus during the stages of formation and in the corresponding microphysical properties and occurrence conditions.

Mahnke et al. (2024) investigated the impact of aviation-induced aerosols on cirrus clouds and climate, which remains uncertain. Understanding these properties is vital for assessing aviation's climate effects. From July 2018 to March 2020, the IAGOS-CARIBIC Flying Laboratory identified over 1100 aircraft plume encounters. Findings showed consistent aerosol properties across various altitudes (upper troposphere, tropopause region, and lowermost stratosphere). The exhaust aerosols were primarily externally mixed, even after 1 to 3 hours, with no increase in larger particles (diameter >250 nm) inside the plumes. The particle number emission indices (EIs) from aged plumes matched engine certification values, indicating that aviation aerosols maintain their emission state during plume expansion and that global models use accurate EI values.

**Observational Constraints**

While there is growing evidence regarding contrail observations in ice-subsaturated environments, it is imperative to acknowledge the inherent limitations associated with measuring contrail properties. These limitations stem from various factors, including:

*Spatial and temporal resolution of instrumentation:* The comprehensive depiction of contrail phenomena may be impeded by the restricted range or sampling frequency of instruments employed for in-situ or remote sensing observations. For instance, satellite-based observations might fail to capture minute-scale contrail features, while in-situ measurements conducted via research aircraft may not fully encompass the spatial extent of a contrail layer.

*Discriminating contrails from natural cirrus formations:* The differentiation between contrails and natural cirrus clouds, particularly in ice-subsaturated conditions, presents a significant challenge and often necessitates sophisticated analytical techniques. Both varieties of ice clouds can exhibit comparable characteristics concerning size and ice crystal properties, rendering the definitive identification of their origin based on limited observations a challenging endeavor.

*Restricted data availability:* Observational datasets, particularly those focusing on fully developed contrail cirrus, may be scarce. This scarcity impedes the ability to draw definitive conclusions regarding the prevalence of contrails in sub-saturated environments and their enduring influence on the climate system.

## 3 Modeling of Contrails and their Impacts

Understanding the multifaceted impacts of contrails necessitates employing models at various scales. Local-scale process and fluid-dynamic models offer valuable insights into how plume turbulence and wake dynamics influence the nucleation and properties of contrail ice crystals. These simulations can then be validated against in-situ aircraft measurements of nucleated ice crystal concentrations and sublimation losses (Kärcher, 2018). For broader-scale analysis, cloud-resolving and regional models are utilized. These models incorporate meteorological boundary conditions and leverage optical parameterizations to capture the shortwave radiative response of micron-sized ice crystals. By co-locating these simulations with data from aircraft and satellite observations, scientists can gain a more comprehensive understanding of the radiative forcing (RF) exerted by contrail cirrus and its potential influence on pre-existing cirrus clouds (Kärcher, 2018).

The limitations of current instrumentation in measuring the microphysical and optical properties of contrail cirrus elevate the importance of models (Table 2) for estimating their RF. To provide robust predictions of climate change driven by aircraft emissions, models simulating ice nucleation and sublimation must effectively translate the current understanding of contrail formation and evolution processes into frameworks that lack the spatial or temporal resolution to explicitly capture the formation stage. This objective can be achieved by synergistically integrating findings from observations with local-scale, process-oriented models.

### 3.1 Modeling of individual contrails

Individual contrails have been studied by various authors through approaches ranging from simplified parameterizations to more complex numerical models. Parameterization approaches normally depend on simplified assumptions to estimate contrail properties based on simplified atmospheric and emission conditions. On the contrary, more sophisticated numerical models, such as the Contrail Cirrus Prediction model (CoCiP), utilize detailed representations of atmospheric processes, ice particle growth, and radiative transfer to simulate the formation and evolution of contrails.

**Table 2.** Modeling the complexities of aircraft-induced cloud radiative effects (Adapted from Kärcher 2018).

| Model scale | Spatial resolution | | Contrail stages | Major challenges | Approach/solution |
|---|---|---|---|---|---|
| | Horizontal | Vertical | | | |
| **Local** | <10 m | <10 m | Formation stage | Ice crystal number and size distribution | Turbulence microphysics coupling |
| **Regional** | <1000 m | <100 m | Spreading stage | Radiative flux changes and interaction with natural clouds | Contrail to contrail cirrus transition |
| **Global** | <1000km | <1km | Full life cycle | Ice crystal formation and ice supersaturation | Parameterization and high resolution |

Improvements in a hierarchy of local- to global-scale models to be realized in conjunction with observations providing data for cloud and radiation parameterization development and overall model validation.

### 3.1.1 Contrail Cirrus Prediction Tool (CoCiP) for individual contrails

Schumann et al. (2017) compiled a dataset of contrail properties from various sources, including previous publications, additional information gathered from experimenters, reanalysis of existing data, and comparisons with the CoCiP database. The dataset expands upon the work of Schumann and Heymsfield (2017) by incorporating data from the ML-CIRRUS campaign (Voigt et al., 2016). The data includes both in-situ measurements of contrails, such as ice particle size spectra obtained using optical particle spectrometers (Baumgardner et al., 2011; Wendisch and Brenguier, 2013), and remote sensing data from ground-based and airborne lidar, spectroradiometers, satellites, cameras, and visual observations. Remote sensing data provides information on contrail properties such as width and optical depth (Spinhirne et al., 1998; Duda et al., 2004).

### 3.1.2 MIT Aircraft Plume Chemistry, Emissions, and Microphysics Model (APCEMM)

APCEMM is applied to assess the impact of non-linear plume chemistry and to deliver an initial estimate of the effects of contrails on atmospheric chemistry. Accurately determining the intricate relationship between contrail microphysics and chemistry often necessitates the use of costly large eddy simulations (LESs). APCEMM, through simplified assumptions about plume dynamics, strives to close the disparity between Gaussian plume models and LESs, as outlined by Fritz et al. (2020). APCEMM simulates the growth and chemical progression of an individual aircraft plume. It computes chemical concentrations and aerosol characteristics for a two-dimensional cross-section of the plume, angled perpendicular to the flight path. Dynamics, chemistry, and microphysics are explicitly modeled within the plume, using two different approaches depending on the age of the plume.

### 3.1.3 NASA's global model for evaluation of individual linear contrails

The NASA Ames Research Center developed a computationally efficient aircraft contrail model designed to simulate aircraft-induced contrail formation. This model relies on the Appleman criterion and operates under static atmospheric conditions (Sridhar et al., 2010; Neil et al., 2010). Subsequently, researchers from NASA Ames extended this model to simulate the dynamic transport of contrails by incorporating a Lagrangian dispersion model and a cloud microphysics model (Li et al., 2013). The computational methods employed are grounded in well-established

approaches utilized in other aircraft contrail models (Pruppacher and Klett, 2000; Schumann et al., 1995).

In comparison with models from Stanford (Naiman et al., 2009; Naiman et al., 2011; Jacobsen et al., 2011) and DLR (Burkhardt and Kärcher, 2009; Burkhardt and Kärcher, 2011; Bier and Burkhardt, 2022), this dynamic contrail model diverges primarily in two aspects: (1) It excludes the initial contrail ice particles down-wash process caused by airplane wake vortex turbulence. This process, typically lasting less than a minute, is crucial in determining contrail ice nucleation, initial contrail ice particle sizes, and displacements through a complex fluid dynamic process dependent on atmospheric, aircraft, and fuel parameters. In this model, the average initial ice particle size is predefined, and the initial contrail location is set based on the cell where contrail formation conditions are met. (2) The results from this model do not yet include additional radiative forcing caused by aircraft contrails. The researchers are in the process of adding a contrail radiative forcing module capable of computing the total aircraft contrail radiative forcing using inputs from the model, such as ice particle size and linear contrail cloud cover area.

### 3.1.4 DLR's ICON-LEM regional model

Verma and Burkhardt (2022) developed and implemented a model for contrail formation into the ICON-LEM (ICOsahedral Non-hydrostatic Large-Eddy Model; Zängl et al., 2014; Dipankar et al., 2015). This model includes parameterizations for ice nucleation in the jet phase and ice crystal loss during the contrail's vortex phase. It facilitates the investigation of modifications to cirrus clouds resulting from contrail formation. ICON is the new German Numerical Weather Prediction/Climate Model co-developed by DWD and MPI. It solves a set of equations on an unstructured triangular grid based on successive refinement of a spherical icosahedron (Zängl et al., 2014).

The model asserts high horizontal resolution coupled with a vertical resolution of approximately 150 m in the upper troposphere, enabling the resolution of pertinent cloud processes. Within ICON-LEM, a contrail scheme has been developed and implemented, incorporating the parameterization of contrail ice nucleation as proposed by Kärcher et al. (2015) and accounting for the survival of ice crystals within the vortex phase. (Unterstrasser, 2016), to study changes in cloud variables due to contrail formation within cirrus. Contrail formation, dependent on atmospheric as well as aircraft and fuel parameters, is calculated, and contrail ice nucleation and ice crystal loss in the contrail's vortex phase are estimated.

### 3.1.5 Large-eddy simulations covering the entire life cycle, from initiation to termination

Lewellen et al. (2014) outlined the utilization of large-eddy simulations with size-resolved microphysics to model persistent aircraft contrails and the resulting contrail-induced cirrus clouds. These simulations aim to depict the dynamic evolution of contrails, spanning from a few wing spans behind the aircraft to their dissipation over an extended period. The emphasis of the study lies in the modeling approach, discussing the development of schemes for efficient numerical computation. The authors introduced dynamic local ice binning and updating, along with coupled radiation, to accurately capture microphysical processes and radiative properties within individual columns. The paper also addresses the challenge of maintaining realistic ambient turbulence over extended simulation times, proposing a "quasi 3D" approach as a computationally feasible approximation of the full dynamics, allowing for exploration across a broader parameter space.

Lewellen (2014) provides an extensive analysis encompassing over 200 instances of long-lived contrails, spanning from their formation at several seconds old to their termination. The study investigated the complete lifespan of long-lived contrails originating from a single aircraft. Various factors were explored, including the effective ice crystal number emission index, temperature, relative humidity concerning ice, stratification, shear, supersaturated-layer depth, uplift/subsidence, and coupled radiation. The analysis delved into the scaling behaviors of contrail lifetime, width, ice mass, and surface area. The simulations unveiled contrail lifetimes exceeding 40 hours, widths surpassing 100 kilometers, and ice masses exceeding 50 kg per meter of the flight path. The paper identified distinct behavioral regimes influenced by radiative forcing and proposed a simplified model to predict these regimes.

Key insights from the simulations highlight the notable impact of ice crystal number loss resulting from competition among different crystal sizes, influencing both young contrails and aging contrail cirrus. The sensitivity of contrail properties to the initial number of ice crystals decreases over time, highlighting the importance of uncertainties in ice crystal deposition coefficients and the Kelvin effect. The influence of atmospheric turbulence on contrail properties and lifetime is also emphasized. Additionally, the paper discusses the effects of ice crystal shape, coupled radiation, precipitation dynamics, and the role of water from fuel consumption in reducing ice crystal loss in colder contrails.

Lewellen's simulations provide valuable insights into contrail behavior, shedding light on the significance of crystal number loss mechanisms, the interaction between shear and ice sedimentation, the depth of the supersaturated layer, and the potential impact of "cold" subvisible contrails. The findings from these simulations contribute to estimating the effects of intricate contrail scenarios, formulating mitigation strategies, and enhancing our comprehension of the dynamics of natural cirrus clouds.

## 3.2 Treatment of contrails in global models

The consideration of contrail cirrus in global climate models has greatly improved in recent years. Despite these advancements, notable uncertainties persist, particularly in the depiction of contrail microphysics and the interaction between contrail-cirrus and cirrus-radiation. As mentioned earlier, the characteristics of young contrail cirrus diverge from those of natural cirrus primarily due to the elevated ice crystal number concentration typical in contrails. Consequently, microphysical process rates in contrail cirrus, influencing its lifespan, can exhibit significant deviations from those observed in natural cirrus. In this section, we examine the treatments of contrail and contrail cirrus in global models. Very few models account for contrails, so our focus is on those models that have been published extensively on contrail impacts, including modeling studies done by the Institute for Atmospheric Physics (DLR), the National Center for Atmospheric Research (NCAR), and Stanford University. The focus is on their modeling approaches and findings. We start with a description of the approach used to treat contrails in these models.

### 3.2.1 DLR Contrail Cirrus Prediction Model (CoCiP)

CoCiP can be implemented within a GCM and used to study the global distributions of contrails and contrail cirrus (Schumann et al., 2015). The model is specifically crafted for the estimation of contrail cirrus coverage and the analysis of contrail climate impact, particularly in the context of aviation system optimization processes. It is engineered to simulate the entire life cycle of contrails. Contrail segments arise between waypoints along individual aircraft tracks in air masses that are cold and humid enough. The initial characteristics of contrails are contingent upon the

specific aircraft involved. The advection and progression of contrails adhere to a Lagrangian
Gaussian plume model. This model treats the contrail life cycle using bulk contrail ice physics,
incorporating several simplifying assumptions. Notably, the model demonstrates efficiency in
handling mixing and cloud processes in a quasi-analytical manner (Schumann, 2012). Contrails
become extinct when the bulk ice content endures sublimation or precipitation.
The model takes into consideration the impact of both aircraft properties and ambient
meteorological conditions. This encompasses established contrail formation thresholds, the effects
of advection, turbulent mixing, and the formation of ice mass from both emitted and ambient
humidity. The number of ice crystals is contingent upon the quantity of soot particles emitted. The
model incorporates simplified approximations for the survival of ice particles in adiabatically
sinking wake vortices and the loss of particles in aged contrails (Schumann, 2012).
CoCiP (Contrail Cirrus Prediction) depicted in Figure 9 simulates the formation of contrails under
specific meteorological conditions, either regionally or globally. Numerical weather prediction
data are utilized to determine ambient meteorological conditions through linear interpolation at
given positions and times. The model has been effectively employed to simulate contrails in both
global and specific cases, and its results have been compared with outcomes from other models
and in-situ measurements (Schumann, 2012).
### 3.2.2 DLR version of the ECHAM5-HAM model
The global climate model ECHAM5-HAM (European Center for Medium-Range Weather
Forecasts (ECMWF) and Hamburg) was developed at the Max Planck Institute (MPI) for
Meteorology in Hamburg, Germany (Roeckner et al., 2003, 2006; Stier et al., 2005), and adapted
to study of aviation effects on climate, including the modeling of the effects from contrails. The
aerosol-climate modeling system ECHAM5-HAM is based on a flexible microphysical approach,
and it predicts the evolution of an ensemble of microphysically interacting internally and externally
mixed aerosol populations as well as their size distribution and composition. Bier and Burkhardt
(2022) applied the ECHAM5-HAM model, incorporating a two-moment microphysical scheme
that was expanded to introduce a novel cloud category-contrail cirrus. The contrail cirrus scheme
encompasses a parameterization addressing contrail ice nucleation, the loss of ice crystals during
the vortex phase, plume dilution, the spreading of contrails influenced by vertical wind shear, and
inclusive microphysical and macrophysical processes aligned with the natural cloud scheme.
The contrail cirrus parameterization implemented in ECHAM5-HAM follows the framework
established by Burkhardt and Kärcher (2009), who introduced contrail cirrus as a distinct cloud
class alongside natural cirrus clouds within the model's natural cloud scheme. The water and heat
budgets are balanced, with natural cirrus and contrails competing for available water vapor.
Prognostic variables, including ice water content, contrail coverage, and the length of contrail
cirrus, are computed based on factors such as persistence, advection, spreading, and
deposition/sublimation.

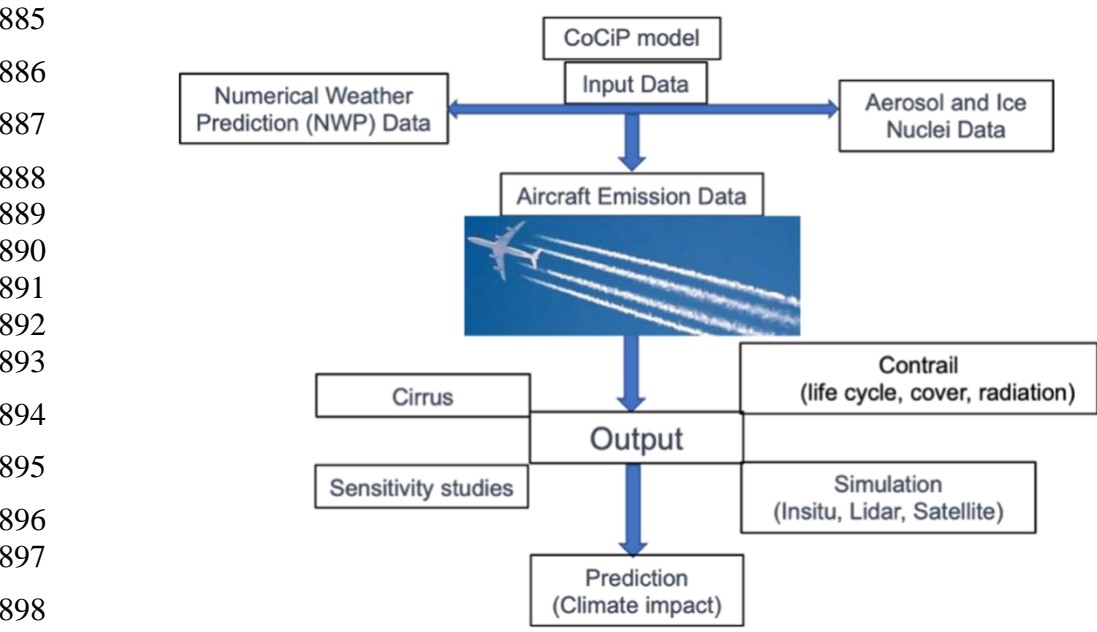

**Figure 9.** Schematic of the CoCiP model. Adapted from Schumann (2012). Contrail Image (Credit: Adrian Pingstone (public domain)

Bock and Burkhardt (2016) extended this parameterization (CCMod) by incorporating a microphysical two-moment scheme (Lohmann et al., 2008). This enhancement significantly improved the representation of microphysical processes by introducing contrail cirrus ice crystal number concentration and volume as additional prognostic variables. This advancement is crucial for studies investigating the impact of aircraft particle number emissions on contrail cirrus properties and their overall climate influence.

Bier and Burkhardt (2022) describe the initialization of contrails within the model. However, the chosen initialization time of 450 seconds (7.5 minutes) warrants further discussion. This value represents half of a model time step, potentially suggesting computational efficiency as a factor in its selection. Nevertheless, a 450-second delay in contrail formation might be too long for accurate simulations, especially for studies focusing on the early stages of contrail evolution. Future investigations need to explore the sensitivity of model results to this initialization time and potentially refine it for a more realistic representation of contrail formation processes.

In instances of contrail cirrus volumes exhibiting very low ice crystal number concentrations, as observed in aged contrails associated with heightened ice crystal sedimentation, the deposition of ice water is constrained, aligning with the depositional growth of ice crystals (Bock and Burkhardt, 2016). In more recent iterations of the model, specifically, ECHAM5-CCMod, Bier and Burkhardt (2019) incorporated the contrail ice nucleation parameterization proposed by Kärcher et al. (2015). In their model approach, they suggest that the plume cools over time due to continuous mixing of the exhaust with ambient air, eventually becoming water-supersaturated when the SAC criterion is met. The number of activated aerosol particles, stemming from soot and ambient particles, is computed based on ambient conditions, fuel/engine characteristics, and exhaust particle properties. Additionally, Bier and Burkhardt (2022) introduced a parameterization for ice crystal loss during the vortex phase and conducted a thorough evaluation of their model.

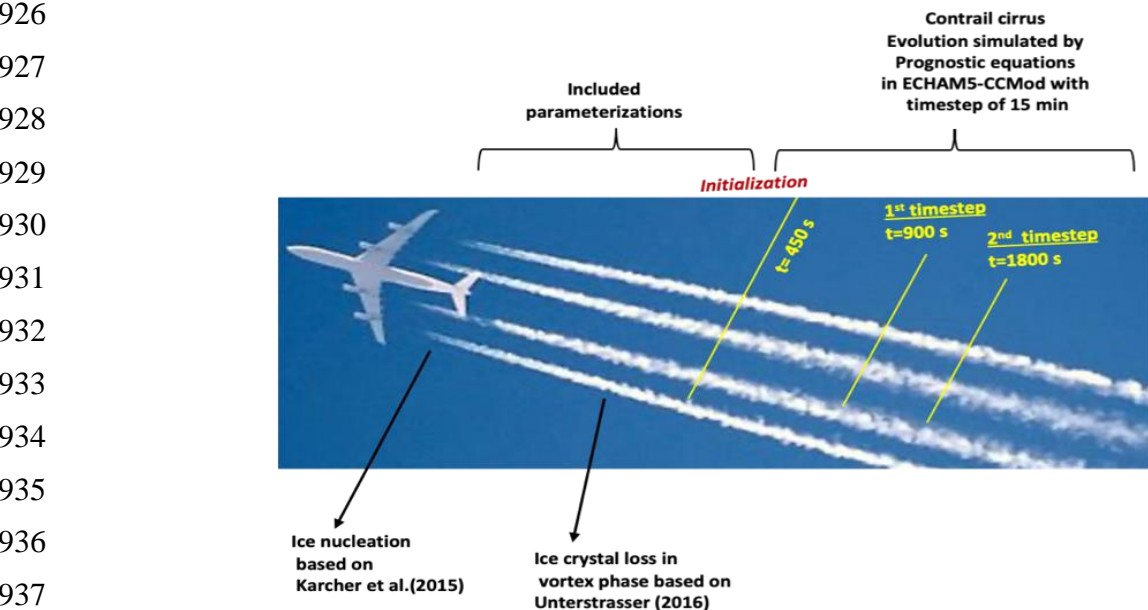

**Figure 10.** Stages of contrails and their corresponding treatment into the global climate model ECHAM5-CCMod. [Adapted from Bier and Burkhardt (2022). Contrail Image (Credit: Adrian Pingstone (public domain)

### 3.2.3 NCAR CAM6 model with Contrails

In the first NCAR study of the radiative forcing of linear contrails and contrail cirrus, Chen and Gettelman (2013) used the Community Atmosphere Model version 5 (CAM5), the atmospheric component of the National Center for Atmospheric Research (NCAR) Community Earth System Model (CESM). More recently, they have used the updated atmospheric component from the new version of CESM2 (Danabasoglu et al., 2020). The current atmospheric model implemented in CESM2 is CAM, version 6.2 (CAM6; Gettelman et al., 2020). CAM6 incorporates a detailed two-moment cloud microphysics scheme (Gettelman and Morrison, 2015) coupled with an aerosol microphysics and chemistry model (Liu et al., 2016; Gettelman et al., 2019). The latest version of the contrail parameterization (Chen et al., 2012) was utilized with CAM6 (Gettelman et al., 2021). While the ice cloud microphysics and aerosols exhibit minimal differences between CAM5 and CAM6, the aerosol activation in CAM6 significantly differs, impacting natural cirrus clouds but not contrails. The assumed emission ice particle diameter was adjusted from the original parameterization (10 μm) to 7.5 μm to better align with observations (e.g., Lee et al., 2021).

For contrail studies at NCAR, the standard version of CESM with 32 levels (to 3 hPa) vertical and ∼ 1∘ horizontal resolution was employed. Winds and optional temperatures were relaxed to NASA's data assimilation analyses, specifically the Modern-Era Retrospective analysis for Research and Applications, version 2 (MERRA2; et al., 2015), with wind nudging (Gettelman et al., 2020, 2021). CESM2 features a fully interactive land surface model (the Community Land Model, version 5; et al., 2020). Sea surface temperatures (SSTs) are fixed to MERRA2 SST, and there is no interactive ocean.

These simulations allow for adjustments in atmospheric and surface temperatures, resulting in radiative flux perturbations representing an Effective Radiative Forcing (ERF). Sensitivity tests were conducted, with temperatures nudged to MERRA2. The results were compared with previous

contrail simulations using this model and others, as well as observational data. The pattern of
contrail-induced changes to cloud fraction closely resembled the previous model documented in
CAM5 (Chen and Gettelman 2013), with peak effects observed in the Northern Hemisphere at
mid-latitudes. The radiative forcing in the CAM6 simulations exceeded that of the earlier Chen
and Gettelman (2013) study due to a smaller initial contrail area (100 vs. 300 m) and smaller initial
ice crystal sizes (7.5 vs. 10 µm diameter). The radiative forcing pattern and magnitude were
qualitatively and quantitatively consistent with the analysis conducted by Lee et al. (2021),
aligning with the intercomparison between contrail simulation models.

### 3.2.4 Stanford global model for contrail evaluation

Although it is no longer actively utilized in aviation studies, Stanford University developed a low-
order contrail model and a Large Eddy Simulation (LES) model (Naiman et al., 2009; Naiman et
al., 2011; Jacobsen et al., 2011). In their work, Jacobsen et al. (2011) assessed mass-conservative,
positive-definite, unconditionally stable, and non-iterative numerical techniques for simulating the
evolution of discrete, size, and composition-resolved aerosol and contrail particles within
individual aircraft exhaust plumes. This simulation was conducted in a global or regional 3-D
atmospheric model, incorporating the coupling of subgrid exhaust plume information to the grid
scale (see Fig. 11). This approach represents a distinct method for simulating the impacts of aircraft
on climate, contrails, and atmospheric composition.
The microphysical processes addressed within each plume include size-resolved coagulation
among and between aerosol and contrail particles, aerosol-to-hydrometeor particle ice and liquid
nucleation, deposition/sublimation, and condensation/evaporation. Each plume is characterized by
its own emission and supersaturation, and the spreading and shearing of each plume's cross-section
are calculated over time. Aerosol and contrail-particle core compositions are tracked for each size
and affect optical properties within each plume. When linear contrails sublimate or evaporate, their
size and composition-resolved aerosol cores and water vapor are introduced to the grid scale,
where they influence large-scale clouds. The model's algorithmic properties were analyzed, and
the final model was evaluated against in situ and satellite data. Table 3 summarizes the cross-
model intercomparison of contrail simulation models covered in this study.

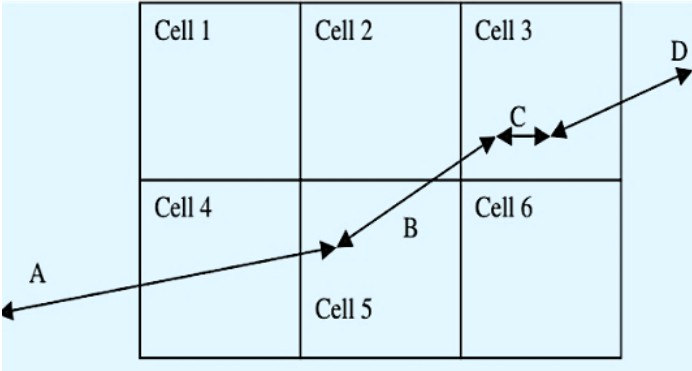

**Figure 11.** Example of how flight segments cross grid cell boundaries in the Jacobsen et al., model.
Segments A, B, C, and D are original segments. These are partitioned or aggregated into individual
model grid cells 2, 3, 4, and 5 to form ''new'' segments. Adapted with permission from Jacobson
et al. (2011).

## 4 Radiative Forcing for Contrail Cirrus in Global Models

### 4.1 Contrail radiative forcing and efficacy

Contrails during the day can warm or cool depending on optical depth, zenith angle, and ice crystal shape (Meerkötter et al., 1999; Stuber et al., 2006; Newinger and Burkhardt, 2012). Stuber et al. (2006) suggested that moving all air traffic to the day decreases RF. Newinger and Burkhardt (2012) stated that we need to consider the lifetime of the contrails since daytime air traffic can cause a large contrail coverage at night. After all, it is the long-lived contrail cirrus outbreaks that are responsible for a large part of the climate impact (Burkhardt et al., 2018). As illustrated in Fig. 1, the overall effect reveals a positive net radiative forcing (RF), signifying a warming impact. The examination of this phenomenon is undertaken here in contrast to the comprehensive RF, the local RF (RF′) can be defined as the instantaneous change in net incoming radiation for 100% contrail coverage in a specific location.

Contrail RF comprises both a long-wave (LW) and a short-wave (SW) component, with each influenced by flight characteristics and the time of day. Unlike the systematic dependency on optical depth ($\tau$), the relationship between RF and ice water path (IWP) is less consistent (De Leon et al., 2012; Schumann et al., 2012). The LW RF is positive both day and night, with the largest impact observed for a cold contrail (located near the tropopause) over a warm, cloud-free Earth surface. The shortwave radiative forcing (RF) is predominantly negative, with the greatest impact observed for contrails over darker surfaces, such as cloud-free oceans (Meerkötter et al., 1999). Studies conducted by Schumann et al. (2012), utilizing both their model and observational evaluations, suggest the existence of substantial regional RF′ values. The calculation of contrail RF within a global model is contingent on factors such as the representation of contrails, the atmospheric conditions, and the radiation transfer model employed (Myhre et al., 2009).

While radiative forcing (RF) provides a valuable measure of the impact of contrails on the radiative balance, it doesn't fully account for the efficacy of different forcing agents in driving surface temperature changes. Recent studies by Bickel et al. (2020) and Ponater et al. (2021) highlight the importance of effective radiative forcing (ERF) as a superior metric. ERF considers feedback mechanisms that can influence the ultimate climate impact of a forcing agent.

In instances of contrail cirrus volumes exhibiting very low ice crystal number concentrations, as observed in aged contrails associated with heightened ice crystal sedimentation, the deposition of ice water is constrained, aligning with the depositional growth of ice crystals (Bock and Burkhardt, 2016). In more recent iterations of the model, specifically, ECHAM5-CCMod, Bier and Burkhardt (2019) incorporated the contrail ice nucleation parameterization proposed by Kärcher et al. (2015). In their model approach, they suggest that the plume cools over time due to continuous mixing of the exhaust with ambient air, eventually becoming water-supersaturated when the SAC criterion is met. The number of activated aerosol particles, stemming from soot and ambient particles, is computed based on ambient conditions, fuel/engine characteristics, and exhaust particle properties. Additionally, Bier and Burkhardt (2022) introduced a parameterization for ice crystal loss during the vortex phase and conducted a thorough evaluation of their model.

Bickel et al. (2020) analyzed the feedback processes and quantified the effective radiative forcing (ERF) of contrail cirrus, which refers to the cloud formations produced by aircraft engine exhaust. They highlighted that feedback analysis is a valuable tool for understanding climate sensitivity and the differences in efficacies between various climate-forcing agents. Previous climate model simulations have suggested a relatively low efficacy of contrails in forcing global mean surface

temperature changes. They employed a climate model that incorporates a state-of-the-art
representation of contrail cirrus and conducted the simulations with fixed sea surface temperatures
to determine the ERF resulting from contrail cirrus.
**Table 3** Cross-model intercomparison of contrail simulation models

| Model Name | Developer | Simulations | Strengths | Weaknesses | Evaluation |
|---|---|---|---|---|---|
| CoCiP | DLR | Full life cycle | Efficient computation, considers aircraft/fuel | Simplified mixing, neglects radiation | In-situ measurements, remote sensing data |
| APCEMM | MIT | Individual microphysics/chemistry | Detailed microphysics | High computational cost, limited global use | Primarily theoretical, limited observations |
| NASA Global Contrail Model | NASA Ames | Individual formation/transport | Computationally efficient | Neglects downwash, excludes radiative forcing | Reanalysis data |
| ICON-LEM (Contrail Scheme) | DLR | Contrail formation (regional model) | Considers ice nucleation/survival | Relies on parameterizations, limited microphysics | Satellite data |
| LES Model | Stanford University | Full life cycle (individual & cirrus) | High-resolution microphysics | Computationally expensive, limited scalability | In-situ measurements |

| Model Name | Validation Method | | Model Uncertainties | | |
|---|---|---|---|---|---|
| CoCiP | In-situ measurements, remote sensing data | | *In-situ measurements*: May not fully capture contrail variability; calibration and processing uncertainties affect accuracy. *Satellite retrievals*: Uncertainties in thin contrails; cloud backgrounds can misidentify or underestimate contrails. *Data Coverage:* CoCiP dataset may not cover all contrail properties, limiting model generalizability. | | |
| APCEMM | Primarily theoretical, limited observations | | *Validation data:* Simplified plume dynamics may not accurately reflect real-world contrail behavior due to limited validation. *Parameterization:* Uncertainties in microphysical process parameterizations can affect model accuracy. | | |
| NASA Global Contrail Model | Reanalysis data | | *Reanalysis biases:* Temperature and humidity biases in reanalysis data affect contrail formation accuracy. *Static atmosphere:* Model ignores wind shear and dynamic factors in contrail evolution. *Downwash exclusion:* Omits ice particle downwash from wake turbulence, impacting contrail properties. | | |
| ICON-LEM (Contrail Scheme) | Satellite data | | *Satellite retrievals:* Cloud backgrounds cause uncertainties in satellite detection and contrail property estimation. *Parameterization:* Uncertainties in ice nucleation and crystal survival parameterizations affect contrail persistence accuracy. | | |
| LES Model | In-situ measurements | | *Spatial and temporal coverage*: In-situ measurements may miss full variability of contrail properties in large-scale simulations. *Computational limits*: LES models are expensive, limiting simulation scope and number of contrails, affecting behavior capture. *Microphysics schemes:* Uncertainties in ice binning and microphysical processes impact contrail microphysics accuracy. | | |



Bickel et al. (2020) noticed that significant scaling up of aviation density is necessary to obtain
statistically significant results from the simulations. Their study found that the ERF of contrail
cirrus is less than 50% of the respective instantaneous or stratosphere-adjusted radiative forcings.
The best estimate of contrail cirrus ERF is approximately 35%. In comparison, the reduction of
ERF is more substantial for contrail cirrus than for a similar magnitude increase in $CO_2$
concentrations. They identified the main factor contributing to the reduction in contrail cirrus ERF
as a compensating effect of natural clouds that provide negative feedback. Additionally, they
observed that the combined water vapor and lapse rate adjustment, which affects the distribution
of water vapor in the atmosphere and the lapse rate (temperature decrease with altitude), has a less
positive impact on contrail cirrus ERF compared to the reference case of $CO_2$ forcing.
Nevertheless, the negative feedback provided by natural clouds has a more pronounced effect in
reducing contrail cirrus ERF compared to these adjustments.
Overall, Bickel et al. (2020) suggested that contrail cirrus has a lower climate impact than initially
thought, with the reduction in ERF attributed to the compensating effect of natural clouds.
Understanding the specific feedback processes and quantifying the ERF of different climate
forcing agents, such as contrail cirrus, is crucial for accurately assessing their contributions to
surface temperature changes and overall climate dynamics.
While radiative forcing (RF) provides a measure of contrail impact on radiative balance, it doesn't
fully account for the efficacy of different forcing agents in driving surface temperature changes.
Recent studies by Ponater et al. (2021) emphasize the importance of effective radiative forcing
(ERF) as a superior metric. ERF considers feedback mechanisms that influence a forcing agent's
ultimate climate impact. Their findings suggest a potentially lower climate impact from contrail
cirrus than previously assumed based on ERF calculations. However, as acknowledged by Ponater
et al. (2021), further research is needed to confirm the efficacy of ERF in assessing the complete
surface temperature response to contrail cirrus. Direct simulations of this response would be
crucial for validating ERF as a reliable metric for contrail cirrus impact.
Similar to findings by Bickel (2023), this study highlights the potential limitations of using RF
alone to estimate contrail cirrus impact. Here, climate model simulations revealed a significant
reduction in the effective radiative forcing (ERF) of contrail cirrus compared to its conventional
RF. This suggests a potentially lower climate impact from contrail cirrus than previously assumed
based solely on RF calculations. The primary contributor to this reduced ERF is likely a negative
cloud adjustment, where contrail cirrus formation leads to a decrease in natural cirrus cover.
Further research is needed to confirm these findings across different climate models and refine our
understanding of contrail cirrus efficacy in driving surface temperature changes.
Long-lasting contrails contribute to global climate change. These contrails can form cirrus clouds,
which are a type of high-altitude cloud that can trap heat in the atmosphere. This trapping of heat
is known as radiative forcing. While the exact contrail contribution to radiative forcing remains
uncertain, it is thought to be a significant factor in climate change (Brasseur et al., 2016). The RF
metric is a backward-looking measure of the effect of emissions on the radiative flux balance and
is commonly used to compare changes in climate forcings (Wuebbles et al., 2010). The RF linked
to non-$CO_2$ aviation emissions arises from processes occurring over different time scales. Contrails
generally exist for a few hours after an aircraft emissions occur, but other emissions can last much
longer. Effects on the distribution of aerosols and on ozone produced from NOx emissions can
remain for a few days to months, while changes in $CH_4$ can be affected for longer than a decade.
The limitations of using RF as a comprehensive metric for global non-$CO_2$ aviation climate
impacts are well-recognized due to such associated spatiotemporal variations (Wuebbles et al.,
2007). The nonlinear interactions involved make it imprecise to represent the sum of RF for various
non-$CO_2$ components as a single value. Similarly, distinct climate responses are observed for
different forcing mechanisms.
In 2016, Brasseur et al. investigated the radiative forcing (RF) through analyses involving seven
global models (CAM4, CAM5, IGSM, GISS-E2, GEOSCCM, GATOR-GCMOM, and GEOS-
CHEM) as part of the Federal Aviation Administration's (FAA) Aviation Climate Change
Research Initiative (ACCRI) program. The assessment covered climate impacts for 2006, and the
initial five models projected impacts for 2050 scenarios, specifically focusing on selected aircraft
emission components. In a separate study, Chen and Gettelman (2016) used the CAM5 model to
estimate RF for contrails and contrail cirrus in the 2050 future scenario, employing the 2006 AEDT
dataset. Notably, the research highlighted the tendency to overestimate global linear contrail (LC)
net RF when using the natural ice cloud optical property parameterization as a stand-in for contrail
counterparts in modeling studies. Furthermore, the distribution of regional RF indicated that in
densely trafficked airspaces, such as the United States, RF could be up to ten times higher than the
global average (Brasseur et al., 2016).
Schumann and Graf (2013) determined a more significant AIC impact by using both observational
data and the contrail cirrus prediction model (CoCiP; Schumann, 2012). They found an "aviation
fingerprint" ascribed to a daily air traffic cycle within the diurnal cycle of cirrus properties in the
North Atlantic region (NAR), associating with the annual mean diurnal patterns of cirrus cover
and outgoing longwave radiation (OLR) derived from Meteosat data (Graf et al., 2012).
Figure 12 shows the global average annual radiative forcing (RF) values and uncertainty ranges
for persistent contrails alone and together with contrail cirrus. The values are compiled from
selected studies and assessments published since 1999 and include a recent development by
Kärcher (2018). Over time, early assessments of contrail RF have been confirmed and the range
of uncertainty has been narrowed down considerably.
The figure shows that the RF due to persistent contrails alone is -0.01 W/m² (with an uncertainty
range of 0.005-0.03 W/m²), and the RF due to persistent contrails together with contrail cirrus is
~0.05 W/m² (with an uncertainty range of 0.02-0.15 W/m²). (Kärcher, 2018).  Figure. 12 shows
that analyses accounting only for linear contrails underestimate the total RF for contrails. More
recent analyses tend to show an overall warming of around ~45mWm$^{-2}$ (Bier and Burkhardt, 2022).
The models applied by DLR and NCAR's groups are summarized in Table 4 (updated from. Bock
and Burkhardt, 2016). Bock and Burkhardt (2016) evaluated the year 2002 using the AERO2k
inventory and for year 2006 using the AEDT 2006 slant distance inventory. The corrected NCAR
analyses and their more recent results are consistent with those from the DLR modeling studies
(Lee et al., 2021). These findings further amplify the conclusions from Fig. 12.

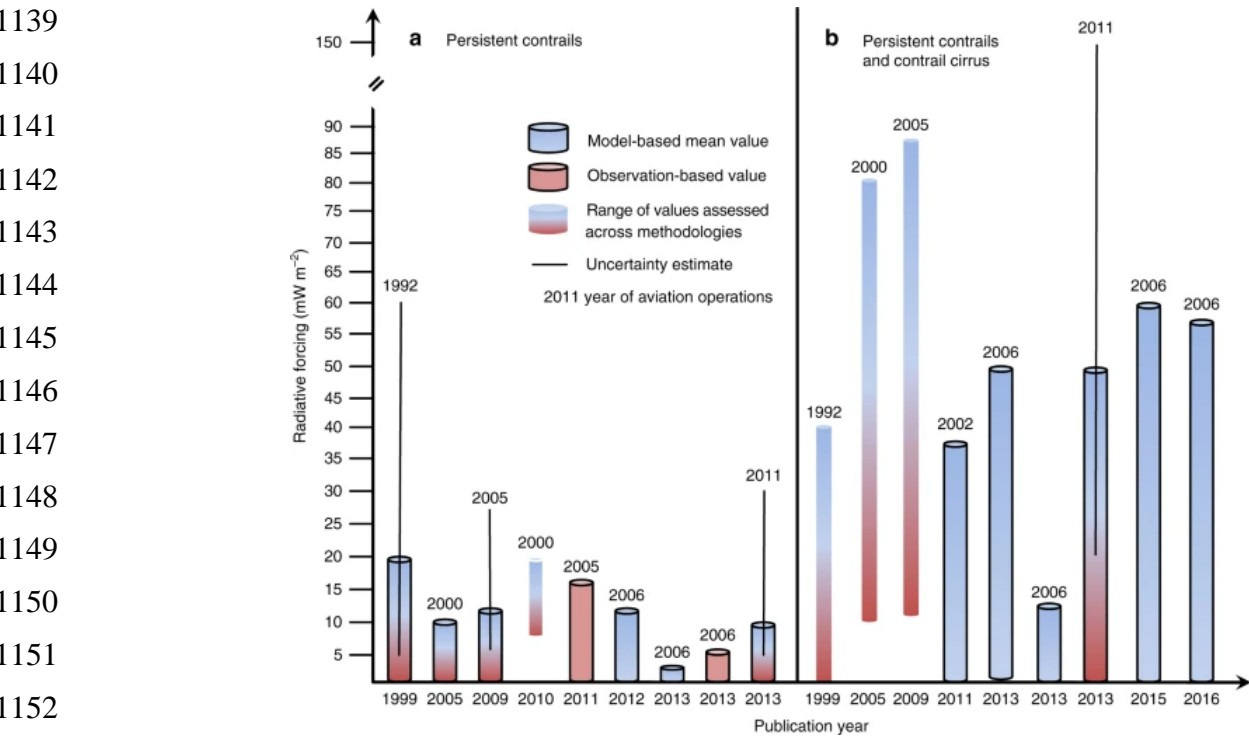

**Figure 12.** Global annual average amount of radiative forcing caused by aircraft-induced clouds
Figure adapted with permission from Kärcher (2018). (Licensed under CC BY 4.0)

## 4.2 Future RF projections

### 4.2.1 DLR's study for future RF projections

Bock and Burkhardt (2019) investigated how changes in air traffic between 2006 and 2050, including both volume increase and upward shift in flight altitudes (0.3-1.5 km), would impact contrail cirrus properties and radiative forcing. Table 5 shows their simulations considering this combined scenario. It projects a fourfold increase in air traffic and a maximum flight density at a lower altitude (200 hPa in 2050 compared to 240 hPa in 2006), which has climate implications.

In Table 5, the air traffic distance is specified as the ground-projected track distance. The coverage is sketched out for all contrail cirrus, and the visibility of contrail cirrus (visible optical depth > 0.05) is indicated in brackets (Bock and Burkhardt, 2016). Radiative forcing estimates exist for both track distance and slant distance within brackets (Table 5). Remarkably, Bock and Burkhardt (2019) determined that the projected future increase in air traffic, coupled with a minor shift to higher altitudes, outcomes in a substantial rise in contrail cirrus coverage, optical depth, and radiative forcing. Particularly, air traffic contrail cirrus radiative forcing is estimated to increase threefold, from 49 to 159 mW m$^{-2}$ (Fig. 13). The findings are a result of a future air traffic inventory, where the measurement of air traffic is signified in terms of track distance (ground projected) rather than slant distance (3-D). Declining the initial contrail ice particle number by 50% results in a substantial drop in the climate impact of contrail cirrus, leading to a global decline in radiative forcing for the year 2050. The reduction is significant, amounting to a 14% reduction from 160 to 137 mWm$^{-2}$ (Fig. 13).

**Table 4.** Summary of existing contrail cirrus simulations and RF from the DLR and NCAR models. Adapted from of Bock and Burkhardt (2016).

| Model | Inventory | Flight distance | RF (mW/m$^2$) | References |
|---|---|---|---|---|
| ECHAM5-CCMod | AERO2k 2002 | track | 35 | |
| ECHAM5-CCMod | AEDT 2006 | track | 49 | *Burkhardt and Kärcher* (2011) |
| ECHAM5-CCMod | AEDT 2006 | slant | 56 | |
| ECHAM4-CCMod | AERO2k 2002 | track | 38 | |
| ECHAM4-CCMod | REACT4C 2006 | track | 45 | *Schumann et al,* (2015) |
| COCIP | AEDT 2006 | flight vectors | 63 | *Schumann et al.* (2015) |
| CAM5 | AEDT 2006 | slant | 13 (57)* | *Chen and Gettelman* (2013) |
| CAM6 | AEDT 2006 | slant | 62 (ERF)** | *Gettelman et al.* (2021) |

* The approximation of Chen and Gettelman (2013) was revised.

**ERF (the scaling technique was derived from Lee et al.2021 for 2006 to 2018 and then 9% per year scaled (2018 to 2019) by Gettelman et al. (2021)

AERO2k – Global aircraft emissions data project for climate impacts evaluation

AEDT- Aviation Environmental Design Tool

REACT4C- Reducing Emissions from Aviation by Changing Trajectories for the benefit of Climate

### 4.2.2 RF projections from the NCAR model

The study conducted by Chen and Gettelman (2016) investigated the radiative forcing curbing from aviation-induced cloudiness, using the Community Atmosphere Model Version 5 (CAM5) for both present (2006) and future (up to 2050) scenarios. The researchers found a projected four-fold increase in global flight distance from 2006 to 2050. Despite this, the simulated radiative forcing from contrail cirrus in 2050 is predicted to reach 87 mWm$^{-2}$, representing a seven-fold rise compared to 2006. This underlines a non-linear correlation between radiative forcing and fuel emission mass, accredited to non-uniform regional escalations in air traffic and changes in contrail radiative forcing sensitivity through different regions.

The CAM5 simulations also indicate that the negative radiative forcing resulting from the indirect effect of aviation sulfate aerosols on liquid clouds in 2050 could be as large as -160 mWm$^{-2}$, a four-fold increase from 2006. Consequently, when considering both aviation aerosols and contrail cirrus, the total radiative forcing in 2050 could possibly employ a cooling impact on the planet. Aerosols, particularly aviation sulfate aerosols distributed at cruise altitudes, may be transferred to the lower troposphere. This procedure raises aerosol concentrations and consequently enhances the cloud drop number concentration and persistence of low-level clouds. On the other hand, the study suggests that aviation black carbon aerosols have an insignificant net forcing affect globally, both in 2006 and 2050.

The researchers focus on specific regions, for instance, Central Europe, eastern North America, and East Asia, and recommend quantitative evaluations of the forcing in each region for several scenarios. For example, in Central Europe, the predicted contrail cirrus radiative forcing in 2050 is anticipated to locally peak at 2 Wm$^{-2}$, marking a 2 to 3-fold increase compared to 2006. In the eastern United States, the contrail cirrus radiative forcing could reach 800 mWm$^{-2}$ in 2050, revealing a higher percentage increase compared to 2006. The most prominent rise in contrail

cirrus radiative forcing is estimated in East Asia, where a 6-fold increase is projected for 2050,
associated with the region's anticipated significant rise in consumption of aviation fuel.
**Table 5.** Overview of the model simulations: Air traffic distance exists as ground-projected track
distance. Coverage is furnished for all contrail cirrus, with coverage for visible contrail cirrus
(visible optical depth $> 0.05$) shown in brackets (Bock and Burkhardt, 2016). The radiative
forcing is represented for both track distance and slant distance, with values enclosed in brackets.
(Adapted from Bock and Burkhardt (2019)

| Background | Inventory | Air traffic volume (km yr$^{-1}$) | Propulsion efficiency | Initial ice number concentration (cm$^{-3}$) | Coverage ( %) | RF (mW m$^{-2}$) |
|---|---|---|---|---|---|---|
| 2006 | 2006 | $3.7 \times 10^{10}$ | 0.3 | 150 | 1.1 (0.7) | 49 (56) |
| 2006 | 2050 Baseline | $15.4 \times 10^{10}$ | 0.3 | 150 | 2.9 (2.0) | 159 (182*) |
| 2050 (RCP6) | 2050 Baseline | $15.4 \times 10^{10}$ | 0.3 | 150 | 2.8 (2.0) | 160 (183*) |
| 2050 (RCP6) | 2050 Scenario1 | $15.4 \times 10^{10}$ | 0.42 | 75 | 2.98(1.7) | 137 (157*) |

RCP6-Representative Concentration Pathway 6.0

*Asterisks denote extrapolated values resultant from the factor resolute by the radiative forcing in 2006, which is connected to air traffic volume and computed using slant distance and track distance (Bock and Burkhardt, 2016).

The inclusion of aviation aerosols in the simulations suggests a slight reduction in positive forcing
over land, as compared to the forcing solely caused by contrail cirrus. On the other hand, over the
ocean, a negative forcing rising from aviation emissions is measured. This can be accredited to the
lower surface albedo and cleaner environment (fewer aerosols) in comparison to land areas. In the
three regions with the highest projected air traffic in 2050 (eastern United States, Central Europe,
and East Asia), aviation aerosols reduce the regionally averaged positive radiative forcing induced
by contrail cirrus by approximately 50%, as indicated by the blue boxes in Fig. 14. The peak
positive forcing within each of these regions is also reduced by 50% due to aviation aerosols. The
study provides detailed regional estimates of these effects (Table 6).
The negative radiative forcing result from aviation aerosols, as detected in this study, supports the
inferences represented by Righi et al. (2013). The extent of the cooling effect is estimated to be
affected by the background cloud drop number concentration.

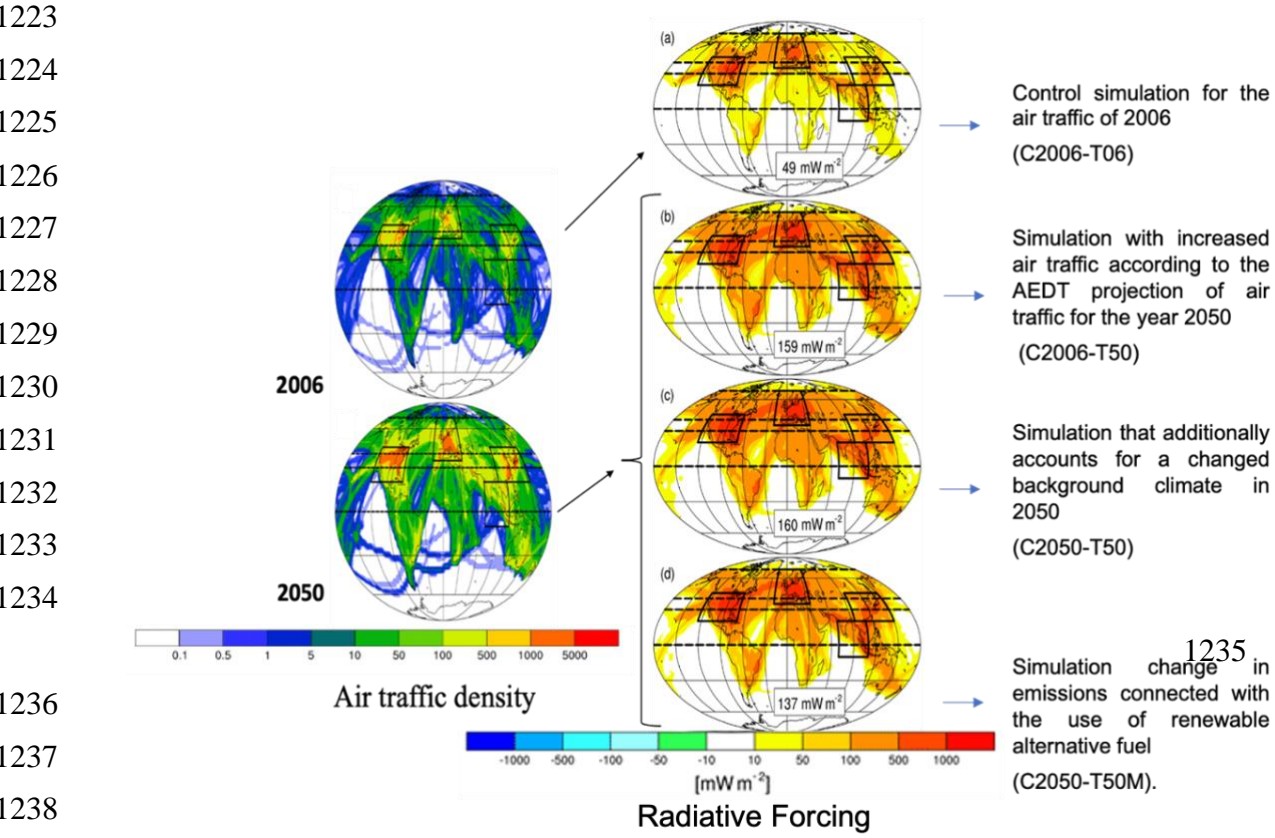

**Figure 13.** The horizontal distribution depicts the vertically incorporated air traffic density (km
m$^{-2}$s$^{-1}$) for the years 2006 and 2050, along with the radiative forcing in scenarios C2006-T06 (a),
C2006-T50 (b), C2050-T50 (c), and C2050-T50M (d) Figure adapted with permission from Bock
and Burkhardt (2019). (Licensed under CC BY 4.0).
**Table 6.** Radiative forcing (mWm$^{-2}$) attributed to aviation H$_2$O emissions, with uncertainties
derived from 2 standard deviations of the four-member ensemble. (Adapted from Chen and
Gettelman, 2016)

| Meteorology | Scenario | Global | North America | Central Europe | East Asia |
|---|---|---|---|---|---|
| Present | 2006 AIR | 12 ± 4 | 195 ± 30 | 483 ± 69 | 41± 8 |
| 2050 RCP4.5 | 2050 BL | 87 ± 6 | 798 ±152 | 1682 ± 535 | 272 ± 39 |
| 2050 RCP4.5 | 2050 SC1 | 76 ± 7 | 724 ± 136 | 1568 ± 489 | 231 ± 36 |
| 2050 RCP4.5 | 2050 SC2 | 76 ± 7 | 724 ± 136 | 1568 ± 489 | 231 ± 36 |
| 2050 RCP4.5 | 2050 SC3 | 80 | 681 | 1952 | 213 |
| 2050 RCP8.5 | 2050 BL | 83 ± 3 | 852 ± 92 | 1558 ± 226 | 248 ± 25 |
| 2050 RCP8.5 | 2050 SC1 | 73 ± 4 | 777 ± 96 | 1442 ±235 | 211 ± 24 |
| 2050 RCP8.5 | 2050 SC2 | 73 ± 4 | 777 ± 96 | 1442 ±235 | 211 ± 24 |
| 2050 RCP8.5 | 2050 SC3 | 75 | 865 | 1597 | 221 |

RCP4.5-Representative Concentration Pathway 4.5
RCP8.5 -Representative Concentration Pathway 8.5.  BL-Base line
SC1. SC2, and SC3- Scenarios 1, 2 and 3

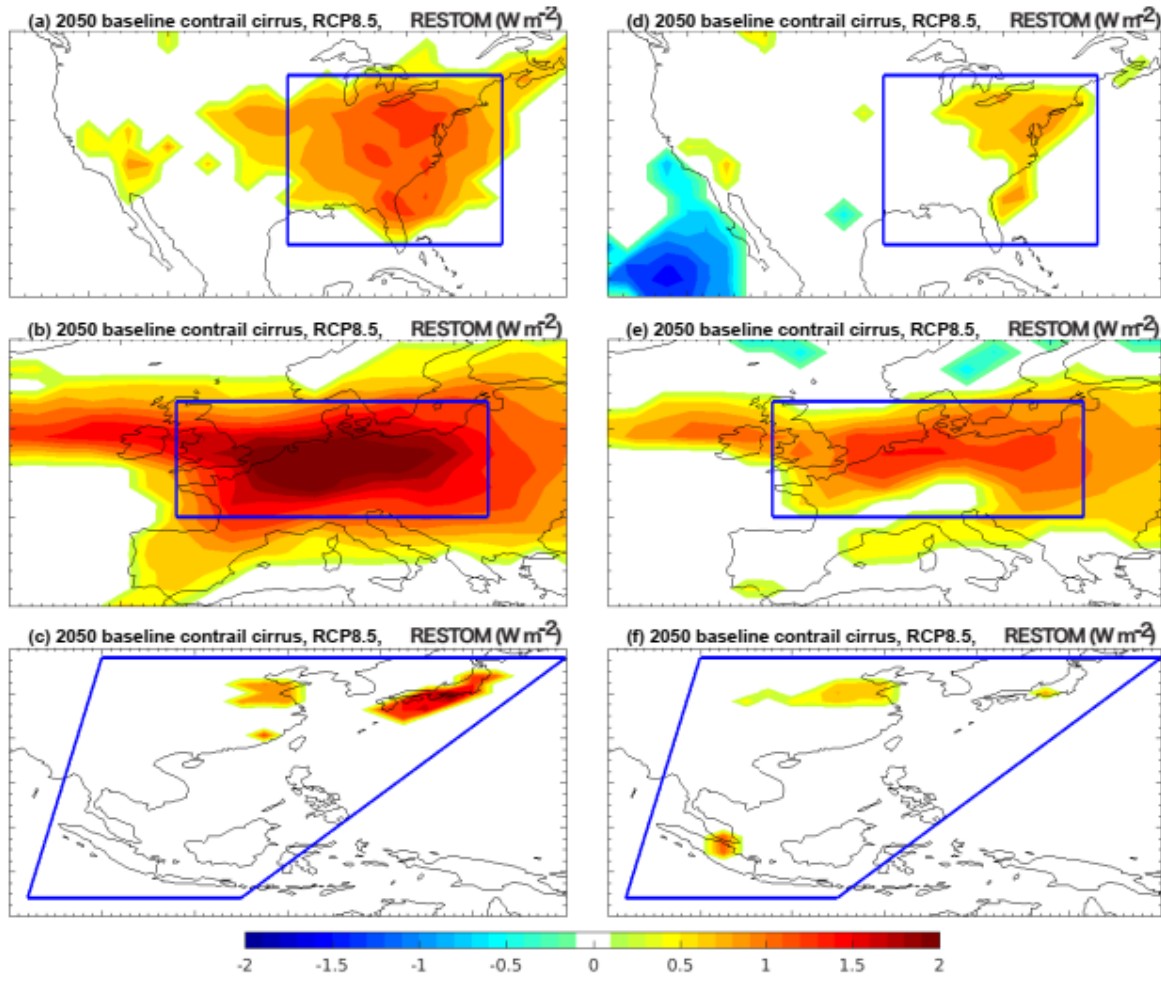


**Figure 14.** Ensemble of the average regional radiative forcing in Wm$^{-2}$, utilizing the baseline emission scenario for the year 2050 with RCP8.5 meteorology, focusing on contrail cirrus alone (a–c) and contrail cirrus combined with aviation aerosols (d–f). The term "RESTOM" represents the alteration in the net residual radiative flux at the top of the model. The plot includes only ensemble-mean perturbations that surpass 2 standard deviations of the averaged control simulations. (Figure adopted with permission from Chen and Gettelman, 2016) (Licensed under CC BY 4.0

**4.3 Issues representing cloud-contrail and contrail-contrail overlaps in modeling studies**

A thorough examination of past methodologies for modeling cloud layer overlaps and several modeling studies recommends that overlapping with other cloud layers is prone to reduce both the shortwave (cooling) and longwave (warming) radiative forcing linked to contrails (Sanz-Morère et al., 2021). However, there is inadequate agreement on how the overlap between clouds and contrails might modify the net radiative forcing due to uncertainties regarding whether it would more strongly mitigate the shortwave or longwave components. Particularly, the precise impact of contrail-contrail overlap on global contrail radiative forcing has yet to be measured. In this section, we review previous literature referring to the treatment of multiple-layer overlap in the context of contrail radiative forcing calculations.

Schumann et al. (2012) determined that the net radiative forcing may augment if contrails overlap
with low-level clouds but could go through significant variations when passing under natural cirrus
clouds. This underscores the significance of precisely modeling natural clouds in contrail
simulations. Nevertheless, when utilizing this method to simulate single contrails, accounting for
the radiative interactions among multiple contrails turns into a challenging task. In prior examples
of modeling contrail-contrail overlap, when simulating contrails in global climate models, various
treatments have been utilized, as outlined in. Contrail parametrizations have been formulated for
ECHAM4 (Ponater et al., 2002; Burkhardt and Kärcher, 2009), where maximum-random overlap
is presumed between contrail and cloud layers, as well as among different contrails (Burkhardt and
Kärcher, 2011; Marquart et al., 2003; Bock and Burkhardt, 2016; Frömming et al., 2011). Rädel
and Shine (2008) and Rap et al. (2010) also use this parameterization, calibrating the outcomes by
using satellite observations. Table 7 summarizes the existing methods for modeling contrail–
contrail overlaps.

**Table 7.** Existing methods for modeling contrail–contrail overlap when estimating global contrail
RF. MRO: maximum-random overlap, defined by Geleyn and Hollingsworth (1978) as assuming
that clouds in adjacent layers maximally overlap, while clouds separated by one or more clear
layers randomly overlap (Adapted from Sanz-Morère et al. 2021).

| Source | Model used to represent contrail–contrail overlap |
|---|---|
| Minnis et al. (1999) | No overlap considered (fractional coverage from observations) |
| Marquart et al. (2003) | MRO in the vertical for each column |
| Rädel and Shine (2008) | Random overlap |
| Rap et al. (2010) | Random overlap |
| Frömming et al. (2011) | MRO in the vertical for each column |
| Burkhardt and Kärcher (2011) | MRO in the vertical for each column |
| Chen and Gettelman (2013) | Zero contrail–contrail overlap in grid box |
| Schumann and Graf. (2013) | Linear RF addition |
| Bock and Burkhardt (2016) | MRO in the vertical for each column |


Chen and Gettelman (2013) included contrails in the CAM5 model by portraying them as an
increase in the 3-D cloud fraction, where the model presumes maximum-random overlap.
Nevertheless, in their method, they considered zero overlaps between newly formed contrails when
placed at the same vertical level (approximately 1 km). Finally, the CoCiP Lagrangian contrail
model (Schumann, 2012) reports contrail-contrail overlaps indirectly by linearly summing the
radiative forcing of all contrails, taking into consideration any observed cirrus present above the
simulated contrail. Nonetheless, this method does not clearly consider the overlap between
simulated contrails. Inconsistencies rise in how contrail-contrail overlaps are modeled across
various studies. The most efficient approach remains uncertain, and as of now, no analysis has
measured the impact of contrail-contrail overlaps on global contrail RF. Since further expansion
of the aviation sector is expected, it is anticipated that an increased number of instances of contrail
overlap will occur. Hence, a more inclusive understanding of the degree and dynamics of contrail-
contrail overlap is a necessity. Table 8 summarizes the agreement and disagreement between
climate models on contrail cirrus radiative forcing.

## 5 Observation Datasets for Contrail Studies

The vast majority of observations, comprised of airborne, satellite, and ground-based approaches, encompass jet aircraft exhaust contrails from 1972 onward. Nevertheless, it is remarkable that records of contrail observations date back to as early as 1915, as published by Ettenreich in 1919.

Early measurements performed behind a propeller-driven aircraft offered data supporting the concept that contrail ice formation requires liquid saturation to originate, associating with the Schmidt-Appleman criterion that is now well-established (Schmidt, 1941, Appleman,1953 and Schumann, 1996.). These remarks, which encompassed the collection of ice particles on impactors and halo observations, also furnished initial insights into the size and shape of both contrail and cirrus ice particles (Weickmann, 1945).

The precise testing of models necessitates reliable quantitative data, covering not only information on contrail and plume properties but also on contrail age, the generating aircraft, atmospheric conditions during contrail formation, and the observational methods employed (Schumann et al., 2017). Mean properties of individual contrails are characterized for a wide range of jet aircraft as a function of age during their life cycle from seconds to 11.5 h (7.4-18.7 km altitude, -88 to -31°C ambient temperature), based on a compilation of about 230 in-situ and remote sensing measurements (Schumann et al., 2017). Contrails from individual aircraft can remain visible for extended periods, as reported in analyses by Minnis et al. 1998, 2013) and Vázquez-Navarro et al. (2015). Schumann et al. (2017) furnished a brief description of individual datasets, integrating new analyses for their study, and combined them to create a "contrail library" (COLI). This dataset was then compared with the outcomes of the Contrail Cirrus Prediction (CoCiP) model. The observations corroborate that the quantity of ice particles in contrails is controlled by both the engine exhaust and the formation process in the jet phase. Some particle losses ensue in the wake vortex phase, with succeeding gradually diminishes over time.

Publicly available air traffic data are usually limited, but certain projects have gathered flight data from sources such as air traffic control or ground-based observations (Schumann et al., 2013; Schumann et al., 2017). Distinguishable projects with air traffic subsets since 2005 are accredited. Garber et al. (2005) compiled traffic data for the United States and southern Canada for the years 2000-2005. In Germany, the Deutsche Flugsicherung (DFS) has been gathering and filing away traffic data from 2006 onwards for more new projects. The Aviation and Climate Change Research Initiative (ACCRI; Brasseur et al., 2016) furnished a global waypoint dataset for the year 2006 to researchers included in the ACCRI project.

The existing observation datasets on contrail properties and formation conditions are constantly expanding with the inclusion of innovative resources. The recently developed Global Aviation Emissions Inventory based on ADS-B (GAIA) (Teoh et al., 2024). leverages historical flight paths derived from Automatic Dependent Surveillance-Broadcast (ADS-B) technology, coupled with reanalyzed weather data from 2019 to 2021. GAIA paints a detailed picture of global aviation activity, encompassing the spatial and temporal variations in flight patterns, fuel consumption, and emissions of key pollutants like CO2, NOx, and particulate matter. This information proves particularly valuable for contrail research by offering insights into the connections between aircraft types, flight routes, and the potential for contrail formation in diverse regions. For example, by analyzing short-haul versus long-haul flight emissions within GAIA, researchers can glean insights into the contribution of various aircraft categories to contrail formation patterns across different geographical areas.

Table 8 . Agreement and disagreement between climate models on contrail cirrus radiative forcing.
(The models discussed include CoCiP, ECHAM5-HAM, CAM6, and the Stanford Global Model. Detailed
descriptions of these models can be found in Sections 3.2.1 through 3.2.4).

| Aspect | Agreement | Disagreement | Potential Causes | Recommendations for Future Research |
|---|---|---|---|---|
| **Net Radiative Forcing (RF)** | Likely positive net RF, signifying warming | | | |
| **Longwave vs. Shortwave RF** | | Models show some variation in the relative contribution of longwave (warming) and shortwave (cooling) components to the net RF depending on factors like ice crystal shape and optical depth. | Complexity of ice crystal parameterizations Treatment of solar radiation transfer | Improve ice crystal parameterizations in models to better capture the variation in radiative properties. Conduct studies to refine the treatment of solar radiation transfer in contrail simulations. |
| **Regional Variations** | Models concur on significant regional variations in RF | | | |
| **High-traffic vs. Low-traffic areas** | | Models show discrepancies in the magnitude of the regional variations. | Input data on air traffic patterns | Improve the quality and resolution of air traffic data used in climate models. |
| **Future RF Projections** | Models predict a significant increase in contrail cirrus RF by 2050 due to rising air traffic and flight altitude | | | |
| **Impact of aviation aerosols** | | Models show variations in the projected net effect (warming vs. cooling) due to the counteracting impact of aviation aerosols on low-level clouds. | Uncertainties in how models represent aviation aerosol properties and their interactions with clouds. | Conduct dedicated studies to improve the representation of aviation aerosols and their interactions with clouds in climate models. |
| **Cloud Overlap** | Overlap with other clouds (natural cirrus, low-level clouds) can affect RF, but the extent of impact (shortwave vs. longwave dominance) is uncertain. | Contrail-contrail overlap is not well understood and might be more frequent in the future. | Need for a better understanding of cloud-contrail and contrail-contrail overlap and their impact on global RF. | Develop improved methods for simulating cloud-contrail and contrail-contrail overlap in climate models. Conduct observational studies to quantify the frequency and radiative effects of contrail overlap. |

- *CoCiP:* Agreements on overall contrail effects, but some discrepancies in initial ice crystal formation.
- *ECHAM5-HAM:* Captures basic contrail formation but has different ice crystal properties affecting long-term impacts.
- *CAM6:* Aligns with others in radiative forcing but shows minor differences in contrail-induced cloud changes.
- *Stanford Global Model:* Agrees on initial formation but has contrasting plume mixing and microphysical schemes leading to varied impact predictions.

By seamlessly integrating GAIA with existing observation datasets like COLI (Schumann et al.,
2017), researchers can cultivate a more nuanced understanding of the interplay between aviation
activity, atmospheric conditions, and contrail occurrence. This enhanced knowledge ultimately
empowers the improvement of contrail prediction models and informs strategies to mitigate their
impact on climate.
Particulars about aircraft assets, including size, mass, speed, fuel consumption, and propulsion
efficiency, were tracked from various references, including the BADA (EUROCONTROL, 2009).
Added engine properties, such as fuel consumption and emissions at surface pressure, can be
attained from the ICAO Aircraft Engine Emissions Databank (EASA, 2023) and the 2006 AEDT
emissions inventory (Barrett, 2010; Wilkerson, 2010).
Poll (2018) devised a straightforward model to estimate fuel burn for commercial aircraft, offering
a transparent, publicly accessible option apart from the BADA and PIANO codes (Piano-X,
software, Lissys Ltd., 2008)). Utilizing this method necessitates understanding the Mach number
and flight level where an aircraft, under specific conditions, achieves its absolute minimum fuel
burn rate. Yet, acquiring this data proves challenging, although an initial effort is outlined in Poll
and Schumann (Poll, 2018), where input files are supplied for 53 aircraft types.
Two sets of observational-based capabilities exist that can provide the coverage needed to help
evaluate the global modeling analyses for treating the effects of contrails, the Contrail Library
(COLI) and the satellite-based record. The Contrail Library results are useful for evaluating limited
areas of contrails but not for global analyses.
**5.1 Contrail Library (COLI)**
The COLI database includes 236 entries describing properties of contrails with known ages,
including mean data for one hundred cases of in-situ measurements, more than 70 cases from
ground-based and airborne lidar observations, 50 cases from satellites, and a few camera
observations. The data come from 33 observation projects during the last 45 years. The comparison
of the data from various measurements shows notable differences among the various instruments
utilized across the dataset. Nevertheless, a notable accord endures among each approach, as well
as uniformity between in-situ and remote sensing results, along with agreement with Schumann et
al (2017) model outcomes. Moreover, in-situ data resultant from contrail measurements, the
investigation integrates remote sensing data found from ground-based or airborne lidar and
spectroradiometer, satellites, cameras, and visual interpretations. Table 9 summarizes some of the
projects that contributed to the COLI database, including the aircraft used and the primary location,
the mode of reference, and the primary reference of the study.
Accurate forecasting of contrail occurrence and persistence holds significant importance in
addressing their potential climate impact. For this purpose, research conducted by Gierens et al.
(2020) underscores the limitations of reanalysis data, primarily due to their coarse vertical
resolution. This study shows the necessity for high-resolution observational data, especially
concerning temperature and relative humidity in the upper troposphere and tropopause region, to
understand contrail formation and behavior better. Initiatives such as the In-service Aircraft for a
Global Observing System (IAGOS) (Gierens et al., 2020) are pivotal in providing such data by
utilizing advanced instruments installed on passenger aircraft. The assessment of contrail
formation models demands dependable data encompassing both contrail characteristics and
atmospheric conditions during their formation. While the study by (Gierens et al., 2020) evaluates
the predictability of contrails using the Schmidt-Appleman criterion based on reanalysis data, it
underscores the critical role of high-resolution data in assessing more advanced contrail prediction
models. The availability of observational datasets containing parameters such as temperature,
relative humidity, and air traffic data, as discussed in this section, is indispensable for such model
assessments.

## 5.2 Satellite-based datasets

Over the past two decades, the detection of contrail in satellite imagery has primarily depended on
the algorithm developed by Mannstein et al. (2003) (e.g., Palikonda et al., 2005; Vazquez-Navarro
et al., 2010). This algorithm contains a series of convolution and thresholding operations related
to brightness temperature images and, subsequently the detection of linear associated components
of fitting size. While the algorithm has been fine-tuned to attain either high precision or high
remembrance in contrail detections, no single model has simultaneously surpassed in both aspects.
Subsequently, empirical observations of contrail coverage often need broad lower-bound/upper-
bound approaches (Duda et al., 2013). Minnis et al. (2013) reported the properties of contrail cirrus
clouds formed during 11 different contrail outbreaks, in the context of objectively determined
linear contrails and their properties. It was observed that the ratio of contrail cirrus to linear
contrails is substantially affected by the satellite analysis algorithm utilized to allocate linear
contrails. Furthermore, this ratio seems to be affected by the presence of overlapping contrails,
which can obscure individual contrails.
The contrail cirrus optical depths were found to be 2-3 times greater than their linear contrail
counterparts and the associated ice crystal particle diameters were 20% greater than the contrail
particle sizes. The analyses of such satellite datasets are likely to be useful for evaluating global
models of contrail radiative effects.
An exception to the application of the Mannstein et al. (2013) algorithm is prominent in Kulik
(2019) and Meijer et al. (2021). These analyses employed a deep learning model for pixel-level
contrail detection using GOES-16 satellite imagery. Upon detecting a contrail, these studies
facilitate the assessment of its lifetime impact and the ascription of potentially relevant flights
(Vazquez-Navarro et al., 2010; Vazquez-Navarro et al., 2013; Vazquez-Navarro et al., 2015.
Mutually, these practices furnish a means to evaluate the efficacy of flight diversions in avoiding
contrail formations. McCloskey et al. (2021) officially released the first large dataset of pixel-level
contrail locations in Landsat-8 satellite imagery labeled by humans. Labelers should be able to
accurately differentiate contrails from naturally occurring cirrus due to Landsat-8's high spatial
resolution and the advected flight history information provided. This dataset will be useful in
benchmarking contrail detection models and validating contrail research in collocated
geostationary satellite imagery.
Satellite-based datasets offer valuable insights into global contrail occurrence and variability,
although challenges arise when estimating contrail radiative forcing. Retrieving contrail optical
thickness, a crucial parameter for such calculations, from current satellite observations, is
particularly difficult. Moreover, distinguishing contrails from overlapping features and natural
cirrus clouds remains challenging, especially with limited spatial resolution. Addressing these
challenges is essential for enhancing the contribution of satellite observations to contrail radiative
forcing estimations. Advancements in retrieval algorithms for contrail optical thickness and
methods to differentiate contrails from overlapping features are necessary for more accurate
assessments.
Despite these challenges, satellite-based contrail datasets have significant potential for monitoring
contrail occurrence and coverage over time. This information is valuable for evaluating the
effectiveness of flight diversion strategies aimed at minimizing contrail formation. Furthermore,
the emergence of human-labeled contrail datasets like McCloskey et al. (2021) provides a valuable
resource for training and validating future advancements in contrail detection using deep learning
and machine learning algorithms. Continuous collaboration among satellite data providers,
atmospheric scientists, and AI researchers is vital for realizing the full potential of satellite-based
contrail datasets for contrail mitigation and climate impact assessments.
Ng et al. (2023) presented a human-labeled dataset named open contrails to train and evaluate
contrail detection models based on GOES-16 Advanced Baseline Imager (ABI) data. Ng et al.
(2023) proposed and evaluated a contrail detection model that incorporates temporal context for
improved detection accuracy. The human-labeled dataset and the contrail detection outputs are
publicly available on Google Cloud Storage at gs://goes_contrails_dataset.
Persuading policymakers and airlines about the climate benefits of flight diversions becomes more
reasonable with the availability of satellite corroboration capabilities. Geostationary satellites like
GOES-16 and low-earth orbit satellites such as Landsat-8 present both complementary strengths
and weaknesses when employed for the task of contrail detection (i.e., coverage versus persistence
of the observations). An automated contrail detection system is essential for developing and
evaluating contrail avoidance systems. Deep neural networks can be employed as an automated
contrail detection system, a form of artificial intelligence (AI) and machine learning (ML), that
has been advanced to connect and categorize contrails in satellite imagery or other relevant data
sources. This system can effectively analyze complex patterns and features associated with
contrails, by leveraging the competencies of deep neural networks. By utilizing deep neural
networks and machine learning techniques, the automated contrail detection system offers an
efficient and reliable solution for detecting and monitoring contrails, facilitating further research
and analysis of their impact on climate and aviation.
**Table 9.** Observations from the year 2000 to 2014 were incorporated into the COLI for the study
of contrail (Adapted from Schumann et al. 2016).

| Year | Project name | Source aircraft | Carrier/location | Measurement | Reference |
|------|--------------|-----------------|------------------|-------------|-----------|
| 2000 | Cluster | airliners | Great Lakes | satellite | Duda et al.,(2004) |
| 2001 | Shutdown | B747+C fighters | Northwestern USA | satellites | Minnis et al.,(2002) |
| 2002 | CRYSTAL-FACE | WB-57 | NASA WB-57 | in situ | Gao et al.,(2006) |
| 2003 | Fallstreaks 2003 | airliners | Goddard | lidar | Atlas and Wang (2010) |
| 2005 | PAZI-2 | airliners | DLR Falcon | in situ | Febvre et al.,(2009) |
| 2006 | CR-AVE | WB-57 | NASA WB-57 | in situ | Flores et al.,(2006) |
| 2008 | CONCERT | airliners | DLR Falcon | in situ | Voigt et al.,(2010) |
| 2008 | ACTA | airliners | Europe andNorth Atlantic | satellite | Vázquez-Navarro et al.,(2015) |
| 2011 | CONCERT2011 | airliners | DLR Falcon | in situ | Kaufmann et al.,(2014) |
| 2011 | COSIC | Bae 146 | FAAM Bae-146 | in situ | Jones et al.,(2012) |
| 2012 | Cameras | airliners | Munich | cameras | Schumann et al.,(2013) |
| 2014 | ML-CIRRUS | B772 or F900 | HALO | in situ | Voigt et al.,(2016) |


Siddiqui (2020) conducted a study where a neural network was used to distinguish contrail cirrus
clouds from regular cirrus clouds on TSI images. The study found that the neural network
triumphed with a high accuracy of 98.5% on the validation set, implying that the model was able
to learn significant visual features of contrails that differentiate them from regular cirrus clouds.
The success of this model opens various practical applications for studying contrails. For instance,
the model can be used to analyze images from different months of data, agreeing for researchers
to investigate the frequencies of contrail occurrences. This can impart valuable insights into air
traffic trends, as well as how physical conditions like temperature and humidity vary seasonally
and their potential impact on contrail formation. Researchers can better understand contrail
behavior and their relationship with various environmental factors by leveraging the neural
network's ability to detect and classify contrails accurately. This knowledge can be used to refine
models and predictions related to contrail formation and persistence and contribute to a more
comprehensive understanding of aviation's impact on the atmosphere.

## 6 Supersonic contrails

Contrail studies have primarily focused on subsonic aviation due to the predominance of subsonic
air traffic since the retirement of the Concorde in 2003. Contrail formation and persistence for
supersonic commercial aircraft pose challenges -- persistent contrails are unlikely to form in the
stratosphere when these aircraft are flying at supersonic speeds but societal requirements for flying
at subsonic speeds over land to avoid noise issues would still potentially produce persistent
contrails. The importance of understanding the global climate effect of contrails linked to
supersonic aviation is well-established (Stenke et al., 2008). Matthes et al. (2022) recently
published a comprehensive review on the climate effects of supersonic aviation, encompassing
non-CO2 effects like contrail formation. Their work highlights the need for a thorough
understanding of existing research in this area and uncertainties persist regarding parameters
controlling contrail ice crystal distribution evolution for these aircraft. Consequently, the presence
or absence of contrails becomes a discernible marker of the aircraft's speed regime. Additionally,
limited research, such as observations by Schumann et al. (2017) suggested the possibility of
contrail formation even at high altitudes (18.7 km) under specific atmospheric conditions. These
factors highlight the need for a comprehensive understanding of contrail formation and persistence
for future supersonic aircraft.

## 7 Uncertainties and Research Gaps

### 7.1 Uncertainties

There are several crucial areas of uncertainty and research gaps that require focused investigation
to achieve a more comprehensive understanding of contrail formation, evolution, and their effects:
- *Humidity Observations and Predictions:* An important uncertainty lies in the accuracy and
reliability of humidity observations and predictions at different altitudes. Improved
measurements and modeling of humidity are critical for assessing the conditions conducive to
persistent contrail formation and their subsequent behavior. Addressing this uncertainty will
lead to more precise insights into contrail persistence and its associated radiative impacts.
- *Interaction with Natural Cirrus Clouds:* The interplay between natural cirrus clouds and
contrails initiates an additional level of uncertainty, fostering obscuring the understanding of
their individual and mutual effects on radiative forcing (RF).
- *Regional Variations in Atmospheric Conditions:* The sensitivity of contrail radiative impacts
to regional variations in atmospheric conditions is not well understood and remains uncertain.

- *Uncertainties and Soot Particles*: The role of soot particles in contrail formation remains subject to uncertainties, particularly regarding their ability to nucleate ice. Despite the presence of numerous ice particles in fresh contrails, ongoing debates, as highlighted in recent studies such as Righi et al. (2021), question whether soot particles directly facilitate ice nucleation. Clarifying this relationship is essential for comprehending the impact of soot emissions on contrail formation and subsequent evolution. To address this knowledge gap, further research efforts are imperative to explore the influence of soot particles on ice nucleation processes within contrails.

- *Radiative Transfer Scheme:* Quantifying the radiative forcing exerted by contrails on Earth's climate remains an intricate task. Studies by Schumann and Graf (2013) identified a distinct signature, an "aviation fingerprint," in cirrus cloud cover and outgoing longwave radiation (OLR) that mirrored air traffic patterns. This observation suggests a robust correlation between aviation activity and cirrus cloud formation. Their analysis even estimated a regional increase in heat-trapping (longwave radiative forcing) of 600-900 $mWm^{-2}$, highlighting the potential influence of aviation on these high-altitude ice clouds. However, a crucial challenge lies in accurately modeling the net impact on Earth's energy balance. This difficulty stems from the fact that different climate models employ various approaches to simulate radiative transfer processes, which govern the movement of heat through the atmosphere. These variations in how models handle these radiative processes can significantly influence the estimated impact of contrails on our planet's energy equilibrium.

Significant uncertainties persist regarding the radiative forcing caused by aircraft contrails. These uncertainties arise from numerous factors, starting with uncertainties in the background meteorology and specific aircraft emissions. When assessing the climate impact of contrail cirrus, the primary uncertainties lie in (1) determining upper tropospheric water vapor concentrations and regions of supersaturation; (2) adequately representing contrail cirrus processes in global models; and (3) understanding the radiative response resulting from the contrail cirrus presence, typically, only approximate assessments of uncertainties associated with these processes are furnished.

Assessing the radiative response of contrail cirrus involves handling several crucial uncertainties, including:

*Uncertainties in Contrail Cirrus Radiative Forcing:* Estimating the radiative forcing (RF) of contrail cirrus clouds is inherently uncertain. Lee et al. (2021) categorizes these uncertainties into two main areas: (A) uncertainties related to the radiative response of contrail cirrus and (B) uncertainties associated with the upper-tropospheric water budget and the contrail cirrus scheme within climate models.

(A) *Uncertainties in Radiative Response*

Radiative Transfer Scheme (A1): Myhre et al. (2009) estimated an uncertainty of approximately 35% arising from the radiative transfer scheme employed in climate models. This scheme simulates how different wavelengths of radiation interact with various atmospheric components, including contrail cirrus.

*Cloud* Heterogeneity (A2*):* The inhomogeneity of ice crystals within a model grid box, vertical cloud overlap, and limitations of plane-parallel geometry contribute an estimated uncertainty of 35% (Carlin et al., 2002; Pomroy and Illingworth, 2000; Gounou and Hogan, 2007).

*Ice Crystal Habit* (A4): Variations in the shape and form (habit) of ice crystals within contrail cirrus can significantly impact how they interact with radiation. Markowicz and Witek (2011) suggest an uncertainty of 20% due to this factor.

*Soot Core Impact* (A5): The radiative transfer uncertainty associated with soot cores embedded within contrail cirrus ice crystals is significant, particularly for shortwave albedo (reflectivity), but remains unquantified (Liou et al., 2013). Soot cores can absorb radiation, potentially leading to a warming effect that is not currently accounted for in models.

(*B) Uncertainties in Water Budget and Contrail Cirrus Scheme*

*Ice Supersaturation* (B1): Uncertainties in the representation of upper-tropospheric ice supersaturation (Lamquin et al., 2012) contribute an estimated 20% uncertainty.

*Ice Crystal Number Density* (B2): and Radiative Transfer for Young Contrails (A3): Uncertainties in ice crystal number densities (Karcher et al., 2015, 2018) are linked to factors like water vapor saturation and soot emissions. Lee et al. (2021) recognized the dependence between A3 (radiative transfer for young contrails) and B2 (ice crystal number density). Consequently, they included the uncertainty from A3 within their overall category B uncertainty estimate.

*Contrail Cirrus Lifetime* (B3): The effect of contrail cirrus lifetime on estimated RF is relatively small, with an associated uncertainty estimated at 5-10% (Chen and Gettelman, 2013; Newinger and Burkhardt, 2012). While Lewellen (2014) highlights the potential importance of lifetime through large-eddy simulations demonstrating contrail cirrus lifetimes exceeding 40 hours and widths exceeding 100 km, several factors contribute to the seemingly counterintuitive finding in this context of estimated RF. Lewellen's (2014) simulations highlight the potential significance of contrail cirrus with extended lifetimes, particularly for their ice crystal surface area and overall radiative impact. However, the study focuses on the integrated effect over the entire lifetime, whereas the uncertainty estimate (5-10%) pertains to the impact of variations in lifetime on the average RF. Further research is needed to better understand the complex interplay between contrail cirrus lifetime, ice crystal growth processes, atmospheric conditions, and their combined effect on radiative forcing.

*Air Traffic Data Resolution* (B4): Sensitivity studies suggest an uncertainty of about 10% arising from the temporal resolution of air traffic data used in climate models (Lee et al., 2021).

*Natural Cloud Feedback* (B5): The feedback mechanism by which contrail cirrus alters the upper tropospheric water budget, affecting natural clouds, introduces a significant uncertainty that remains unquantified (Burkhardt and Karcher, 2011; Schumann et al., 2015). Lee et al. (2021) assumed an uncertainty of 15% for this factor.

*Initial Ice Crystal Assumptions* (B6): The estimation of RF by Chen and Gettelman (2013) is subject to uncertainties associated with assumptions about initial ice crystal radii and contrail cross-sectional areas, leading to an estimated uncertainty of 33%.

Overall Uncertainty

Following Lee et al. (2021), the uncertainty in the radiative response to contrail cirrus (excluding soot cores) is estimated to be around 55%, assuming independence of different uncertainties. This aligns with their findings and highlights the dominance of radiative response uncertainties (category A) in the overall uncertainty budget. Category B uncertainties (water budget and contrail

cirrus scheme) remain significant and require further investigation, particularly the combined
effect of ice crystal number density and radiative transfer for young contrails.
The Uncertainty Tightrope
While scientists continue to refine their understanding of how contrails form and influence our
climate, the ever-present threat of climate change demands that we don't wait for complete
certainty before taking action. The evidence of our warming planet is undeniable, and aviation's
contribution to this issue necessitates exploring all possible solutions, even in the face of some
lingering unknowns.
Fortunately, research like that conducted by Frias et al. (2024) offers a glimmer of hope. Their
study suggests that practical strategies to avoid contrail formation can be implemented with
minimal disruption to flight operations, while significantly reducing the overall climate impact of
these contrails. This finding underscores the potential for immediate action alongside ongoing
scientific inquiry. By embracing a two-pronged approach that combines continued research with
real-world mitigation efforts, we can work towards minimizing aviation's environmental footprint.
**7.2 Research needs and gaps**
In the pursuit of a comprehensive understanding of contrail impacts on climate, there is a critical
need to establish clear research needs and requirements.
Improving weather models demands better predictions of humidity and clouds with enhanced
resolution in time, space, and altitude. The models must also adapt to evolving weather conditions.
Accurate water vapor data during cruise altitudes is crucial, emphasizing the need for small, cost-
effective humidity sensors. To achieve this, there is a need for the production, testing, and
evaluation of precise sensors. Lidar technology and weather model assessments play a key role in
advancing observational capabilities for more accurate predictions. Enhancing cirrus cloud
forecasts requires a dual focus on data and artificial intelligence. Ensemble simulations are vital
for estimating uncertainties in cirrus cloud predictions. Additionally, there is an urgent need for
more climate and contrail models, ensuring a comprehensive understanding and evaluation of these
complex atmospheric processes.
Gierens et al. (2020) highlighted a significant issue with the ERA-5 reanalysis data, showing that
it underestimates the frequency and degree of ice supersaturation. This underestimation impacts
the reliability of forecasts for contrail persistence and the accuracy of estimates for contrail optical
thickness and instantaneous radiative forcing. Their research, which involved comparing
predictions from weather and climate models with actual atmospheric data from the
IAGOS/MOZAIC project and the EMAC model, revealed that while temperature predictions are
generally accurate, relative humidity predictions are often significantly off. This discrepancy
undermines the effectiveness of using reanalysis data to predict real-world contrail formation along
aircraft trajectories. The study underscores the necessity of improving ice supersaturation
predictions to enhance the reliability of contrail avoidance strategies and ensure they can be
operationally viable (Gierens et al.,2020).
Research by Ovarlez et al. (2000) highlights discrepancies between measured water vapor content
and data from weather forecasts. Their findings emphasize the importance of accurate humidity
measurements, particularly for studies on contrail formation and persistence. Weather forecasts
might underestimate actual humidity, potentially leading to inaccurate assessments of contrail
behavior. This underlines the need for ongoing research to improve the accuracy of humidity data
in weather models.
Furthermore, high-quality observations, effectively integrated into weather models, are crucial for
setting accurate starting points for forecasts (Bauer et al., 2015). This highlights the need for robust
data collection strategies that combine information from diverse sources. These sources can
include in-situ measurements taken directly within the atmosphere, radar systems that track
precipitation patterns, and satellite sensors that provide a global view. Advanced data assimilation
techniques then play a vital role by merging these observations with model outputs. This combined
approach creates a more precise picture of the initial atmospheric state, ultimately leading to
improved weather forecasts (Bauer et al., 2015).
Sun and Roosenbrand (2023) reported the limitations in traditional computer vision approaches
for contrail detection, particularly in handling the complexity of satellite images under varying
conditions. Earlier machine learning methods relied on simpler convolutional neural network
models, primarily demanding extensive labeled data for contrail presence determination. Notably,
there is a gap in the existing literature, lacking research specifically tailored to machine learning
approaches for contrail detection, setting it apart from other image segmentation or detection tasks.
A significant challenge lies in the absence of adequate loss functions optimized for linear features
like contrails during training, making detection notably difficult at lower resolutions, especially
when multiple contrails are close. Addressing these gaps is crucial for advancing the efficacy of
contrail detection methods (Sun and Roosenbrand, 2023).
While a complete understanding of contrail effects on climate may not be achievable, further
research is crucial to reduce uncertainties and improve our ability to predict and mitigate their
impact. Here are some of the key areas of research gaps that should be explored to address existing
uncertainties:
▪ *Properties of Contrail Particles*: Understanding contrail effects on climate requires
reflection of key factors such as particle size, layer height, cloud overlap, and optical
properties. Larger particles tend to have a different radiative impact than smaller ones. The
altitude of contrail formation impacts their interaction with solar and terrestrial radiation.
Precise climate modeling requires proper accounting for cloud overlap, impacting radiative
calculations. Contrail optical properties, comprising albedo and emissivity, determine their
reflective and trapping abilities for solar and terrestrial radiation. The concentration and
size distribution of contrail particles affect their formation and radiative impact. Accurate
estimation of contrail forcing, important for climate modeling and policy decisions,
depends on understanding particle number, size, and layer height.
▪ *Assessment of Effective Radiative Forcing:* Effective Radiative Forcing (ERF) is a valuable
tool for evaluating the global mean radiative impact of contrail cirrus on climate. However,
it's important to recognize that ERF primarily focuses on this average effect and may not
fully capture regional variations or complex feedback mechanisms associated with contrail
formation and persistence.
▪ *Emissions and Alternative Fuels*: Corroborating a precise connection between soot and
contrail ice crystal numbers is important for understanding the environmental impact of
sustainable aviation fuels, such as biofuels. Biofuels are being studied as a more sustainable
alternative to conventional aviation fuels, and evaluating their impact on contrail formation
is crucial for assessing environmental benefits. Scientific research, notifying regulatory
decisions, relies on the interpretation of the correlation between soot and ice crystal

numbers. This connection is integral for complete visions of the environmental implications of alternative aviation fuels. Precise data on contrail formation, containing the impact of biofuels, is crucial for climate models predicting the impact of aviation emissions on climate. The experimental establishment of the soot-ice crystal connection gives this understanding.

- *Properties of Soot Particle*: A study into the bimodal size distributions of soot particles in aircraft emissions is critical for focusing uncertainties on contrail formation and properties, with inferences for climate science and aviation practices (Schumann et al., 2002). In-flight measurements typically show a single peak in the number-size distribution of freshly emitted soot particles (Schumann et al., 2002). However, some cases reveal a second, larger particle mode in near-field contrails and dry plumes. The source of this larger mode remains uncertain, suggesting that the bimodal distribution might be present in unrefined jet engine emissions rather than being solely caused by contrail processing or coagulation during plume development (Schumann et al., 2002). Precise data on soot particle properties, including any potential bimodal size distribution, is essential for climate models incorporating contrail effects. Resolving the origin of this bimodal distribution is crucial to improving model representations of contrail properties. Policymakers and regulatory bodies rely on scientific research for progressing aviation-related environmental policies, making an understanding of soot particle size distributions significant for up-to-date decision-making.

- Examining the microphysical interaction between soot particles and contrail ice crystals emerges as a critical research priority. Laboratory experiments simulating contrail formation conditions offer valuable insights into the ice nucleating potential of soot particles under controlled environments. Moreover, field measurements focusing on the in-situ characterization of soot particles and ice crystals within nascent contrails provide indispensable data for validating models. By integrating these methodologies, researchers can advance their understanding of the role of soot particles in contrail formation and refine climate models accordingly.

- *Remote Sensing for Contrail-Cirrus*: Organized regional campaigns are important for estimating key variables in aging contrail-cirrus and aircraft plumes. This contains in-situ and remote sensing to portray the growth, decay, and trajectories of contrail ice particles, delivering essential data for interpretation of transformation and radiative effects. Estimation of contrail particles, ambient aerosols, and gaseous aerosol precursors notifies climate modeling and environmental assessments, connecting emissions to contrail properties. Gathered data validates climate models, enhancing their accuracy and improving the representation of contrail-cirrus effects. Understanding contrail-cirrus development supports developing mitigation strategies, informing contrail avoidance, and reducing aviation's climate impact.

In addition to these research issues, there is a need to bridge the gaps with the assessment of the above-reported uncertainties and to investigate and improve the understanding of contrail avoidance and climate tradeoffs. Research should explore how to mitigate the climate impact of contrails without compromising aviation safety and efficiency.

## 7.3 Contrail avoidance and climate tradeoffs

While existing studies explore contrail avoidance strategies and their impact on fuel burn and climate, a comprehensive large-scale evaluation remains elusive. Ideally, such an evaluation would encompass a significant number of routes, account for weather variations, enable full flight level optimization, quantify contrail impacts on individual flights, and compare results to a fuel-optimal baseline to isolate the specific effects of contrail avoidance. This comprehensive approach would provide a more definitive understanding of the fuel-climate tradeoffs associated with contrail avoidance strategies.

Recent studies investigating targeted contrail avoidance strategies offer promising results. Teoh et al. (2020) suggest that focusing on a small percentage of flights with the highest contrail impact could be an effective approach, minimizing the overall increase in fuel consumption and CO2 emissions. Additionally, advancements in engine design hold promise as a long-term solution for reducing contrail formation.

Molloy et al. (2022) address a critical step towards implementing effective contrail avoidance strategies in practice. Their work outlines the practical considerations for conducting large-scale trials, which are essential for evaluating the feasibility and effectiveness of these strategies in real-world conditions. The proposed deployable options based on existing air traffic management processes offer a promising path forward for initiating such trials.

Frias et al. (2024) present a compelling case for the practicality and economic viability of contrail avoidance. Their findings demonstrate that significant reductions in contrail climate impact can be achieved with minimal operational burdens. This study provides strong motivation for airlines and air traffic controllers to embrace contrail avoidance strategies as a means to reduce aviation's overall climate footprint.

Gierens et al. (2008) proposed several potential ways to mitigate the impact of contrails, encompassing both technical and operational approaches. Regarding the technical aspect, they recognized the challenge of completely preventing contrail formation due to its primarily thermodynamic nature. However, they suggested the exploration of new engine cycles and technical infrastructures that might help suppress contrails. Additionally, reducing the number of emitted particles was considered to mitigate contrail formation. While such evaluations would not eliminate contrails completely, they would produce thinner contrails with rarer but larger crystals. These heavier ice crystals would incline more rapidly on average, causing shorter contrail lifetimes. The analysis also investigated operational mitigation options. Striking strict constraints, such as fixed maximum flight levels, was studied unreasonably due to the significant challenges it would pose for air traffic controllers and the resulting safety concerns. Their proposed strategies focused on addressing contrail formation both at the technical level through engine and infrastructure improvements and at the operational level by implementing flexible measures that consider real-time weather conditions. These approaches aimed to strike a balance between contrail reduction and the practical needs of aviation.

In a recent study, Sausen et al. (2023) presented an experiment that aimed to avoid contrail formation during real-world operations. This experiment took place in the Maastricht Upper Area Control region, covering parts of Germany, the Benelux countries, and the North Sea, in the year 2021. The researchers highlighted that contrail avoidance could serve as an effective method for mitigating the climate impact of aviation. To conduct their trial experiment, air traffic was

deliberately diverted every other day by adjusting the flight altitude, either increasing or decreasing
it by up to 2000 ft, whenever potential persistent contrails were predicted. The effectiveness of
these deviations was assessed by analyzing satellite images of high clouds and employing a
contrail detection algorithm that utilized contrail properties. Despite the ongoing challenge of
accurately forecasting persistent contrails, the trial achieved a significant level of success, reaching
97.5%, suggesting that on average, persistent contrails can be avoided during regular flights in the
real world through minor adjustments in the vertical flight path. Contrail avoidance through minor
adjustments in flight paths represents a practical approach to mitigate the formation of persistent
contrails. It demonstrates the feasibility of implementing such measures within air traffic
management to reduce the environmental impact of aviation. The study highlights the potential for
real-world applications of contrail avoidance as a means of addressing the climate effects of
contrails. The findings of this experiment mark an important milestone in implementing
operational measures in air traffic management to decrease the impact of climate from aviation.
The limitations of simply avoiding all contrails become evident when we consider their diverse
climate effects. While some contrails can have a net cooling effect, others contribute to warming.
This distinction necessitates a more nuanced approach to contrail avoidance. We should focus on
mitigating the formation of contrails that contribute to warming, while potentially allowing or even
encouraging the formation of those with a net cooling effect. This distinction requires reliable
methods for differentiating between the two contrail types.
*Future considerations: optimizing contrail avoidance.*
The Sausen et al. (2023) study represents a significant step forward, demonstrating the feasibility
of operational contrail avoidance through minor flight adjustments. To optimize this approach for
maximum climate benefit, future research should focus on three key areas:
*Distinguishing warming from cooling contrails:* Developing robust methods to differentiate
between these contrail types is crucial. Improved weather forecasting models and contrail
prediction algorithms that consider factors like altitude, atmospheric conditions, and cirrus cloud
cover can contribute to this goal.
*Large-scale implementation:* Evaluating the broader applicability of contrail avoidance strategies
necessitates conducting large-scale trials across diverse airspace with varying weather patterns.
This will provide a more comprehensive understanding of the effectiveness and potential trade-
offs associated with contrail avoidance in different operational contexts.
*Integration with air traffic management:* For widespread adoption, efficient protocols for
integrating contrail avoidance measures within existing air traffic management systems are
essential. This may involve developing streamlined decision-making tools and communication
channels for air traffic controllers and pilots.
By addressing these considerations, contrail avoidance can evolve from a general strategy to a
targeted approach that maximizes its potential for mitigating aviation's climate impact. This
approach would focus on reducing the formation of warming contrails while allowing for or even
encouraging the formation of cooling contrails.
In a recent Ph.D. thesis, Elmourad (2023) evaluated the fuel-climate tradeoffs arising from contrail
avoidance strategies applied on a large scale. This study raises several key questions about
contrails avoidance: What fraction of aviation's global total contrail length can be avoided using
vertical re-routing exclusively? What is the fuel penalty of such an avoidance strategy? How does

the fuel penalty or contrail reduction vary between flights or seasons? Can the fuel penalty be constrained? What effect does that have on the ability to do contrail avoidance? What net climate benefit can be achieved from contrail avoidance strategies? How does this differ from avoiding all contrails or only nighttime contrails? What are the relative orders of magnitude between the climate benefits from contrail avoidance and the climate damages from additional fuel burn? How does this difference compare with the uncertainties associated with contrail impacts?

As the global volume of air traffic continues to rise, the aviation industry grapples with a significant challenge in mitigating its environmental impact on climate change. Contrails, which contribute to global warming by trapping terrestrial radiation, have the potential to counteract the benefits of reduced emissions resulting from optimized flight paths. In a study by Roosenbrand et al. (2023), the authors conducted a global assessment of flights that contribute to contrail formation and assessed the altitude adjustments required to avoid these contrail-prone areas. This analysis utilized a combination of data from the Integrated Global Radiosonde Archive (IGRA), which provides measurements from weather balloons with global coverage and high vertical resolution, and flight data from OpenSky. The study identified Mid-Western Europe, the Eastern United States of America, and Japan as regions characterized by both high air traffic volumes and a substantial percentage of flights forming contrails. Importantly, these regions offer opportunities for altitude adjustments of less than one kilometer to minimize contrail formation. The research also pinpointed other regions where relatively minor operational interventions could yield significant climate benefits.

*Fuel-climate tradeoffs*: While there have been numerous studies on operational contrail avoidance, there are still some research gaps in the field when it comes to understanding the fuel-climate tradeoffs that would follow a large-scale global application of contrail avoidance strategies.

*Sustainable Aviation Fuel (SAF)*: Much still needs to be understood about what SAF will look like as it becomes the dominant fuel of the future. The emissions of soot and other particles from SAF are still not well understood and will have a significant effect on future contrail production and lifetimes. According to Bräuer et al. (2021b), sustainable aviation fuels have been identified as capable of reducing both contrail ice numbers and the radiative forcing caused by contrail cirrus. Their study involved the measurement of apparent ice emission indices across different fuels with varying aromatic content at altitudes ranging from 9.1 to 9.8 km and 11.4 to 11.6 km. The data were collected during the ECLIF II/NDMAX flight experiment in January 2018, encompassing a variety of fuels differing in aromatic quantity and type. A comparison between a sustainable aviation fuel blend and a reference fuel Jet A-1 revealed a maximum reduction of 40% in apparent ice emission indices, highlighting the potential impact of sustainable aviation fuels on mitigating contrail-related environmental effects.

Moore et al. (2017) examined the impact of biofuel blends compared to traditional jet fuel on aircraft emissions. They found that biofuel blends significantly reduced the quantity and weight of particles emitted, suggesting biofuels can reduce aviation's environmental impact. Their measurements showed a notable decrease in total and non-volatile particles with the biofuel blend and a shift to smaller particle sizes. This is due to a larger reduction in bigger soot particles acting as condensation nuclei. The drop in soot emissions enhances the formation of new particles. Mixing petroleum fuels with HEFA bio-jet fuel cut volatile and non-volatile particle emissions by 50%-70% during cruise conditions. However, soot particle emissions remain high, indicating biofuels help but don't eliminate issues related to contrail formation.

*Alternative aircraft and new fuels*: There is a lot of ongoing discussion about the potential development of aircraft using hydrogen as fuel. Such an aircraft engine would still emit water vapor and therefore could produce contrails. However, the major reduction in particulates, e.g., no soot, could affect the lifetime of the contrail and whether it could result in contrail cirrus. Research is needed to evaluate the potential for contrail production and resulting lifetimes. Other alternative fuels would also need to be evaluated.

There is a necessity for government research organizations as well as industry partners to venture into the investigation and establishment of sensor prototypes. An example is a humidity sensor prototype with a range of great precision covering 20 parts per million by volume to more than 10,000 ppmv. The purpose is to improve the accuracy of measurements, especially on parameters like temperature and relative humidity, at altitudes preserved during flight to counteract contrail mitigation. Additionally, improving measurement at lower altitudes is vital for improving the precision of weather forecasting. This accumulated data will play a pivotal role in evaluating and integrating these insights into forecast models, enhancing the robustness of these calculations. An additional area of focus is the refinement of model calculations on the impact of contrails, necessitating improved handling of variables such as clouds, aerosols, and specific aircraft characteristics.

Tackling the pressing challenge of climate change requires the exploration and adoption of effective contrail mitigation strategies. Recent studies show promise in the utilization of alternative fuels, particularly sustainable aviation fuels (SAFs), which boast lower soot particle emissions compared to conventional jet fuel. Initial flight trials have shown a marked decrease in contrail formation when SAFs are used, potentially attributed to the reduced concentration of ice nucleation particles in SAF exhaust. Moreover, ongoing research focuses on optimizing flight path planning and air traffic management (ATM) to minimize contrail formation and persistence. Research conducted by Sausen et al. (2023) demonstrates the feasibility of contrail avoidance through minor adjustments in flight paths during real-world operations. Continued progress and application of these strategies, coupled with advancements in contrail observation technologies such as high-precision humidity sensors, offer significant potential to mitigate the climate impact of aviation.

## 8 Conclusions

This study reviews the current understanding of the impacts of contrail formation by aircraft on the Earth's climate system; it provides insights into the current state of contrail research, offering perspectives on contrail formation, characteristics, life cycle, and potential future impacts. The study underscores the importance of confronting uncertainties in climate models and emphasizes the need for further research to enhance our understanding of contrail effects.

Aviation emissions, encompassing both contrail cirrus clouds and carbon dioxide emissions, exert an important influence on the Earth's climate. Extensive research efforts have significantly improved our understanding of contrail formation and aging; however, key microphysical processes remain a subject of ongoing investigation. Our examination of global observational data and projections highlights the potential for a significant increase in contrail cirrus radiative forcing by the mid-21st century, driven by factors such as anticipated growth in air traffic, potential improvements in fuel efficiency, and evolving atmospheric conditions. While uncertainties regarding the precise magnitude of this increase exist, the potential consequences necessitate proactive mitigation strategies. Several promising approaches are being explored, including operational measures to optimize flight paths, the development of new contrail suppression

technologies, and the adoption of sustainable aviation fuels. Implementing a combination of these strategies, while continuously refining our understanding of contrail effects, is crucial for minimizing the overall climate impact of aviation.

**Author contributions.** DKS prepared the manuscript with partial support from SS and supervision from DJW

**Competing interests.** The author declares that they have no competing interests.

**Acknowledgment.**
This work was supported by the National Aeronautics and Space Administration (NASA) under award number NNA16BD14C for NASA Academic Mission Services (NAMS).
We thank Drs. Andrew Gettelman (Pacific Northwest National Laboratory, PNNL, USA), Ulrike Burkhardt (German Aerospace Center, DLR, Germany), Bernd Kärcher (DLR), Phillip J Ansell (University of Illinois at Urbana-Champaign, USA), D.S. Lee (Manchester Metropolitan University, UK), and Steven L Baughcum (The Boeing Company) for reviewing prior drafts of this report.
**Financial support**. The authors would like to thank for support from the Universities Space Research Association (USRA) through project subcontract 08600-031.

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
