# Peer review of "Understanding the Role of Contrails and Contrail Cirrus in Climate Change: A Global Perspective"

_EGUsphere, 2024_

## Referee Comment (RC1)

**Review of "Understanding the Role of Contrails and Contrail Cirrus in Climate Change: A Global Perspective" by Singh et al. (2024)**

**Overview**

The manuscript summarises the microphysical processes concerning contrail formation, ageing and transition into cirrus clouds. The current understanding of the role of aviation soot in affecting contrail formation and cirrus activation is reviewed, from all three perspectives of in situ measurements, remote observations, and modelling. Recently gained knowledge of contrail cirrus occurrence from filed observations is highlighted in the paper. The article emphasised two key impactors that causes the high uncertainties in estimating the climate impact of contrail cirrus, namely contrail coverage and soot indirect effect. Available contrail observation datasets models for contrail property and evolution studies, and for understanding the climate impact of contrail now and future are listed in the manuscript. The uncertainties and research gaps in contrail cirrus and its radiative forcing are identified by the authors. Research needs for a better understanding of the impact of contrail cirrus are pointed out in the manuscript to bridge the knowledge gaps. Finally, the manuscript looks into the current status of contrail avoidance and climate trade-offs from the aspects of slightly deviating the aircraft from contrail forming regions and using climate-friendly aviation fuels.

The manuscript is well written with a clear structure and is easy to follow. However, to have a comprehensive understanding of the role of contrails and contrail cirrus in climate change and the status of the knowledge, there are some more aspects that the authors should elaborate.

**Major comments:**

1. The authors detailed the different processes evolved in contrail formation, evolution and transition into cirrus clouds. It is of high importance to understand these processes, which also lay the foundation for the modelling of contrails and contrail cirrus's radiative impact. However, the radiative impact of contrail cirrus (including linear contrails) is closely associated with optical properties and the coverage of contrail cirrus, in which the age of contrail cirrus plays an important role. As this is a review article, it would be good to provide an overview (a summary table or view graph) of contrail characteristics (size, number density, extinction properties, width, length, optical depth...) at various stages of contrail formation and evolution, namely jet, vortex, dissipation, and diffusion regimes, respectively, especially in the diffusion stage, during which the radiative impact is the most critical, from the perspectives of both measurement and modelling.

2. In Sect. 3, the authors introduced difference models for contrail properties and evolution in individual and overlapped contrails cases. Microphysical processes implemented in the models are described briefly. The treatment of contrails in global models is also reviewed. The Hadley Centre climate model HadGEM2 from the UK Met Office has also been implemented with contrail parameterization for evaluate contrail's radiative forcing.

   Rap, A., P. M. Forster, A. Jones, O. Boucher, J. M. Haywood, N. Bellouin, and R. R. De Leon (2010), Parameterization of contrails in the UK Met Office Climate Model, *J. Geophys. Res.*, 115, D10205, doi:10.1029/2009JD012443

I suggest the authors add a cross-model intercomparison and assessment table, especially for the multiple models for simulating contrail properties and lifespan and evaluate the models against observations. From this, the readers will get a comprehensive overview of the performance of current available contrail models. It will also outline what is need in the future for model development from the perspective of properly characterise contrails and contrail cirrus.

3. The same for Sect. 4. The authors reported independently the results from various climate impact models and future RF projections. The agreement and disagreement between models on the climate impact of contrail cirrus now and in the future are not discussed. And if there are any disagreements, what are potential causes for the models to behave differently.

4. (1) In Sect. 5, several observational datasets that are directly linked to contrail and detection, such as air traffic data, engine emission indices, in-situ contrail observations and satellite image libraries, are included. However, it is also highly important to include datasets with critical parameters for contrail formation, such as temperature and relative humidity with respect to ice in the upper troposphere and the tropopause region. One of such datasets is provided by the In-service Aircraft for a Global Observing System (IAGOS) research infrastructure with high spatial and temporal resolution, which has already been used for contrail occurrence studies and assessment of forecast models, e.g., Gierens et al., 2020 (doi:10.3390/aerospace7120169).

(2) Regarding the satellite-based contrail dataset, the human-label contrail datasets, the AI detection and machine learning algorithms are helpful in monitoring contrails, perhaps even for contrail lifetime and coverage studies. The case studies mentioned by the authors are mostly based on contrails with clear line shapes. However, these datasets are rather small and are limited to certain regions, camera angels and quantities and spatial resolution. I am not convinced it is of great help to improve the estimation of contrail's radiative forcing which are closely associated with contrail ice particle microphysical and optical properties. To achieve this, contrail optical thickness should be retrieved, which are not well resolved by satellite images and rely on robust retrieval algorithms. Furthermore, satellites cannot detect contrails until tens of minutes later, and overlapped contrails and embedded natural cirrus add extra challenges to contrail detection and parameterization for satellite observations. Finally, ice supersaturated regions in the UTLS, where contrails prefer to occur, exhibit strong variability in locations, time and scales. From all above, I think the authors should be prudent in judging the capability of satellite contrail datasets and AI detection algorithms to support contrail mitigation and robust estimation of contrails' climate impact.

5. Sect. 7 Microphysical processes of contrail formation in relation to the role of soot/background aerosols and heterogeneous/homogeneous freezing mechanism also poses big uncertainties in understanding the effects of contrails. Though it is known that fresh contrail contains thousands of ice particles, it is still debated on the ice nucleating properties of soot particles for contrail formation, see e.g.,

Righi et al., Exploring the uncertainties in the aviation soot–cirrus effect, Atmos. Chem. Phys., 21, 17267–17289, https://doi.org/10.5194/acp-21-17267-2021, 2021

Testa et al., Soot aerosol from commercial aviation engines are poor ice nucleating particles at cirrus cloud temperatures, EGUsphere [preprint], ttps://doi.org/10.5194/egusphere-2023-2441, 2023

Testa et al., Contrail processed aviation soot aerosol are poor ice nucleating particles at cirrus temperatures, EGUsphere [preprint], https://doi.org/10.5194/egusphere-2024-151, 2024

Contrail-cirrus observation, either with remote-sensing or in-situ measuring technique, has high requirements for instrumentation, facilities, and platforms, technically and financially. In-situ instruments provide high temporal and resolution data but must meet the harsh conditions of cold temperatures, vibrations and limited dimensions for airborne measurements. The measurement of soot cores at extremely cold temperatures ~60°C is challenged so far with ice nucleating particle counter.  To integrate a complete set of high precision instruments for contrail cirrus properties is also demanding. Remote-sensing techniques are vulnerable to the blender of natural and contrail cirrus and should be provide high enough temporal and spatial resolution to track the width, length, depth of the contrails, etc.

Further research is also needed to address these uncertainties and gaps.

6. Water vapour observation and humidity sensors. Dense, wide-coverage and high-resolution temperature, water vapour and relative humidity with respective to ice (RHi) are essential for studying and predicting contrail occurrences and properties. In particular, these data should be provided with high accuracy and precision. They are also valuable for evaluating weather and contrail forecast models. The IAGOS infrastructure provides quality checked temperature and RHi data obtained from routine passenger aircraft measurement since 1994, which have been used in various studies. The IAGOS capacitive hygrometer is the same prototype of Humicap sensor from Vaisala that are widely used in weather balloon sondes. It is one of the three operational global water vapour observation programmes as documented in the WMO report on supporting water vapour measurements within an aircraft meteorological data relay programme. The other two technology types for aircraft-base water vapour measurement are the Tropospheric Airborne Meteorological Data Reporting (TAMDAR) sensor system and the Water Vapor Sensing System II (WVSS-II) developed by FLYHT, which provide data to meteorological centres. Rolf et al. (2023) has conducted an evaluation of compact hygrometers for continuous airborne measurements in the range of 20 to 20,000 ppmv.

   Rolf et al., Evaluation of compact hygrometers for continuous airborne measurements, Meteorol. Z. 2023 Vol. 20 Issue 20, DOI: 10.1127/metz/2023/1187

   Basically, the technology for airborne water vapour and humidity sensor are well established. What matters more is the quality of the data, maintenance of the programmes and the joint between datasets and programmes. How to incorporate the programmes and networks to facilitate contrail observation, reconcile observations with contrail forecasting, and conduct safe contrail mitigation are still missing and should address in future research.

7. In the conclusion, the author should elaborate what is the well-established knowledge and where are the research difficulties and gaps that careful investigation and dedicated research is required.

**Minor comments:**

Some of the figures and tables are not well clarified in short captions with abbreviations, e.g., Figure 7 and Table 2, Table 4, etc. The colour bar scales and ticks are not readable, e.g., Figure 13.

Line 55: Why not just use "Aviation-induced cirrus" instead of "Contrails and resulting contrail clouds formation"?

Line 78: "Many global climate chemistry models used to study … do not incorporate the impacts Models that do account for contrails…": What impacts does the author mean, e.g., contrail cirrus characteristics, contrail parameterization or so?

Line 136: "optical characteristics and lifespan of contrails are significant dynamics of contrail formation and…": "dynamic" is usually referred to the movement of air parcels. Does the author mean "optical characteristics and lifespan are key aspects of contrail formation"?

Line 232: Previous studies have studied the occurrence frequency distributions of ice-supersaturated regions (ISSRs) with high temporal-resolution aircraft measurements. The variabilities of ISSRs in the upper troposphere and in different regions with high air traffic density have also been explored using aircraft and weather forecast reanalysis data, such as:

> Petzold et al., Ice-supersaturated air masses in the northern mid-latitudes from regular in situ observations by passenger aircraft: vertical distribution, seasonality and tropospheric fingerprint Atmos, Chem. Phys. 2020 Vol. 20 Issue 13 Pages 8157-8179

> Reutter et al., Ice supersaturated regions: properties and validation of ERA-Interim reanalysis with IAGOS in situ water vapour measurements, Atmos. Chem. Phys. 2020 Vol. 20 Issue 2 Pages 787-804

Line 242-246: The two sentences repeat each other.

Line 271: The impact of soot on contrails and soot was discussed based on mostly old literature. There were new studies appearing in the past years, e.g., Testa et al. (2023 and 2024). The authors should sample more recent studies to provide the current understanding in this topic.

Line 280: The authors described the cirrus observation near Munich in 2012 and traced soot emissions related to the cirrus formation with back trajectories. Can the authors provide the source of the case study? Or give more details on the observation and back-trajectory calculations?

Line 334: The authors cited Wolf et al. (2023) for their exploration of the dependence of contrail cirrus RE on different parameters. Can the authors refine the main messages of Wolf et al and provide readers an overview on the impact of these parameters or conclude which parameter have the greatest impact on cirrus RF?

Line 357: In this paragraph, the authors discussed dynamics and ice crystal evolution in the wake within the regimes of vortex dissipation and diffusion. Figure 6 is used here to illustrate the contrail formation processes in aircraft vortex. However, the processes in Figure 6 are happening within ~100 sec upon the

emission of aircraft exhaust, namely the het and vortex regimes in Figure 5. It fits more in the previous paragraph.

Line 446: "In exhaust rich in soot, the number of soot particles… increases with decreasing ambient temperature. This, in turn, raises plume cooling rates and levels of plume supersaturation over liquid water." The causal relationship is reversed. The increased cooling rates and supersaturation is caused by decreasing temperature in the plume, not by the increasing of soot particles. The freezing of water vapour on particles would actually release heat and consume the supersaturation in plume air masses.

Line 462: in the context of contrails formation → in the context of contrail formation

Line 526: Sect. 2.7. The authors sampled recent studies reporting the observations of contrails and contrail cirrus in slightly ice-subsaturated environments, which is new knowledge of contrail formation and evolution. However, as this subsection is about lessons learned from observations of contrails and their properties, the authors may also consider from measurement perspective the limiting factors of contrail property observations.

Line 569: Mahnke et al. (2022) was cited, but it didn't appear in the reference list.

Line 638: In this study → In this model

Line 671 and Line 679: The order of the two paragraphs can be reversed for readability.

Line 719: This sentence repeats the previous one.

Sect. 3.2.2: Paragraph 2-4. Regarding the development of the contrail cirrus parameterization in the mode ECHAM5-HAM, the content jumps back and forth. Following a chronological order of the citations may show the improvement of the parameterizations in the model better. Paragraph 3 and 4 can be combined by saying "Bock and Burkhardt (2016) extended the contrail cirrus parameterization (CCMod) initially proposed by Burkhardt and Kärcher (2009) by introducing contrail cirrus ice crystal number concentration and volume as additional prognostic variables.".

Line 990: I guess the authors mean that the maximum flight density will occur at a higher, rather than a lower, altitude around 200 hPa in2050 compared to 240 hPa in 2006, if flight altitudes are expected to rise by 0.3 to 1.5 km. From Table3, readers cannot recognise which row represents the projection resulted from altitude increase. The authors should explain shortly the different simulated scenarios. "They found that the maximum flight density would occur at a lower attitude in 2050" is redundant (Line 991).

Line 1016: mWm-2 -> mW m$^{-2}$, the same elsewhere in the context.

Line 1037: With the values listed in the table, a 3-4-fold increase is expected in central Europe, and over 400% increase in North America. For readability, the authors should note shortly what stands for RCP4.5, RCP 8.5, SC1, 2 and 3.

Line 1133: The Schmidt-Appleman criterion is commonly cited as Schmidt, 1941, Appleman,1953 and Schumann, 1996.

Line1413: "rarer but more ice crystals" -> rarer but larger crystals

Line 1470: The paper by Moor et al. (2017) reported soot reduction at cruises with adequate measurements and should be added here.

Line 1472: "Our study involved the measurement …" -> Their study…

**Formatting issues:**

1. Line 110: The formatting of subsection title is not in coherence with the other subsections. There is a mixture of British and American English, e.g., "Ageing" and "Aging" (Line 106).

2. Punctuation. Period is missing, e.g., Line 78, Line 1041, or Line 1120 repeated period, Line 570 mistake, and Line 491 with uppercase after a comma).

3. Citations. Author names are missing in Line 827 – 828. Citation formatting error, e.g., Line 1140: Minnis et al. (1998 and Minnis et al. 2013).

4. References. The references are not in the same format. Some have a DOI, while many other don't. A DOI should be provided to each reference. The font style and size are not the same through the list.

---

## Referee Comment (RC2)

Anonymous Review

April 18, 2024

**Understanding the Role of Contrails and Contrail Cirrus in Climate Change: A**
**Global Perspective**

by Dharmendra Kumar Singh, Swarnali Sanyal, Donald J. Wuebbles

The topic of the paper is of high actual interest and well suited for ACP.

As stated in lines 100 ff, the paper aims to provide a comprehensive review of our
understating of contrails… and to address the uncertainties….

Such a comprehensive review does not exist in the literature. It seems that the not all
authors have extensive experiences in contrail research by themselves so many of their
comments may raise critical discussion. The paper reads like a literature study in
preparation of more specific own research by the authors on this topic.

But different from reviews by acknowledged experts, who like their own work most, this
paper is rather unbiased in the selection of materials and assessments, and this is
refreshing.

The coverage is quite good but can be improved. Here I have several additions and offer
comments on detail, see below. The paper is (of course) mostly a  collection of existing
knowledge. The conclusions are not really surprisingly new. I found the comments on
Machine Learning aspects particularly interesting (I think I should try to learn more about
them). Section 7, "Research needs and gaps " is worthwhile to have though not
objective in all parts and partly hard to validate because of incomplete explanations or
citations. I think it is worthwhile to have estimated numbers on the magnitudes of
uncertainty but presently these number are not well justified.

The review misses many important publications. That should be improved.

The paper is partly too long to keep me interested in all details, collects lot of materials.
Still the paper is worth to read but it needs amendments before it can be published.

Such a long paper is hard to review, and I can give only a few specific comments. I
would formulate many parts differently.

Abstract.

Line 12: The sentence "contrail cirrus enhances the impact of natural clouds on the
climate" is misleading, because I read it first as if contrails would have no effect without natural cirrus. But that is not the case. Better: Contrail cirrus is similar to natural cirrus in
having an impact on climate.

Then I would start a new sentence on the uncertainties – without "although" or "however"
or "but". Simply say that this paper discusses existing uncertainties

The statement "there are still unresolved questions" is trivial. There will be always
unresolved questions. In fact, with further research more and new questions may likely
arise. So, you should condition this sentence a bit more, like "presently the state of the
art is insufficient to assess the magnitude of the climate impact of contrails, e.g. on
surface temperature, and its error bounds with high confidence" – or similar.

Line 16: add "mixing" as one of the issues.

Line 56: this was known basically since long (see IPCC, 1999).  Other reviews of
relevance are, e.g. (Schumann, 1994; Sassen, 1997; Brasseur et al., 1998; Fahey and
Schumann, 1999; Prather and Sausen, 1999; Sausen et al., 2005; Brasseur et al., 2016)

Line 68: what do you mean with atmospheric warming. It is correct for RF. But the ERF
relates RF to the global mean surface temperature change.

In fact, perhaps you can discuss how reasonable it is to assess the impact of contrails in
terms of global metrics. Contrails are local events, are short lived and have certainly a
very small global mean surface temperature climate impact. Contrails warm mainly the
upper troposphere, not so much the surface. Most contrails occur over continents with
little chance or long-lives effects by heating the oceans.  The regional effects are likely
far larger than global effects and should be easier to be detected and quantified. And
regional effects are of importance for the densely populated latitudes in the Northern
hemisphere where most air traffic occurs.

Line 78: sentence(s) incomplete and unclear.

Line 81 is biased to one team and one model type only. Other should be included.

Line 85: there is far more literature on this aspect.

Fig 1 caption. "Current best estimate.." The estimate is already no longer the last one.
Several studies have added new estimates. I would suggest:  "Climate forcing estimate
as published in 2021" or similar.

Fig 2. misses to mention the importance of mixing of the jet exhaust gases with ambient
air. Mixing controls early or late contrail formation and contrail lifetime (Lewellen and
Lewellen, 1996; Gerz and Ehret, 1997; Sussmann and Gierens, 1999; Lewellen, 2014;
Paoli and Shariff, 2016; Schumann and Heymsfield, 2017).

Line 152. These 2 references do not cover this topic sufficiently. There are many more. I
see that you cite them later. But this line is irritating.  See line 178. See e.g. Sassen
(1997).

Line 197. Further new reference: (Märkl et al., 2024)   .

Line 246, Contrail formation was observed far earlier. See (aufm Kampe, 1942; 1943;
Weickmann, 1945; Brewer, 1946; Schumann and Wendling, 1990; Schumann, 1994;
Busen and Schumann, 1995).

Line 256. The visible constraint were introduced earlier: Appleman (1953).

Line268 and Fig 4: a similar figure with a contrail spiral was given earlier in Fig. 11.2
(Schumann, 2002).

Impressive contrail observations were also presented by Laken et al. (2012) and
(Mannstein and Schumann, 2005).

Line 277: it is now pretty clear that the soot acts as condensation nucleus so that liquid
droplets form which then freeze. The resultant ice particles likely do include the soot
particles on which the water condensed. This does not exclude that some ice particle
sublimate or that some soot remains dry throughout the contrail formation process, so
that also dry soot is found inside contrails. See upcoming paper by Dischl et al.
(Egusphere, to be published soon).

Lines 279 to 288 has been taken from Schumann and Heymsfield (2017), but the
reference  is missing.

Line 446: How do we know the current ranges of in-flight soot emission. Here you should
cite papers by the team around Marc Stettler and Roger Teoh and colleagues (Teoh et
al., 2019; Teoh et al., 2020a; Teoh et al., 2020b; Teoh et al., 2022a; Teoh et al., 2022b;
Teoh et al., 2024a; Teoh et al., 2024b).

Lines 462 to 483: For the discussion of the impact of the initial contrail ice particle
number concentration (and hence the soot emissions) on contrails the cross-section
integrated optical extinction and hence on climate forcing you should cite details of the study and explanation around Eq. (5) in  Lewellen (2014). See also Section [17] in (Schumann et al., 2013).

Page 9: The summaries of Urbanek, Kärcher and Wolf read like abstracts from these papers. The review does not really combine these result critically with related results from other studies.  This is a serious deficiency of this review and makes me uncertain whether this is an acceptable review.

Lines 516 ff: Sedimentation. This section misses citations of related literature. See, e.g., Spichtinger and Gierens (2009)

Line 563 : 290 m/s, That is nearly speed of sound. I doubt that this value is correct (230 m/s would sound more reasonable).

Lines 574 to 588: difficult to read. I have not understood what you want to say.

Line 606, replace 2016 by 2017.

Lines 617: the LES is of course far more expensive than a plume model and hard to apply globally. We need improved plume models, e.g., based on LES. By the way: Much work in this respect has been done by Unterstrasser (2016).

Line 732. The comparisons have been done in several follow-up papers of that cited here.

Line 780: You refer the age of 450 s without comments. To me 450 s is far too long.

Line 83: neither 10 nor 7.5 um is correct. That value varies by large factors.

Line 528 see earlier work, e.g., Sassen (1997).

Line 876: This was first discussed by Meerkötter et al. (1999).

Line 1122, Overlap effects. See (Schumann et al., 2021; Teoh et al., 2024a)

Line 1130. Ettenreich observed in 1915, but published in   1919.

Line 1153: Which reference do you refer to with Schumann, 2016?

Line 1158: See the GAIA data set (Teoh et al., 2024b).

Line 1163. See also the development of open access performance codes: (Poll, 2018; Poll and Schumann, 2022; 2024).

Line 1252: Limited literature exists". See, e.g., measurements of ice particles by the
Geophysica aircraft at about 20 km height above tropical convection by measuring its
own contrail  (Schumann et al., 2017).

The paper by Myhre (2009) is certainly outdated There are several recent studies citing
this paper.

Line 1299 How do you come to the value 55 %? It seems too large to me.

Linem1346: Contrail effects on climate will never be "fully" understood.

But also many other things are incompletely understood and nevertheless practically
addressed. (See, e.g., smog in cities, observed in the 1950's and soon thereafter
regulated by US federal institutions (was it ERL, I forgot the name) and later by ICAO.

Line 1359 Again: ERF is an insufficient climate metric because it does not account for
regional (northern mid-latitudes) phenomena  and other climate changes besides mean
surface temperature.

Line 1503: As said before "Uncertainties persist"´. I must tell you: uncertainties will be
there also in 100 years from now.  The existence of uncertainties is not a sufficient
argument to start mitigation actions soon. We only have to make sure that the actions
reduce and not increase the climate effect. (It is like building a bridge over a river. The
only thing which counts is that the likelihood that the bridge fails is sufficiently small to be
acceptable. That problem was solved at least 2000 years before today, in spite of very
large uncertainties. I suggest to carefully rethink about talking about uncertainties.

Chapter 7 "Uncertainties and Research Gaps" is certainly an important part of the paper.

Here many questions arise.

You provide number of the uncertainties in percentages. How are these number
derived? That is unclear in most case. Please explain. I assume you will note: there are
not only uncertainties but also uncertainties of how large the uncertainties are.

The quotes percentage values add (linearly) up to a total of more than 100 %. Please
discuss.

Line 1284: The reference to Myhre et al. (2009) in assessing radiative transfer schemes
is outdated. There are several new discussions. The key parameter is the shortwave to
longwave ratio in contrail radiative forcing. E.g. Newinger and Burkhardt (2012)  used an
outdated Radiation scheme which strongly underestimated the SW/LW ratio. See
discussion in   Schumann and Graf (2013) and the important poster publication  of
Ponater et al. (2013).

Line 1287: add Forster et al. (2012).

Line 1311: Why is the effect of lifetime of lesser importance? I would expect it is very important (Lewellen, 2014)

Line 1327: add: well assimilated observation are important to setup good initial conditions for accurate weather predictions (Bauer et al., 2015).

Line 1359: ERF is not a measure for climate change in general. It is a measure of the RF impact on global mean surface temperature change only.

The bi-model size distribution of soot??? I am not aware of this. Please provide a reference. See Fig. 4 in (Schumann et al., 2002).

Line 1402: The word "fully" excludes everything, but see (Teoh et al., 2020a; Teoh et al., 2020b; Molloy et al., 2022; Teoh et al., 2022a; Martin Frias et al., 2024; Teoh et al., 2024a).

Line 1413: replace "more" by "larger"

Please note the following comment on Sausen and similar studies. It makes no sense to avoid formation of all contrails. That would imply that you also avoid cooling contrails and it overburdens air traffic management. It is sufficient and far more effective to avoid the warming contrails and to allow for (even more) cooling contrails. I gave several references for this above, e.g., (Molloy et al., 2022; Martin Frias et al., 2024).

Line 1498: Please delete the word "valuable". Such an applause should come from the readers not from the authors.

Line 1505. "Uncertainties persist" that has been said often enough and does not get more correct by repeating.

Line 1519: replace Berndt by Bernd.

Lien1524 replace "the" by "for "

In looking to the list of references:

Please consider the important work of (Bickel et al., 2020; Ponater et al., 2021; Bickel, 2023)

Also (Duda et al., 2023).

Very important: (Iwabuchi et al., 2012)

and on ice supersaturation: (Ovarlez et al., 2000; Jensen et al., 2001; Ovarlez et al., 2002)

Line 1732 and elsewhere: Please list all co-authors.

Line 1916: What does the single "S" mean?

Line 1949: please set in capitals (here and elsewhere): MSG and SEVIRI

Line 1854 The method's name is RRUMS not drums.

Overall:  A carefully revised version should become interesting, worthy to read, and
acceptable.

References:

Appleman, H. (1953). The formation of exhaust contrails by jet aircraft. *Bull. Amer.*
*Meteorol. Soc.*, *34*, 14-20.
aufm Kampe, H. J. (1942). Kondensation und Sublimation in der oberen
Troposphäre*Rep. Deutsche Luftfahrtforschung, Deutsche Forschungsanstalt für*
*Segelflug, FB 1491, http://elib.dlr.de/107959/*, 60 pp.
aufm Kampe, H. J. (1943). Die Physik der Auspuffwolken hinter Flugzeugen. *Luftwissen*,
*10*, 171-173, available at https://ntrl.ntis.gov and TIB Hannover, Germany.
Bauer, P., A. Thorpe, and G. Brunet (2015). The quiet revolution of numerical weather
prediction. *Nature*, *525*(7567), 47-55, doi: 10.1038/nature14956.
Bickel, M., M. Ponater, L. Bock, U. Burkhardt, and S. Reineke (2020). Estimating the
effective radiative forcing of contrail cirrus. *J. Clim.*, *33*, 1991-2005, doi:
10.1175/JCLI-D-19-0467.1.
Bickel, M. (2023). Climate Impact of Contrail Cirrus*Rep.*, 133 pp, DLR and LMU Munich,
DLR FB 2023-14.
Brasseur, G. P., R. A. Cox, D. Hauglustaine, I. Isaksen, J. Lelieveld, D. H. Lister, R.
Sausen, U. Schumann, A. Wahner, and P. Wiesen (1998). European scientific
assessment of the atmospheric effects of aircraft emissions. *Atmos. Env.*, *32*(13),
2329 - 2418.
Brasseur, G. P., M. Gupta, B. E. Anderson, S. Balasubramanian, S. Barrett, D. Duda, G.
Fleming, P. M. Forster, J. Fuglestvedt, A. Gettelman, R. N. Halthore, S. D. Jacob,
M. C. Jacobson, A. Khodayari, K.-N. Liou, M. T. Lund, R. C. Miake-Lye, P.
Minnis, S. C. Olsen, J. E. Penner, R. Prinn, U. Schumann, H. B. Selkirk, A.
Sokolov, N. Unger, P. Wolfe, H.-W. Wong, D. W. Wuebbles, B. Yi, P. Yang, and
C. Zhou (2016). Impact of aviation on climate: FAA's Aviation Climate Change
Research Initiative (ACCRI) Phase II. *Bull. Amer. Meteorol. Soc.*, *97*(4), 561-583,
doi: 10.1175/BAMS-D-13-00089.1.
Brewer, A. W. (1946). Condensation trails. *Weather*, *1*(2), 34-40, doi: 10.1002/j.1477-
8696.
Busen, R., and U. Schumann (1995). Visible contrail formation from fuels with different
sulfur contents. *Geophys. Res. Lett.*, *22*(11), 1357-1360, doi:
10.1029/95GL01312.
Duda, D. P., W. L. Smith Jr., S. Bedka, D. Spangenberg, T. Chee, and P. Minnis (2023).
Impact of COVID-19-related air traffic reductions on the coverage and radiative effects of linear persistent contrails over conterminous United States and surrounding oceanic routes. *J. Geophys. Res.*, *128*, e2022JD037554, https://doi.org/10.1029/2022JD037554.

Fahey, D. W., and U. Schumann (1999). Aviation-Produced Aerosols and Cloudiness. in *Aviation and the Global Atmosphere. A Special Report of IPCC Working Groups I and III*, edited by J. E. Penner, D. H. Lister, D. J. Griggs, D. J. Dokken and M. McFarland, pp. 65-120, Cambridge University Press, New York.

Forster, L., C. Emde, S. Unterstrasser, and B. Mayer (2012). Effects of three-dimensional photon transport on the radiative forcing of realistic contrails. *J. Atmos. Sci.*, *69*(July), 2243-2255, doi: 10.1175/JAS-D-11-0206.1.

Gerz, T., and T. Ehret (1997). Wingtip vortices and exhaust jets during the jet regime of aircraft wakes. *Aerosp. Sci. Techn.*, *1*(7), 463-474.

Iwabuchi, H., P. Yang, K. N. Liou, and P. Minnis (2012). Physical and optical properties of persistent contrails: Climatology and interpretation. *J. Geophys. Res.*, *117*, D06215, doi: 10.1029/2011JD017020.

Jensen, E. J., O. B. Toon, S. A. Vay, J. Ovarlez, R. May, P. Bui, C. H. Twohy, B. Gandrud, R. F. Pueschel, and U. Schumann (2001). Prevalence of ice-supersaturated regions in the upper troposphere: Implications for optically thin ice cloud formation. *J. Geophys. Res.*, *106*, 17253-17266, doi: 10.1029/2000JD900526.

Laken, B. A., E. Palla, D. R. Kniveton, C. J. R. Williams, and D. A. Kilham (2012). Contrails developed under frontal influences of the North Atlantic. *J. Geophys. Res.*, *117*(11), D11201, doi:  10.1029/2011JD017019.

Lewellen, D. C., and W. S. Lewellen (1996). Large-eddy simulations of the vortex-pair breakup in aircraft wakes. *AIAA J.*, *34*, 2337-2345.

Lewellen, D. C. (2014). Persistent contrails and contrail cirrus. Part II: Full lifetime behavior. *J. Atmos. Sci.*, *71*, 4420-4438, doi: 10.1175/JAS-D-13-0317.1.

Mannstein, H., and U. Schumann (2005). Aircraft induced contrail cirrus over Europe. *Meteorol. Z.*, *14*(4), 549 - 554, 10.1127/0941-2948/2005/0058.

Märkl, R. S., C. Voigt, D. Sauer, R. K. Dischl, S. Kaufmann, T. Harlaß, V. Hahn, A. Roiger, C. Weiß-Rehm, U. Burkhardt, U. Schumann, A. Marsing, M. Scheibe, A. Dörnbrack, C. Renard, M. Gauthier, P. Swann, P. Madden, D. Luff, R. Sallinen, T. Schripp, and P. L. Clercq (2024). Powering aircraft with 100% sustainable aviation fuel reduces ice crystals in contrails. *Atmos. Chem. Phys.*, *24*, 3813-3837, https://acp.copernicus.org/articles/24/3813/2024/.

Martin Frias, A., M. L. Shapiro, Z. Engberg, R. Zopp, M. Soler, and M. E. J. Stettler (2024). Feasibility of contrail avoidance in a commercial flight planning system: an operational analysis. *Environ. Res.: Infrastruct. Sustain. *, *4*, doi: 10.1088/2634-4505/ad310c, https://iopscience.iop.org/article/10.1088/2634-4505/ad310c.

Meerkötter, R., U. Schumann, P. Minnis, D. R. Doelling, T. Nakajima, and Y. Tsushima (1999). Radiative forcing by contrails. *Ann. Geophysicae*, *17*(8), 1080-1094, doi: 10.1007/s00585-999-1080-7.

Molloy, J., R. Teoh, S. Harty, G. Koudis, U. Schumann, I. Poll, and M. E. J. Stettler
(2022). Design principles for a contrail minimizing trial in the North Atlantic.
*Aerospace*, *submitted*.

Myhre, G., M. Kvalevag, G. Rädel, J. Cook, K. P. Shine, H. Clark, F. Karcher, K.
Markowicz, A. Karda, O. Wolkenberg, Y. Balkanski, M. Ponater, P. Forster, A.
Rap, and R. Rodriguez de Leon (2009). Intercomparison of radiative forcing
calculations of stratospheric water vapour and contrails. *Meteorol. Z.*, *18*(6), 585-
596, doi: 10.1127/0941-2948/2009/0411.

Newinger, C., and U. Burkhardt (2012). Sensitivity of contrail cirrus radiative forcing to
air traffic scheduling. *J. Geophys. Res.*, *117*, D10205, doi:
10.1029/2011JD016736.

Ovarlez, J., P. van Velthoven, G. Sachse, S. Vay, H. Schlager, and H. Ovarlez (2000).
Comparison of water vapor measurements from POLINAT2 with ECMWF
analyses in high humidity conditions. *J. Geophys. Res.*, *105*(D3), 3737-3744.

Ovarlez, J., J. F. Gayet, K. Gierens, J. Ström, H. Ovarlez, F. Auriol, R. Busen, and U.
Schumann (2002). Water vapor measurements inside cirrus clouds in northern
and southern hemispheres during INCA. *Geophys. Res. Lett.*, *29*(16), 60-61 - 60-
64, doi: 10.1029/2001gl014440.

Paoli, R., and K. Shariff (2016). Contrail modeling and simulation. *Annu. Rev. Fluid*
*Mech.*, *48*, 393–427, doi: 10.1146/annurev-fluid-010814-013619.

Poll, D. I. A. (2018). On the relationship between non-optimum operations and fuel
requirement for large civil transport aircraft, with reference to environmental
impact and contrail avoidance strategy. *Aero. J.*, *122*(1258 ), 1827-1870, doi:
10.1017/aer.2018.121

Poll, D. I. A., and U. Schumann (2022). An estimation method for the fuel burn and other
performance characteristics of civil transport aircraft. Part 3 Generalisation to
cover climb, descent and holding. *Aero. J.*, *submitted*.

Poll, D. I. A., and U. Schumann (2024). On the conditions for absolute minimum fuel
burn for turbofan powered, civil transport aircraft and a simple model for wave
drag. *Aero. J.*, *online*, https://doi.org/10.1017/aer.2024.10.

Ponater, M., S. Dietmüller, C. Frömming, and L. Bock (2013). The shortwave to
longwave ratio in contrail radiative forcing as evident in two radiation schemes
used for global GCMs. In *13th EMS annual meeting, 9.-13. Sept. 2013, Reading,*
*UK.* , edited.

Ponater, M., M. Bickel, L. Bock, and U. Burkhardt (2021). Towards determining the
contrail cirrus efficacy. *Aerospace*, *8*(42), 1-10, doi: 10.3390/aerospace8020042.

Prather, M., and R. Sausen (1999). Potential climate change from aviation. in *Aviation*
*and the Global Atmosphere. A Special Report of IPCC Working Groups I and III*,
edited by J. E. Penner, D. H. Lister, D. J. Griggs, D. J. Dokken and N. Mc
Farland, pp. 185-215, Cambridge University Press, Cambridge, UK.

Sassen, K. (1997). Contrail-cirrus and their potential for regional climate change. *Bull.*
*Amer. Meteorol. Soc.*, *78*(9), 1885-1903.

Sausen, R., I. Isaksen, D. Hauglustaine, V. Grewe, D. S. Lee, G. Myhre, M. O. Köhler,
G. Pitari, U. Schumann, F. Stordal, and C. Zerefos (2005). Aviation radiative forcing in 2000: An update on IPCC (1999). *Meteorol. Z.*, *14*, 555 - 561, 10.1127/0941-2948/2005/0049.

Schumann, U., and P. Wendling (1990). Determination of contrails from satellite data and observational results. In: Air Traffic and the Environment – Background, Tendencies and Potential Global Atmospheric Effects. U. Schumann (Ed.), Lecture Notes in Engineering. *Springer Berlin*, 138-153.

Schumann, U. (1994). On the effect of emissions from aircraft engines on the state of the atmosphere. *Ann. Geophysicae*, *12*(5), 365-384.

Schumann, U. (2002). Contrail Cirrus. in *Cirrus*, edited by D. K. Lynch, K. Sassen, D. O'C. Starr and G. Stephens, pp. 231-255, Oxford Univ. Press, Oxford.

Schumann, U., F. Arnold, R. Busen, J. Curtius, B. Kärcher, A. Petzold, H. Schlager, F. Schröder, and K. H. Wohlfrom (2002). Influence of fuel sulfur on the composition of aircraft exhaust plumes: The experiments SULFUR 1-7. *J. Geophys. Res.*, *107*(D15), 4247, doi: 10.1029/2001JD000813.

Schumann, U., and K. Graf (2013). Aviation-induced cirrus and radiation changes at diurnal timescales. *J. Geophys. Res.*, *118*(5), 2404-2421, doi: 10.1002/jgrd.50184.

Schumann, U., P. Jeßberger, and C. Voigt (2013). Contrail ice particles in aircraft wakes and their climatic importance. *Geophys. Res. Lett.*, *40*(11), 2867-2872 doi: 10.1002/grl.50539.

Schumann, U., and A. Heymsfield (2017). On the lifecycle of individual contrails and contrail cirrus. *Meteor. Monogr.*, *58*(3), 3.1-3.24, doi: 10.1175/AMSMONOGRAPHS-D-16-0005.1.

Schumann, U., C. Kiemle, H. Schlager, R. Weigel, S. Borrmann, F. D'Amato, M. Krämer, R. Matthey, A. Protat, C. Voigt, and C. M. Volk (2017). Long-lived contrails and convective cirrus above the tropical tropopause. *Atmos. Chem. Phys.*, *17*, 2311-2346, doi: 10.5194/acp-17-2311-2017.

Schumann, U., I. Poll, R. Teoh, R. Koelle, E. Spinielli, J. Molloy, G. S. Koudis, R. Baumann, L. Bugliaro, M. Stettler, and C. Voigt (2021). Air traffic and contrail changes over Europe during COVID-19: A model study. *Atmos. Chem. Phys.*, *21*(10), 7429–7450, doi: 10.5194/acp-21-7429-2021.

Spichtinger, P., and K. M. Gierens (2009). Modelling of cirrus clouds – Part 1a: Model description and validation. *Atmos. Chem. Phys.*, *9*, 685–706, doi: 10.5194/acp-9-685-2009.

Sussmann, R., and K. Gierens (1999). Lidar and numerical studies on the different evolution of vortex pair and secondary wake in young contrails. *J. Geophys. Res.*, *104*, 2131-2142.

Teoh, R., M. E. J. Stettler, A. Majumdar, U. Schumann, B. Graves, and A. Boies (2019). A methodology to relate black carbon particle number and mass emissions. *J. Aeros. Sci.*, *132*(June), 44-59, doi: 10.1016/j.jaerosci.2019.03.006.

Teoh, R., U. Schumann, A. Majumdar, and M. E. J. Stettler (2020a). Mitigating the climate forcing of aircraft contrails by small-scale diversions and technology adoption. *Environmental Science and Technology*, *54*(5), 2941–2950, doi: 10.1021/acs.est.9b05608.

Teoh, R., U. Schumann, and M. E. J. Stettler (2020b). Beyond contrail avoidance:
Efficacy of flight altitude changes to minimise contrail climate forcing. *Aerospace*,
*7*, (9), 121, doi: 10.3390/aerospace7090121.

Teoh, R., U. Schumann, E. Gryspeerdt, M. Shapiro, J. Molloy, G. Koudis, C. Voigt, and
M. E. J. Stettler (2022a). Aviation contrail climate effects in the North Atlantic from
2016-2021. *Atmos. Chem. Phys.*, *22*, 10919–10935, https://doi.org/10.5194/acp-
22-10919-2022.

Teoh, R., U. Schumann, C. Voigt, T. Schripp, M. Shapiro, Z. Engberg, J. Molloy, G.
Koudis, and M. E. J. Stettler (2022b). Targeted use of sustainable aviation fuel to
maximise climate benefits. *Environmental Science and Technology*, *56*(23),
17246–17255, https://doi.org/10.1021/acs.est.2c05781.

Teoh, R., Z. Engberg, U. Schumann, C. Voigt, M. Shapiro, S. Rohs, and M. E. J. Stettler
(2024a). Global aviation contrail climate effects from 2019 to 2021. *Atmos. Chem.
Phys.*, *accepted*, https://doi.org/10.5194/egusphere-2023-1859.

Teoh, R., Z. Engberg, M. Shapiro, L. Dray, and M. E. J. Stettler (2024b). The high-
resolution Global Aviation emissions Inventory based on ADS-B (GAIA) for 2019–
2021. *Atmos. Chem. Phys.*, *24*, 725-744,
https://acp.copernicus.org/articles/24/725/2024/.

Unterstrasser, S. (2016). Properties of young contrails – a parametrisation based on
large-eddy simulations. *Atmos. Chem. Phys.*, *16*, 2059-2082, doi: 10.5194/acp-
16-2059-2016.

Weickmann, H. (1945). Formen und Bildung atmosphärischer Eiskristalle. *Beitr. Phys.
Atmos. (Contr. Atm. Phys.)*, *28*, 12-52.

---

## Author Response (AR1)

**Anonymous Reviewer #1**

Review of "Understanding the Role of Contrails and Contrail Cirrus in Climate Change: A Global Perspective" by Singh et al. (2024)

Overview

The manuscript summarizes the microphysical processes concerning contrail formation, aging, and transition into cirrus clouds. The current understanding of the role of aviation soot in affecting contrail formation and cirrus activation is reviewed, from all three perspectives of in situ measurements, remote observations, and modeling. Recently gained knowledge of contrail cirrus occurrence from filed observations is highlighted in the paper. The article emphasized two key impactors that cause the high uncertainties in estimating the climate impact of contrail cirrus, namely contrail coverage and soot indirect effect. Available contrail observation datasets models for contrail property and evolution studies, and for understanding the climate impact of contrail now and future are listed in the manuscript. The uncertainties and research gaps in contrail cirrus and its radiative forcing are identified by the authors.

Research needs for a better understanding of the impact of contrail cirrus are pointed out in the manuscript to bridge the knowledge gaps. Finally, the manuscript looks into the current status of contrail avoidance and climate trade-offs from the aspects of slightly deviating the aircraft from contrail-forming regions and using climate-friendly aviation fuels.

The manuscript is well written with a clear structure and is easy to follow. However, to have a comprehensive understanding of the role of contrails and contrail cirrus in climate change and the status of the knowledge, there are some more aspects that the authors should elaborate.

**Major comments:**

**1. Comments from referee:**
The authors detailed the different processes that evolved in contrail formation, evolution, and transition into cirrus clouds. It is of high importance to understand these processes, which also lay the foundation for the modeling of contrails and contrail cirrus's radiative impact. However, the radiative impact of contrail cirrus (including linear contrails) is closely associated with optical properties and the coverage of contrail cirrus, in which the age of contrail cirrus plays an mportant role. As this is a review article, it would be good to provide an overview (a summary table or view graph) of contrail characteristics (size, number density, extinction properties, width, length, optical depth…) at various stages of contrail formation and evolution, namely jet, vortex, dissipation, and diffusion regimes, respectively, especially in the diffusion stage, during which the radiative impact is the most critical, from the perspectives of both measurement and modeling.

**Author's response:** Thanks to the reviewer for the valuable suggestions. To address your comment, we have added two tables, Table 1 (qualitative) Overview of the qualitative comparison of contrail characteristics across different stages of evolution, and Table 2. (quantitative) Summary of contrail properties during formation, evolution, and dissipation. (The value ranges are based on the understanding of contrail studies from Heymsfield et al., (2002); Schumann et al., (2018); Chauvigné et al., (2012); Sassen and Cho, (1999); Rädel and Hess, (2001).

**2. Comments from referee** :
In Sect. 3, the authors introduced different models for contrail properties and evolution in individual and overlapped contrail cases. Microphysical processes implemented in the models are described briefly. The treatment of contrails in global models is also reviewed. The Hadley Centre climate model HadGEM2 from the UK Met Office has also been implemented with contrail parameterization for evaluate the contrail's radiative forcing.
Rap, A., P. M. Forster, A. Jones, O. Boucher, J. M. Haywood, N. Bellouin, and R. R. De Leon (2010), Parameterization of contrails in the UK Met Office Climate Model, J. Geophys. Res., 115, D10205, doi:10.1029/2009JD012443.
I suggest the authors add a cross-model intercomparison and assessment table, especially for the multiple models for simulating contrail properties and lifespan, and evaluate the models against observations. From this, the readers will get a comprehensive overview of the performance of currently available contrail models. It will also outline what is needed in the future for model development from the perspective of properly characterizing contrails and contrail cirrus.

**Author's response:** Table 4 has been added to clarify the cross-model intercomparison based on our understanding.

**3. Comments from referee**
The same for Sect. 4. The authors reported independently the results from various climate impact models and future RF projections. The agreement and disagreement between models on the climate impact of contrail cirrus now and in the future are not discussed. And if there are any disagreements, what are the potential causes for the models to behave differently?

**Author's response:** Thanks to the reviewer for valuable feedback on Section 4. We've addressed this oversight by adding text and a new table that explores these points. (Table 9, Agreement and disagreement between climate models on contrail cirrus radiative forcing)

**4. Comments from referee**
(1) In Sect. 5, several observational datasets that are directly linked to contrail and detection, such as air traffic data, engine emission indices, in-situ contrail observations, and satellite image libraries, are included. However, it is also highly important to include datasets with critical parameters for contrail formation, such as temperature and relative humidity with respect to ice in the upper troposphere and the tropopause region. One such dataset is provided by the Inservice Aircraft for a Global Observing System (IAGOS) research infrastructure with high spatial and temporal resolution, which has already been used for contrail occurrence studies and assessment of forecast models, e.g., Gierens et al., 2020 (doi:10.3390/aerospace7120169).

**Author's response:** Thank for the reviewer for your valuable comment on Section 5. We completely agree that including datasets with critical parameters for contrail formation is crucial for a comprehensive understanding of contrail occurrence and impacts. A new paragraph is added to section 5.

(2) Regarding the satellite-based contrail dataset, the human-label contrail datasets, the AI detection and machine learning algorithms are helpful in monitoring contrails, perhaps even for contrail lifetime and coverage studies. The case studies mentioned by the authors are mostly based on contrails with clear line shapes. However, these datasets are rather small and are limited to certain regions, camera angles and quantities, and spatial resolution. I am not convinced it is of great help to improve the estimation of contrail's radiative forcing which are closely associated with contrail ice particle microphysical and optical properties. To achieve this, contrail optical thickness should be retrieved, which is not well resolved by satellite images and relies on robust retrieval algorithms. Furthermore, satellites cannot detect contrails until tens of minutes later and overlapped contrails and embedded natural cirrus add extra challenges to contrail detection and parameterization for satellite observations. Finally, ice-supersaturated regions in the UTLS, where contrails prefer to occur, exhibit strong variability in locations, time, and scales. From all above, I think the authors should be prudent in judging the capability of satellite contrail datasets and AI detection algorithms to support contrail mitigation and robust estimation of contrails' climate impact.

**Author's response:** Thanks for your valuable comments on Section 5.2 regarding satellite-based contrail datasets. We acknowledge the limitations you've highlighted and agree that these datasets require careful consideration when estimating contrail radiative forcing. We have addressed your concerns by incorporating the revisions. We have added new paragraphs explicitly acknowledging the limitations of current satellite-based contrail detection methods. These paragraphs discuss the points you raised.

**5. Comments from referee**

**Sect. 7** Microphysical processes of contrail formation in relation to the role of soot/background aerosols and heterogeneous/homogeneous freezing mechanism also pose big uncertainties in understanding the effects of contrails. Though it is known that fresh contrail contains thousands of ice particles, it is still debated on the ice nucleating properties of soot particles for contrail the formation, see e.g., Righi et al., Exploring the uncertainties in the aviation soot–cirrus effect, Atmos. Chem. Phys., 21, 17267–17289, https://doi.org/10.5194/acp-21-17267-2021, 2021Testa et al., Soot aerosol from commercial aviation engines are poor ice nucleating particles at cirrus cloud temperatures, EGUsphere [preprint], https://doi.org/10.5194/egusphere-2023-2441, 2023 Testa et al., Contrail processed aviation soot aerosol are poor ice nucleating particles at cirrus temperatures, EGUsphere [preprint], https://doi.org/10.5194/egusphere-2024-151, 2024.

Contrail-cirrus observation, either with remote-sensing or in-situ measuring technique, has high requirements for instrumentation, facilities, and platforms, technically and financially. In-situ instruments provide high temporal and resolution data but must meet the harsh conditions of cold temperatures, vibrations, and limited dimensions for airborne measurements. The measurement of soot cores at extremely cold temperatures ~60°C is challenged so far with ice nucleating particle counter. To integrate a complete set of high-precision instruments for contrail cirrus properties is also demanding. Remote-sensing techniques are vulnerable to the blender of natural and contrail

cirrus and should provide high enough temporal and spatial resolution to track the width, length, and depth of the contrails, etc.

Further research is also needed to address these uncertainties and gaps.

**6. Comments from referee**

Water vapor observation and humidity sensors. Dense, wide-coverage and high-resolutionctemperature, water vapor and relative humidity with respect to ice (RHi) are essential for studying and predicting contrail occurrences and properties. In particular, these data should be provided with high accuracy and precision. They are also valuable for evaluating weather and contrail forecast models. The IAGOS infrastructure provides quality-checked temperature and RHi data obtained from routine passenger aircraft measurements since 1994, have been used in various studies. The IAGOS capacitive hygrometer is the same prototype of the Humicap sensor from Vaisala that is widely used in weather balloon sondes. It is one of the three operational global water vapor observation programs as documented in the WMO report on supporting water vapor measurements within an aircraft meteorological data relay program.

The other two technology types for aircraft-base water vapor measurement are the Tropospheric Airborne Meteorological Data Reporting (TAMDAR) sensor system and the Water Vapor Sensing System II (WVSS-II) developed by FLYHT, which provide data to meteorological centers. Rolf et al. (2023) has conducted an evaluation of compact hygrometers for continuous airborne measurements in the range of 20 to 20,000 ppmv. Rolf et al., Evaluation of compact hygrometers for continuous airborne measurements, Meteorol. Z. 2023 Vol. 20 Issue 20, DOI: 10.1127/metz/2023/1187.

Basically, the technology for airborne water vapor and humidity sensors is well established. What matters more is the quality of the data, maintenance of the programs, and the joint between datasets and programs. How to incorporate the programs and networks to facilitate contrail observation, reconcile observations with contrail forecasting, and conduct safe contrail mitigation are still missing and should be addressed in future research.

**Author's response (to comments 5 and 6):** We greatly appreciate your valuable insights on Section 7. Your comments have prompted us to address the following key points:

Emphasizing uncertainties related to soot particles and ice nucleation in contrail formation, as referenced in Righi et al. (2021) and Testa et al. (2023, 2024).

Recognizing the challenges associated with contrail-cirrus observations using both in-situ and remote sensing techniques.

Highlighting the significance of enhancing data quality, integrating programs, and applying existing water vapor observation technologies (such as IAGOS, TAMDAR, and WVSS-II) in contrail studies.

In response to your feedback, we have made a series of revisions to Section 7:

We have included a new paragraph specifically addressing the ongoing debate surrounding the ice nucleating properties of soot particles and their potential impact on contrail formation.

The discussion on observation challenges faced with in-situ and remote sensing techniques for contrail-cirrus characterization has been expanded.

We have acknowledged the presence of established water vapor observation technologies and underscored the importance of future research aimed at enhancing data quality, program integration, and their relevance in contrail observations.

**7. Comments from referee**
In the conclusion, the author should elaborate what is the well-established knowledge and where are the research difficulties and gaps that careful investigation and dedicated research is required.

**Author's response:** We appreciate the reviewer's suggestion to further differentiate between well-established knowledge and research gaps. We have revised to address this point by:

Clearly separating the established knowledge about contrail formation, characteristics, life cycle, and potential climate impacts.
Elaborating on the key research difficulties and gaps in our understanding, including microphysical processes, radiative forcing uncertainties, and contrail representation in climate models.
We have also added a new paragraph to the conclusions section that highlights emerging mitigation strategies and technological advancements.

**Minor comments:**
**1. Comments from referee**
Some of the figures and tables are not well clarified in short captions with abbreviations, e.g., Figure 7 and Table 2, Table 4, etc. The color bar scales and ticks are not readable, e.g., Figure 13.
Line 55: Why not just use "Aviation-induced cirrus" instead of "Contrails and resulting contrail clouds formation"?

**Author's response:** Thanks, the reviewer for pointing this out. Now we have revised it as per the suggestion.

**2. Comments from referee**
Line 78: "Many global climate chemistry models used to study … do not incorporate the impacts Models that do account for contrails…": What impacts does the author mean, e.g., contrail cirrus characteristics, contrail parameterization or so?
**Author's response:** Thanks for pointing this out! The sentence clarified to better distinguish between modeled contrail coverage and their actual effects on climate. (all the changes are in blue font in the current revised manuscript)

**3. Comments from referee**
Line 136: "optical characteristics and lifespan of contrails are significant dynamics of contrail formation and…": "dynamic" is usually referred to the movement of air parcels. Does the author mean "optical characteristics and lifespan are key aspects of contrail formation"?
**Author's response:** Thank you for your insightful comment! You're right, "aspects" is a more accurate term in this context. We have revised the paragraph to reflect this change, as shown below.
Changes made:
Replaced "dynamics" with "key aspects".

Improved clarity by adding "these" before "meteorological factors" in the second sentence.

**4. Comments from referee**
Line 232: Previous studies have studied the occurrence frequency distributions of ice-supersaturated regions (ISSRs) with high temporal-resolution aircraft measurements. The variabilities of ISSRs in the upper troposphere and different regions with high air traffic density have also been explored using aircraft and weather forecast reanalysis data, such as Petzold et al., Ice-supersaturated air masses in the northern mid-latitudes from regular in situ observations by passenger aircraft: vertical distribution, seasonality and tropospheric fingerprint Atmos, Chem. Phys. 2020 Vol. 20 Issue 13 Pages 8157-8179. Reutter et al., Ice supersaturated regions: properties and validation of ERA-Interim reanalysis with IAGOS in situ water vapor measurements, Atmos. Chem. Phys. 2020 Vol. 20 Issue 2 Pages 787-804

**Author's response:** Thank you for the insightful suggestions! We've added two new paragraphs to the manuscript that address your points:
The first paragraph discusses the occurrence frequency distributions of ice-supersaturated regions (ISSRs) based on the work of Petzold et al. (2020).
The second paragraph explores the validation of ISSRs in reanalysis data using the research by Reutter et al. (2020).

**5. Comments from referee**
Line 242-246: The two sentences repeat each other.
**Author's response:** We have corrected the redundancy

**6. Comments from referee**
Line 271: The impact of soot on contrails and soot was discussed based on mostly old literature. There were new studies appearing in the past years, e.g., Testa et al. (2023 and 2024). The authors should sample more recent studies to provide the current understanding of this topic.

**Author's response:** Thank you for pointing out the dated literature! We have updated the paragraph to include recent studies, such as Testa et al. (2023, 2024), which explore the interaction of soot with contrails and its potential role in ice nucleation.
(Note: Testa et al. (2024) is not yet published, " It is a preprint/under review to ACP/AGU"

**7. Comments from referee**
Line 280: The authors described the cirrus observation near Munich in 2012 and traced soot emissions related to the cirrus formation with back trajectories. Can the authors provide the source of the case study? Or give more details on the observation and back-trajectory calculations?

**Author's response:** Thanks for the insightful comment. We acknowledge that the information readily available in the referenced paper (Schumann and Heymsfield, 2017) doesn't explicitly mention the source of the case study itself.
Observation Details: We can consult the additional reference, Schumann et al. (2013a), to obtain specifics about the camera observation method and potentially gain insights from ground-based lidar and radar measurements used to characterize the cirrus cloud.

 While the specific method for back-trajectory calculations isn't explicitly stated, the source indicates the identification of aircraft emissions upstream roughly 12 hours prior to the cirrus formation.

**8. Comments from referee**

Line 334: The authors cited Wolf et al. (2023) for their exploration of the dependence of contrail cirrus RE on different parameters. Can the authors refine the main messages of Wolf et al and provide readers an overview of the impact of these parameters or conclude which parameter have the greatest impact on cirrus RF?

**Author's response:** We appreciate your feedback on Wolf et al. (2023). We have clarified their role by explaining the influence of various microphysical, macrophysical, and optical properties on the cirrus cloud radiative effect in the revised manuscript.

**9. Comments from referee**

Line 357: In this paragraph, the authors discussed dynamics and ice crystal evolution in the wake of the regimes of vortex dissipation and diffusion. Figure 6 is used here to illustrate the contrail formation processes in aircraft vortexes. However, the processes in Figure 6 are happening within ~100 sec upon the emission of aircraft exhaust, namely the jet and vortex regimes in Figure 5. It fits more in the previous paragraph.

**Author's response:** Thanks to the reviewer for feedback on this. We've moved it to the previous paragraph as suggested, to better align.

**10. Comments from referee**

Line 446: "In exhaust rich in soot, the number of soot particles… increases with decreasing ambient temperature. This, in turn, raises plume cooling rates and levels of plume supersaturation over liquid water." The causal relationship is reversed. The increased cooling rates and supersaturation are caused by decreasing temperature in the plume, not by the increasing of soot particles. The freezing of water vapor on particles would release heat and consume the supersaturation in plume air masses.

**Author's response:** Thanks to the reviewer for clarifying the causal relationship. We've revised the sentence to emphasize that decreasing ambient temperature is the primary driver of cooling and supersaturation, with soot particles playing a secondary role in ice nucleation (especially for soot-rich exhaust).

**11. Comments from referee**

Line 462: in the context of contrails formation → in the context of contrail formation

**Author's response:** Thanks, now it is corrected.

Line 526: Sect. 2.7. The authors sampled recent studies reporting the observations of contrails and contrail cirrus in slightly ice-subsaturated environments, which is new knowledge of contrail formation and evolution. However, as this subsection is about lessons learned from observations

of contrails and their properties, the authors may also consider from a measurement perspective the limiting factors of contrail property observations.

**Author's response:** Thank you for your suggestion regarding observational limitations in Section 2.7. We've addressed this by adding new paragraphs that discuss the challenges associated with observation constraints.

**12. Comments from referee**
Line 569: Mahnke et al. (2022) was cited, but it didn't appear in the reference list.

**Author's response:** Thanks to the reviewer for pointing this out, it is now listed in the reference section.

**13. Comments from referee**
Line 638: In this study → In this model

**Author's response:** Thanks, It has been changed now.

**14. Comments from referee**
Line 671 and Line 679: The order of the two paragraphs can be reversed for readability.

**Author's response:** Thanks to the reviewer for pointing this out, now the paragraphs are reversed for convenience to the reader.

**15. Comments from referee**
Line 719: This sentence repeats the previous one.

**Author's response:** Thanks, now we have deleted the repeated sentence.
**16. Comments from referee**
Sect. 3.2.2: Paragraphs 2-4. Regarding the development of the contrail cirrus parameterization in the mode ECHAM5-HAM, the content jumps back and forth. Following a chronological order of the citations may show the improvement of the parameterizations in the model. Paragraphs 3 and 4 can be combined by saying "Bock and Burkhardt (2016) extended the contrail cirrus parameterization (CCMod) initially proposed by Burkhardt and Kärcher (2009) by introducing contrail cirrus ice crystal number concentration and volume as additional prognostic variables."
.

**Author's response:** Thank the reviewer for the suggestion regarding the order of paragraphs in Section 3.2.2. We've combined paragraphs 3 and 4 to create a clearer chronology of the contrail cirrus parameterization development in ECHAM5-HAM.

**17. Comments from referee**
Line 990: I guess the authors mean that the maximum flight density will occur at a higher, rather than a lower, altitude of around 200 hPa in 2050 compared to 240 hPa in 2006 if flight altitudes

are expected to rise by 0.3 to 1.5 km. From Table 3, readers cannot recognize which row represents the projection resulting from altitude increase. The authors should explain shortly the different simulated scenarios.

"They found that the maximum flight density would occur at a lower altitude in 2050" is redundant (Line 991).

**Author's response:** Thank you for pointing out the redundancy in the explanation of air traffic changes and Table 3 (now Table 6). We've revised the text to combine the information about air traffic volume increase and flight altitude shift (0.3-1.5 km) for improved clarity. Table 6 now directly references this combined scenario, which projects a fourfold increase in air traffic and a lower maximum flight density (200 hPa in 2050 compared to 240 hPa in 2006).

**18. Comments from referee**
Line 1016: mWm-2 -> mW m-2, the same elsewhere in the context.
**Author's response:** Now it is corrected

**19. Comments from referee**
Line 1037: With the values listed in the table, a 3-4-fold increase is expected in central Europe, and over 400% increase in North America. For readability, the authors should note shortly what stands for RCP4.5, RCP 8.5, SC1, 2, and 3.

**Author's response:** Now we have noted about RCP4.5, RCP 8.5, SC1, 2, and 3.

**20. Comments from referee**
Line 1133: The Schmidt-Appleman criterion is commonly cited as Schmidt, 1941, Appleman,1953 and Schumann, 1996.

**Author's response:** Thanks for pointing this out, now we have cited as per the suggestion

**21. Comments from referee**
Line1413: "rarer but more ice crystals" -> rarer but larger crystals
**Author's response:** Now, it is corrected

**22. Comments from referee**
Line 1470: The paper by Moor et al. (2017) reported soot reduction at cruises with adequate. measurements and should be added here.
**Author's response:** Thanks to the reviewer for suggesting the article, now we have added in detail the finding by Moor et al. (2017)

**23. Comments from referee**
Line 1472: "Our study involved the measurement …" -> Their study…
**Author's response:** Now it is corrected

**Formatting issues:**
**1. Comments from referee**

Line 110: The formatting of the subsection title is not in coherence with the other subsections. There is a mixture of British and American English, e.g., "Ageing" and "Aging" (Line 106).

**Author's response:** We have updated the text based on American English

**2. Comments from referee**
Punctuation. The period is missing, e.g., Line 78, Line 1041, or Line 1120 repeated period, Line 570 mistake, and Line 491 with uppercase after a comma).

**Author's response:** Thanks to the reviewer for pointing these out; we have addressed these punctuations.

**3. Comments from referee**
Citations. Author names are missing in Line 827 – 828. Citation formatting error, e.g., Line 1140:Minnis et al. (1998 and Minnis et al. 2013).

**Author's response:** Thanks to the reviewer for pointing these out, now we have addressed these errors.

**4. Comments from referee**
References. The references are not in the same format. Some have a DOI, while many others don't. A DOI should be provided for each reference. The font style and size are not the same throughout the list.

**Author's response:** Thanks to the reviewer for pointing these out, we have revised the reference list and added the doi.

**Anonymous Reviewer #2**

Anonymous Review
April 18, 2024

Understanding the Role of Contrails and Contrail Cirrus in Climate Change: A Global Perspective by Dharmendra Kumar Singh, Swarnali Sanyal, Donald J. Wuebbles.

The topic of the paper is of high actual interest and well-suited for ACP. As stated in lines 100 ff, the paper aims to provide a comprehensive review of our understating of contrails… and to address the uncertainties….
Such a comprehensive review does not exist in the literature. It seems that not all authors have extensive experience in contrail research by themselves so many of their comments may raise critical discussion. The paper reads like a literature study in preparation for more specific research

by the authors on this topic. But different from reviews by acknowledged experts, who like their own work most, this paper is rather unbiased in the selection of materials and assessments, and this is refreshing.

The coverage is quite good but can be improved. Here I have several additions and offer comments on detail, see below. The paper is (of course) mostly a collection of existing knowledge. The conclusions are not really surprisingly new. I found the comments on Machine Learning aspects particularly interesting (I think I should try to learn more about them). Section 7, "Research needs and gaps " is worthwhile to have though not objective in all parts and partly hard to validate because of incomplete explanations or citations. I think it is worthwhile to have estimated numbers on the magnitudes of uncertainty but presently these numbers are not well justified.

The review misses many important publications. That should be improved. The paper is partly too long to keep me interested in all the details and collects a lot of materials. Still, the paper is worth reading but it needs amendments before it can be published. Such a long paper is hard to review, and I can give only a few specific comments. I would formulate many parts differently.

**Abstract**
**1. Comments from referee**
Line 12: The sentence "contrail cirrus enhances the impact of natural clouds on the climate" is misleading, because I read it first as if contrails would have no effect without natural cirrus. But that is not the case. Better: Contrail cirrus is similar to natural cirrus in having an impact on climate. Then I would start a new sentence on the uncertainties – without "although" or "however" or "but". Simply say that this paper discusses existing uncertainties.
The statement "there are still unresolved questions" is trivial. There will be always unresolved questions. In fact, with further research more and new questions may likely arise. So, you should condition this sentence a bit more, like "presently the state of the art is insufficient to assess the magnitude of the climate impact of contrails, e.g. on surface temperature, and its error bounds with high confidence" – or similar.

**Author's response:** Thanks to the reviewer for thoughtful feedback on our abstract. We've carefully considered your suggestions and made the necessary revisions.

**2. Comments from referee**
Line 16: add "mixing" as one of the issues.

**Author's response:** Thanks to the reviewer for reminding us to add "mixing", now it has been added.

**3. Comments from referee**
Line 56: this has been known basically for long (see IPCC, 1999). Other reviews of relevance are, e.g. (Schumann, 1994; Sassen, 1997; Brasseur et al., 1998; Fahey and Schumann, 1999; Prather and Sausen, 1999; Sausen et al., 2005; Brasseur et al., 2016

**Author's response:** Now we have cited some of the additional references.

**4. Comments from referee**
Line 68: what do you mean by atmospheric warming? It is correct for RF. However, the ERF relates RF to the global mean surface temperature change. Perhaps you can discuss how reasonable it is to assess the impact of contrails in terms of global metrics. Contrails are local events, are short-lived, and have certainly a very small global mean surface temperature climate impact. Contrails warm mainly the upper troposphere, not so much the surface. Most contrails occur over continents with little chance of long-lived effects by heating the oceans. The regional effects are likely far larger than global effects and should be easier to detect and quantify. Regional effects are of importance for the densely populated latitudes in the Northern Hemisphere where most air traffic occurs.

**Author's response:** Thanks to the reviewer for his/her valuable feedback concerning the use of global metrics for contrail impacts. We appreciate raising the limitations of radiative forcing (RF) and effective radiative forcing (ERF) in capturing localized effects. We've incorporated this point by acknowledging these limitations in the manuscript and emphasizing your observation about potentially greater regional impacts, especially in areas with high air traffic density.

**5. Comments from referee**
Line 78: sentence(s) incomplete and unclear.

**Author's response:** Now we have revised the paragraph.

**6. Comments from referee**
Line 81 is biased to one team and one model type only. Others should be included.

**Author's response:** Thanks to the reviewer for pointing this out, Now we have added others also.

**7. Comments from referee**
Line 85: there is far more literature on this aspect.

**Author's response:** Thanks to the reviewer for pointing this out, Now we have added more references.

**8. Comments from referee**
Fig 1 caption. "Current best estimate.." The estimate is already no longer the last one. Several studies have added new estimates. I would suggest: "Climate forcing estimate as published in 2021" or similar.

**Author's response:** Thanks to the reviewer for the suggestions. Fig. 1 and text were chosen to account for the time period.

**9. Comments from referee**

Fig 2. fails to mention the importance of mixing the jet exhaust gases with ambient air. Mixing controls early or late contrail formation and contrail lifetime (Lewellen and Lewellen, 1996; Gerz and Ehret, 1997; Sussmann and Gierens, 1999; Lewellen, 2014; Paoli and Shariff, 2016; Schumann and Heymsfield, 2017).

**Author's response:** Thanks to the reviewer for the suggestions. Fig 2 has been selected to convey basic knowledge of aircraft emissions to a non-expert audience, particularly focusing on the 'Schematic overview of exhaust plumes (contrail formation in low temperature) and their composition'. We appreciate the suggestions, and the suggested articles have been incorporated into other sections of this manuscript.

**10. Comments from referee**
Line 152. These 2 references do not cover this topic sufficiently. There are many more. I see that you cite them later. But this line is irritating. See line 178. See e.g. Sassen (1997).

**Author's response:** Now suggested references are added.

**11. Comments from referee**
Line 197. Further new reference: (Märkl et al., 2024).

**Author's response:** Thanks to the reviewer for suggesting including the latest publication, It is included.

**12. Comments from referee**
Line 246, Contrail formation was observed far earlier. See (aufm Kampe, 1942; 1943; Weickmann, 1945; Brewer, 1946; Schumann and Wendling, 1990; Schumann, 1994; Busen and Schumann, 1995).

**Author's response:** These publications have been added to the text.

**13. Comments from referee**
Line 256. The visible constraints were introduced earlier: by Appleman (1953).

**Author's response:** Added as suggested.

**14. Comments from referee**
Line 268 and Fig 4: a similar figure with a contrail spiral was given earlier in Fig. 11.2(Schumann, 2002). Impressive contrail observations were also presented by Laken et al. (2012) and (Mannstein and Schumann, 2005).

**Author's response:** We appreciate the reviewer's suggestion for an additional figure. However, we believe the current Figure 1 effectively conveys the key points discussed in the manuscript. We have also considered the references suggested by the reviewer and incorporated relevant information into the text.

**15. Comments from referee**

Line 277: it is now pretty clear that the soot acts as a condensation nucleus so that liquid droplets form which then freezes. The resultant ice particles likely do include the soot particles on which the water condensed. This does not exclude that some ice particles sublimate or that some soot remains dry throughout the contrail formation process so that also dry soot is found inside contrails. See the upcoming paper by Dischl et al. (Egusphere, to be published soon).

**Author's response:** Thank you for the insightful comment regarding the role of soot in contrail formation. We have added a paragraph as per the suggestion.

**16. Comments from referee**
Lines 279 to 288 have been taken from Schumann and Heymsfield (2017), but the reference is missing.

**Author's response:** Thanks to the reviewer for pointing this out, we have added the missed reference.

**17. Comments from referee**
Line 446: How do we know the current ranges of in-flight soot emission, Here you should cite papers by the team around Marc Stettler and Roger Teoh and colleagues (Teoh et al., 2019; Teoh et al., 2020a; Teoh et al., 2020b; Teoh et al., 2022a; Teoh et al., 2022b; Teoh et al., 2024a; Teoh et al., 2024b).

**Author's response:** The hatched area in Figure 7 represents the approximate range of current in-flight soot emission indices. We thank the reviewer for suggesting new articles. However, in section 2.5 *(Contrail ice crystal nucleation),* we could not find the relevant content from Teoh et al. for potential additions to the revised manuscript.

**18. Comments from referee**
Lines 462 to 483: For the discussion of the impact of the initial contrail ice particle number concentration (and hence the soot emissions) on contrails the cross-section integrated optical extinction and hence on climate forcing you should cite details of the study and explanation around Eq. (5) in Lewellen (2014). See also Section [17] in (Schumann et al., 2013).

**Author's response:** We appreciate the suggestion to incorporate details from Lewellen (2014) regarding the impact of initial ice crystal concentration (Ni) on contrail properties. To address this point, we've revised the manuscript to include a new paragraph highlighting key findings from his study. This paragraph discusses how higher Ni values lead to increased contrail lifetime, spreading, and optical effects, further emphasizing the role of Ni through the inclusion of Equation (5) from Lewellen (2014). We believe this revision strengthens our discussion and provides a more comprehensive understanding of Ni's influence on contrail properties.
*We could not find Section [17] in (Schumann et al., 2013) related to the above discussion https://amt.copernicus.org/articles/6/3597/2013/*

**19. Comments from referee**
Page 9: The summaries of Urbanek, Kärcher, and Wolf read like abstracts from these papers. The review does not combine these results critically with related results from other studies. This is a serious deficiency of this review and makes me uncertain whether this is an acceptable review.

**Author's response:** We're grateful for the reviewer's insightful suggestion to enhance our analysis of studies on contrail cirrus formation and interactions. In response, we've made revisions to the manuscript by adding new paragraphs that discuss the findings of Urbanek et al. (2018), Kärcher et al. (2021), and Wolf et al. (2023). These paragraphs shed light on the potential connection between aircraft emissions and a unique process of ice cloud formation (Urbanek et al., 2018), raises questions about the impact of contrail cirrus soot on overall cloud ice content, drawing from modeling of Kärcher et al. (2021), and underscores the vital role of ice crystal properties in determining the radiative effect of cirrus clouds (Wolf et al., 2023). We believe that these revisions bolster the critical analysis and offer a more thorough understanding of the intricate relationship between contrails, cirrus clouds, and aviation emissions.

**20. Comments from referee**
Lines 516 ff: Sedimentation. This section misses citations of related literature. See, e.g., Spichtinger and Gierens (2009)

**Author's response:** Thank you for the suggestion regarding sedimentation. We have incorporated a new paragraph addressing this topic, including a citation to Spichtinger and Gierens (2009). This paragraph discusses the relationship between ice crystal size and sedimentation velocity, highlighting its impact on contrail persistence.

**21. Comments from referee**
Line 563: 290 m/s, That is nearly the speed of sound. I doubt that this value is correct (230 m/s would sound more reasonable).

**Author's response:** Thanks to the reviewer for pointing this out, however, the value 290 m/s is taken from the published paper in ACP/EGU.
ML-CIRRUS 2014 campaign, Li et al., (2023)  https://acp.copernicus.org/articles/23/2251/2023/

**22. Comments from referee**
Lines 574 to 588: difficult to read. I do not understand what you want to say.

**Author's response:** Thank you for pointing out the complexity of the paragraph on contrail modeling (Lines 574-588/preprint manuscript). We've revised it to improve clarity. The revised text will emphasize the two main points:
*Different models address contrail impacts at various scales:* Local-scale models simulate how turbulence and wake dynamics influence contrail ice crystals, while larger-scale models analyze regional effects on radiative forcing (RF).
*Modeling plays a crucial role due to measurement limitations:* Uncertainties in measuring contrail cirrus properties necessitate robust models for estimating RF (Table 3).
This revision focuses on the core concepts and simplifies the language for better understanding.

**23. Comments from referee**
Line 606, replace 2016 by 2017.

**Author's response:** Thanks to the reviewer for pointing this out, It is replaced.

**24. Comments from referee**
Lines 617: the LES is of course far more expensive than a plume model and hard to apply globally. We need improved plume models, e.g., based on LES. By the way: Much work in this respect has been done by Unterstrasser (2016).

**Author's response:** We understand the reviewer's concern. Unterstrasser (2016) is cited and discussed in a different paragraph within the manuscript.

**25. Comments from referee**
Line 732. The comparisons have been done in several follow-up papers of that cited here.

**Author's response:** We agree.

**26. Comments from referee**
Line 780: You refer to the age of 450 s without comments. To me, 450 s is far too long.

**Author's response:** Thank you for raising the important point about the 450-second contrail initialization time. We have revised the paragraph, We agree that this value warrants further discussion. We acknowledge the potential limitations of a 450-second delay, especially for early contrail evolution studies. Future work needs to explore the sensitivity of model results to this parameter and potentially refine it for a more realistic representation.

**27. Comments from referee**
Line 823: neither 10 nor 7.5 um is correct. That value varies by large factors.

**Author's response:** We discuss that the assumed emission ice particle diameter was adjusted from the original parameterization (10 µm) to 7.5 µm to better align with observations (e.g., Lee et al., 2021).

**28. Comments from referee**
Line 876: This was first discussed by Meerkötter et al. (1999)

**Author's response:** Thanks to the reviewer for pointing this out. Discussion has been added to the text.

**29. Comments from referee**
Line 1163. See also the development of open-access performance codes: (Poll, 2018; Poll and Schumann, 2022; 2024).

**Response:** Thanks to the reviewer for the suggestions. We have added (Poll, 2018)

**30. Comments from referee**
Line 1153: Which reference do you refer to with Schumann, 2016?

**Author's response:** We have corrected the year Schumann, 2017 for (flight data from sources such as air traffic control or ground-based observations)

**31. Comments from referee**
Line 1130. Ettenreich observed in 1915 but published in 1919.

**Author's response:** Thanks to the reviewer for pointing this out, we have corrected this.

**32. Comments from referee**
Line 1158: See the GAIA data set (Teoh et al., 2024b).

**Author's response:** Thank you for suggesting the GAIA dataset (Teoh et al., 2024b). We've incorporated a new paragraph highlighting its value for contrail studies. GAIA's detailed picture of global aviation activity, including spatiotemporal variations in emissions, offers valuable insights into the link between aircraft types, flight routes, and contrail formation potential. Integrating GAIA with existing datasets like COLI can enhance our understanding of contrail occurrence and inform mitigation strategies.

**33. Comments from referee**
Line 1252: Limited literature exists". See, e.g., measurements of ice particles by the Geophysica aircraft at about 20 km height above tropical convection by measuring its own contrail (Schumann et al., 2017).

**Author's response:** Thank you for suggesting the recent review. We've incorporated it to highlight the need for a comprehensive understanding of contrail effects, particularly for supersonic aircraft where flight patterns and atmospheric conditions pose unique challenges.

**34. Comments from referee**
The paper by Myhre (2009) is certainly outdated There are several recent studies citing this paper.

**Response:** We have revised the paragraph with the addition of one more study.

**35. Comments from referee**
Line 1299 How do you come to the value of 55 %? It seems too large to me.

**Author's response:** Thanks to the reviewer for pointing this out. We have revised the text to further explain this.
Lee et al. (2021) estimated the overall uncertainty in the radiative response of contrail cirrus to be around 55%, assuming the independence of different uncertainties. This value excludes uncertainty A3 (related to radiative transfer for young contrails) and the impact of ice crystal soot cores. They arrived at this value by summing the individual uncertainties from category A (radiative transfer scheme, cloud heterogeneity, ice crystal habit) – estimated at 35%, 35%, and 20% respectively. It's important to remember that these uncertainties may not be perfectly independent, and the true combined effect could be slightly higher or lower than a simple linear addition.

**36. Comments from referee**
Linem1346: Contrail effTableects on climate will never be "fully" understood. But also many other things are incompletely understood and nevertheless practically addressed. (See, e.g., smog in cities, observed in the 1950's and soon thereafter regulated by US federal institutions (was it ERL, I forgot the name) and later by ICAO.

**Author's response:** Thank you for your valuable feedback. We've revised the sentence on contrail effects to acknowledge the ongoing nature of scientific understanding. We emphasize the importance of further research in reducing uncertainties and improving our ability to predict and mitigate the impact of contrails on climate.

**37. Comments from referee**
Line 1359 Again: ERF is an insufficient climate metric because it does not account for regional (northern mid-latitudes) phenomena and other climate changes besides mean surface temperature.

**Author's response:** Thank you for highlighting the limitations of Effective Radiative Forcing (ERF) as a sole climate metric. We acknowledge that ERF primarily focuses on the global mean radiative impact and may not fully capture regional variations or complex feedback mechanisms. We have modified the paragraph in the revised manuscript.

**38. Comments from referee**
Line 1503: As said before "Uncertainties persist"′. I must tell you: uncertainties will be there also in 100 years from now. The existence of uncertainties is not a sufficient argument to start mitigation actions soon. We only have to make sure that the actions reduce and not increase the climate effect. (It is like building a bridge over a river. The only thing that counts is that the likelihood that the bridge fails is sufficiently small to be acceptable. That problem was solved at least 2000 years before today, in spite of very large uncertainties. I suggest carefully rethinking talking about uncertainties.

**Author's response:** Thank you for your insightful comment regarding the discussion of uncertainties. We agree that uncertainties are inherent to scientific inquiry, and they will likely persist to some degree even in the future.
We've revised the paragraphs to acknowledge this ongoing research while emphasizing the potential consequences of projected increases in contrail cirrus radiative forcing. We believe that despite uncertainties, the potential impact necessitates a proactive approach to mitigation. The revised paragraphs highlight several promising mitigation strategies currently being explored, alongside the importance of continuous refinement as our understanding of contrail effects evolves.

**39. Comments from referee**
Chapter 7 "Uncertainties and Research Gaps" is certainly an important part of the paper.
Here many questions arise. You provide a number of the uncertainties in percentages. How are these numbers derived? That is unclear in most cases. Please explain. I assume you will note: there are not only uncertainties but also uncertainties of how large the uncertainties are. The quote percentage values add (linearly) up to a total of more than 100 %. Please discuss.

**Author's response:** The reviewer's valuable comments on Chapter 7's uncertainties are appreciated. To address concerns about how uncertainty percentages were derived, references will be added for each value, citing studies that quantify the impact of various factors on contrail cirrus radiative forcing (RF) through techniques like sensitivity analyses. We acknowledge the "uncertainty of uncertainties" and will clarify that percentages represent independent estimates, with the actual combined effect potentially falling outside the summed values. The manuscript will discuss how non-linear interactions between uncertainties (e.g., ice crystal size and soot cores) and the possibility of missing uncertainties can contribute to this. By incorporating references, addressing limitations, and acknowledging potential non-linearities and missing factors, Chapter 7 will provide a more transparent and comprehensive discussion of uncertainties surrounding contrail cirrus RF.

**40. Comments from referee**

Line 1284: The reference to Myhre et al. (2009) in assessing radiative transfer schemes is outdated. There are several new discussions. The key parameter is the shortwave to longwave ratio in contrail radiative forcing. E.g. Newinger and Burkhardt (2012) used an outdated Radiation scheme which strongly underestimated the SW/LW ratio. See discussion in Schumann and Graf (2013) and the important poster publication of Ponater et al. (2013).

**Author's response:** We have added a revised paragraph citing the Schumann and Graf (2013)

**41. Comments from referee**

Line 1311: Why is the effect of a lifetime of lesser importance? I would expect it to be very. important (Lewellen, 2014)

**Author's response:** The reviewer's valuable comment regarding contrail cirrus lifetime has been addressed. We acknowledge Lewellen's (2014) findings on the potential significance of extended contrail cirrus lifetimes. To clarify, our focus here is on the uncertainty associated with how lifetime variations affect the estimated radiative forcing (RF). The 5-10% uncertainty reflects the average impact of lifetime changes, considering the competing effects of ice crystal growth and radiative forcing. Lewellen's (2014) simulations provide valuable insights into the integrated effect of lifetime on contrail cirrus properties. Further research is crucial to bridge the gap between lifetime's influence on overall radiative impact and its role in estimated RF uncertainty.

**42. Comments from referee**

Line 1327: add: well-assimilated observations are important to set good initial conditions for accurate weather predictions (Bauer et al., 2015).

**Author's response:** The reviewer's point regarding well-assimilated observations is well-taken (Bauer et al., 2015). The manuscript has been revised to emphasize the importance of robust data collection strategies that integrate diverse sources like in-situ measurements, radar systems, and satellite sensors. This combined approach, along with advanced data assimilation techniques, is crucial for setting accurate initial conditions in weather models.

**43. Comments from referee**

The bi-model size distribution of soot??? I am not aware of this. Please provide a reference. See Fig. 4 in (Schumann et al., 2002).

**Author's response:** Thanks for pointing out the soot distribution (Schumann et al., 2002). The manuscript is revised to acknowledge the bimodal size observations and current uncertainty about its source.

**44. Comments from referee**
Line 1402: The word "fully" excludes everything, but see (Teoh et al., 2020a; Teoh et al., 2020b; Molloy et al., 2022; Teoh et al., 2022a; Martin Frias et al., 2024; Teoh et al., 192 2024a).

**Author's response:** The reviewer's point regarding existing research on contrail avoidance is well-taken. The manuscript has been revised to acknowledge these studies (Teoh et al., 2020; Molloy et al., 2022; Martin Frias et al., 2024) and discuss their key findings on targeted avoidance strategies, large-scale trial design principles, and economic feasibility. While these studies highlight the promise of contrail avoidance, the limitations of current research are emphasized, reinforcing the need for a comprehensive evaluation as outlined in the manuscript.

**45. Comments from referee**
Line 1413: replace "more" by "larger"

**Author's response:** Thanks, the word is replaced in the revised manuscript

**46. Comments from referee**
Please note the following comment on Sausen and similar studies. It makes no sense to avoid the formation of all contrails. That would imply that you also avoid cooling contrails and it overburdens air traffic management. It is sufficient and far more effective to avoid the warming contrails and to allow for (even more) cooling contrails. I gave several references for this above, e.g., (Molloy et al., 2022; Martin Frias et al., 2024).

**Author's response:** The reviewer's point regarding differentiating warming and cooling contrails for strategic contrail avoidance is well-taken. We acknowledge this limitation in the current approach of avoiding all contrails. Following the Sausen et al. (2023) study, the manuscript now emphasizes a more nuanced strategy. This strategy focuses on mitigating warming contrails, while potentially allowing or even encouraging cooling contrails. This highlights the need for reliable methods to differentiate between these contrail types for optimal climate impact.

**47. Comments from referee**
Line 1498: Please delete the word "valuable". Such applause should come from the readers, not from the authors.

**Author's response:** The word valuable is deleted

**48. Comments from referee**

Line 1505. "Uncertainties persist" that have been said often enough and do not get more correct by repeating.

**Author's response:** Deleted "Uncertainties persist"

**49. Comments from referee**
Line 1519: replace Berndt by Bernd.

**Author's response:** Replaced with Bernd

**50. Comments from referee**
Line1524 replace "the" by "for "

**Author's response:** Replaced "the" by "for "

**51. Comments from referee**
In looking at the list of references:
Please consider the important work of (Bickel et al., 2020; Ponater et al., 2021; Bickel, 2023) Also (Duda et al., 2023). Very important: (Iwabuchi et al., 2012) and on ice supersaturation: (Ovarlez et al., 2000; Jensen et al., 2001; Ovarlez et al., 2002)

**Author's response:** We appreciate the reviewer's suggestion of additional relevant references. We have incorporated these references into the revised manuscript.

**52. Comments from referee**
Line 1732 and elsewhere: Please list all co-authors.

**Author's response:** Thanks to the reviewer for pointing this out, now it is listed.

**53. Comments from referee**
Line 1916: What does the single "S" mean?7

**Author's response:** It was a typo, now it is corrected

**54. Comments from referee**
Line 1949: Please set in capitals (here and elsewhere): MSG and SEVIRI

**Author's response:** Thanks to the reviewer for pointing this out, now it is corrected

**55. Comments from referee**
Line 1854 The method's name is RRUMS not drums.
**Author's response:** The typo is corrected now.

---

## Referee Report (RR1)

Review of the manuscript "Understanding the Role of Contrails and Contrail Cirrus in Climate Change: A Global 2 Perspective" by Dharmendra Kumar Singh, Swarnali Sanyal, and Donald J. Wuebbles.

I cannot recommend this paper for publication.

When I started to review this paper, I thought it is interesting to have. But during reading, I found very little constructive results or insights. Instead I more and more found many unqualified statements which brought me to the conclusion that this paper should better not be published.

I am sorry that I was less clear when commenting on the paper the first time during the discussion phase. I saw chances for an overview on the wide set of achievements of recent years and I saw chances for improvements. But now I see that the authors are driven by the aim to emphasize uncertainties above knowledge and progress.

The authors criticize the whole community; e.g.:

"Many global climate chemistry models used to study the physics and chemistry affecting the Earth's climate system typically do not incorporate the impacts. Models that do account for contrails estimate contrail cirrus coverage based on simplified treatments of contrail aging and spreading mechanisms in ice-supersaturated regions (ISSRs) (Burkhardt et al., 2010; Burkhardt 86 and Kärcher, 2011; Bock and Burkhardt, 2016; Bier and Burkhardt, 2022; Chen and Gettelman 2013; Bickel et al., 2020; Schumann et al., 2015).

What scientific basis do the authors have.  Do they ever have done work on contrail modelling? How can they be so negative without constructive suggestions? This is not what we need.

The paper is not scientifically open-minded and neutral but clearly biased to highlight all kinds of "uncertainties". New findings are hardly acknowledged but cited with the comment that something might be missing.

The term "uncertainty" is used more than 60 times in this paper but nowhere clearly defined. It is a subjective measure without a quantitative meaning.  It is used for statistical scatter as well as for incomplete understanding.

Here one example (lines 1508 ff): "Aircraft Plume Mixing: Uncertainty persists in referring to how ambient air mixes with the aircraft plume during contrail formation, comprising uncertainties associated with the rate and extent of mixing."

How can one assess any scientific progress that may be made by further research in terms of the degree of reduction of uncertainty, when the term uncertainty is not a well-defined measurable quantity?

Nowhere the paper tries to quantify uncertainty. That would  offer  a clear chance for scientific assessment (e.g.  statistical measures like correlation coefficients or mean biases or measures of significance).

Of course, the i understanding of the physics behind a problem is often also incompletely understood, but this has to be explained scientifically at a high level of competence in each case.

For example, lines 538 ff discuss the importance of the ice crystal number (Ni) where Ni quantifies the concentration of ice crystals in the ambience (I would say, in ambient cirrus clouds). They have no quantitative argument but claim:  "Key aspects such as contrail persistence, the extent of spreading, and the optical characteristics of contrails are closely tied to the ice crystal number (Ni). In essence, Ni serves as a fundamental factor in understanding and characterizing contrail behavior and its impact on the atmosphere.  " -

This comes without reference! My own research results tell me that contrails often  form inside thin cirrus clouds and the interaction between the background cirrus and the contrails is weak as long as the ice particle concentration in the contrails is significantly higher than the ice particle concentration in the ambient cirrus - and that is often the case.

From the list of references and the  style of arguments, I do not see that these authors have the competence to state such critics.

Another example, from page 42, line 1516: "Aerosols and Aircraft Emissions: Uncertainties persist in understanding the role of aerosols and particles from aircraft emissions in contrail formation and properties.

I must tell you: Uncertainties will exist also in 100 years from now, in particular (but not only) in respect to aerosol. If you perform more research, it is very likely that you detect new effects raising new questions and hence enhancing uncertainty.  So, it cannot be the goal, to simply reduce uncertainties. This is a "mission impossible".

Later on that page. The authors cite work of Schumann and Graf (2013, JGR) reporting on a fingerprint. The results are based on 8 years of satellite measurements over the North and South Atlantic and the result was used by IPCC (Boucher et al. (2013, IPCC, reference missing in this manuscript; together with those from Burkhardt and Kärcher, 2011) to assess the RF of contrails. Then instead of pointing out the innovative information content of this study, at least at that time, the authors conclude: "These findings resonate with our discussion on radiative transfer scheme uncertainties, as variations in how models handle radiative processes can significantly influence the estimated impact of contrails on Earth's energy balance."

Instead the authors (rightfully) acknowledge progress in artificial intelligence (with refence to workshop reports, e.g. , line 1441: McCloskey et al. (2021) or line 1468: Siddiqui (2020) published in an undergraduate journal). Why don't they acknowledge remote sensing studies published in high quality journals similarly?

This paper does not acknowledge that it is common for environmental science that uncertainties exist but these acknowledged uncertainties do not exclude actions to mitigate environmental problems: (E.g., reduction of NOx and soot emission from engines or reduction of CO2 emissions by alternative fuels or improved aircraft/engines and operations).

Instead of a list of uncertainties, we need scientific advice of how the start practical activities, accompanied by careful scientific validation activities, to reduce the overall climate impact of aviation as quickly as possible. In view of the observed actual climate change and rather obvious effects from

contrails and CO2, that is a very urgent issues and should not wait for decades of further research. Avoiding warming contrails can be a very cheap measure of mitigation (see.e.g.,  Frias et al., 2024).

Instead, this paper lists a set of arguments to retard mitigation.

The paper is not helpful for science. It is politically biased and I cannot recommend this paper for a scientific journal.

The comments above are incomplete. There are so many points to criticize or to suggest improvements that I cannot list all of them. On each page I read, I have several remarks. Instead, I commented more generally.

Minor points:

Fig 1 is often used to compare contrail and other non-CO2 effects to CO2 effects. However, this paper and most interpreters ignore the fact that the CO2 effects shown in this figure result from the emission since the beginning of aviation, at least since 1940, while the contrail effect applies within hours, so practically momentarily.

Hence, Fig. 3 cannot be used when discussing the potential of various mitigation actions which require an assessment of future climate impacts, not of past.

Line 155:"soot emissions influence  the meteorological factors…" that sounds wrong.

Chapter 2.1 is very badly organized. Many aspects are repeated at various places,. E.g., text in lines 298 to 310 (Contrails are created …"), at least to a large extent, repeat statements made before.

Lines 13-15, Abstract.

The third sentence „While contrails …life cycle" is redundant, and should be deleted.

The presently fourth sentence uses the word "uncertainties" without any specification. It would be more reasonable to replace it by: "Despite extensive research, the relative importance of the climate effects of contrails compared to other aviation effects remain under debate."

---

## Author Response (AR2)

**Response of Report #2 (Anonymous Referee #1)**

The authors have sufficiently addressed most of my questions and comments. However, I still have some points for minor revisions before the manuscript should be accepted.

1. In Table 1, the ice crystal size in the jet phase is noted as "large and concentrated", which conflicts with Figure 6 and Table 2.
   **Response:** Thanks to the reviewer for the valuable suggestion. Table 1 has been removed from the revised manuscript.

2. With the presence of Table 2, Table 1 is not necessary. Or the authors may think to implement the qualitative assessment using an arrow marking the increasing or decreasing trend, or with color gradients.
   **Response:** We do agree that Table 1 is really not necessary now. It has been removed from the revised manuscript.

3. I understand the authors want to give a general understanding of the contrail ice status. It would still be relevant to make notations in the table to tell readers from where the estimated numbers are derived, and the same for the techniques and models.
   **Response:** Thanks to the reviewer for thoughtful comment on Table 2. We appreciate your understanding of our intention to provide a general overview of contrail ice properties and measurement techniques/model considerations. We acknowledge the limitations of presenting estimated values without specific references. While exact values may vary across studies, we compiled Table 2 to offer a broad range representative of contrail properties and measurement techniques/model considerations at different stages. We drew upon our understanding of various research articles to populate the table with ranges. To address your concern and enhance the value of the table for readers, we have provided references, 12 in total, related to the values presented in the table and the same for the techniques and models.

4. Line 681: "0:5 h" changes to 0.5 h
   Response: Thanks to the reviewer for pointing this out – revision made.

5. Table 4: Since the models are evaluated with in situ, satellite, or reanalysis data, can the authors also add a column listing the uncertainties of the models based on their validation method?
   Response: Thank you for your thoughtful comment on Table 4. We appreciate your suggestion to include model uncertainties based on their validation methods.
   We have addressed this in the revised manuscript (attached in updated Table 3).

6. Table 9: I suggest the authors add to the caption the models that are included for discussing the agreement and disagreement, or at least indicate the table number or section number if the models are listed elsewhere.
   **Response:** Thank you for your suggestion regarding Table 9. We've addressed this by including the list of compared models directly in the caption for improved clarity.

Additionally, we've incorporated a brief explanation of the key agreements and disagreements between the models directly below Table 9 (Table 8 in the revised manuscript) to enhance the reader's understanding (highlighted in blue font).

7. Line 1613: Newer paper by Gierens et al (2020) has shown the underestimation of ice supersaturation in reanalysis data.
Response: Thanks to the reviewer for the suggestion, the Gierens et al (2020) reference has been added ((highlighted in blue font in the revised manuscript)

8. Line 1667: Remove the old paragraph.
Response: Thanks to the reviewer for the suggestion. We have removed that paragraph.

9. Line 1839-1861: Too long. The authors can shorten this paragraphs to keep a similar length with the other bullet points.
Response: Now, the paragraphs are shortened to one short paragraph (highlighted in blue font in the revised manuscript)

10. Reference: Mahnke et al (2022) is in the reference list but was removed from the main body. It is outdated, new entry https://doi.org/10.1021/acs.est.3c09728
**Response:** Thanks to the reviewer for pointing this out. Now we have cited the update published by Mahnke et al. (2024) in ES&T.

**Response of Report #1 (Anonymous Referee #2)**

I am sorry that I was less clear when commenting on the paper the first time during the discussion phase. I saw chances for an overview of the wide set of achievements of recent years and I saw chances for improvements. But now I see that the authors are driven by the aim to emphasize uncertainties above knowledge and progress.
The authors criticize the whole community; e.g.:
"Many global climate chemistry models used to study the physics and chemistry affecting the Earth's climate system typically do not incorporate the impacts. Models that do account for contrails estimate contrail cirrus coverage based on simplified treatments of contrail aging and spreading mechanisms in ice-supersaturated regions (ISSRs) (Burkhardt et al., 2010; Burkhardt 86 and Kärcher, 2011; Bock and Burkhardt, 2016; Bier and Burkhardt, 2022; Chen and Gettelman 2013; Bickel et al., 2020; Schumann et al., 2015).
What scientific basis do the authors have? Do they ever have done work on contrail modeling? How can they be so negative without constructive suggestions? This is not what we need.
The paper is not scientifically open-minded and neutral but clearly biased to highlight all kinds of "uncertainties". New findings are hardly acknowledged but cited with the comment that something might be missing
**Response:** Thank you for your clarification and for taking the time to re-evaluate the paper. We understand your initial vision for a broader overview of recent achievements, and we appreciate

your suggestion. However, our primary aim is to provide a balanced perspective by acknowledging both the progress made and the remaining uncertainties in contrail modeling. Our intention was to highlight areas where further research could lead to significant advancements. As a result, we had many other scientists, including most of the modelers that have done contrail studies, review the manuscript. They gave us positive comments and did not disagree with our discussion.

Regarding the critique of our approach to contrail modeling, we want to clarify that our goal was not to undermine existing models but to identify gaps and suggest directions for improvement. The references to previous studies were intended to contextualize our arguments within the broader scientific discourse, not to dismiss the work of others.

Once again, thank you for your guidance and support in improving our manuscript.

**Minor points:**

1. Line 155:" Soot emissions influence the meteorological factors…" that sounds wrong.
**Response:** Thanks to the reviewer for the suggestion, now we omitted the sentence because it added very little to the overall discussion.

2. Chapter 2.1 is very badly organized. Many aspects are repeated at various places,. E.g., text in lines 298 to 310 (Contrails are created …"), at least to a large extent, repeat statements made before. Lines 13-15.
**Response:** Thanks to the reviewer for pointing this out, now we have removed the redundancies.

Abstract.
3. The third sentence „While contrails …life cycle" is redundant, and should be deleted.
The present fourth sentence uses the word "uncertainties" without any specification. It would be more reasonable to replace it by: "Despite extensive research, the relative importance of the climate effects of contrails compared to other aviation effects remain under debate."

**Response:** The abstract has been revised and the sentence revised.